



# Prognostic simulations of mixed-phase clouds with model 1D-AC v1.0: The impact of freezing parameterizations on ice crystal budgets

Yijia Sun[1], Ann M. Fridlind[2], Israel Silber[3], Nicole Riemer[4], Daniel A. Knopf[1]

[1]Stony Brook University, Stony Brook, NY 11794, USA
5 [2]NASA Goddard Institute for Space Studies, New York, NY 10025, USA
[3]Pacific Northwest National Laboratory, Richland, WA 99352, USA
[4]University of Illinois at Urbana-Champaign, Urbana, IL 61801, USA

*Correspondence to*: Daniel A. Knopf (daniel.knopf@stonybrook.edu)

10 **Abstract.** Mixed-phase clouds at high latitudes contribute to the uncertainty in predicting cloud feedbacks and climate sensitivity, mainly due to the complexity of microphysical processes that influence the partitioning between the supercooled liquid and ice phases, and hence, cloud radiative effects on regional scales. Particularly in Arctic mixed-phase clouds, the activation of ice-nucleating particles (INPs) from various aerosol populations remains a leading source of uncertainty. Our study employs a one-dimensional aerosol-cloud model informed by large-eddy simulations to probe the impact of INP 15 representation on predicted ice crystal number concentrations ($N_i$) and ice crystal budgets in mixed-phase Arctic stratus. We apply three immersion freezing (IMF) parameterizations, two time-independent (deterministic) and one time-dependent (classical nucleation theory), to predict the evolution of the INP reservoir and resulting ice crystal budget from polydisperse mineral dust, organic (humic-like substances), and sea spray aerosol particle size distributions. Our analysis focuses on how variations in aerosol number concentration and cloud system parameters such as cloud cooling rate, cloud-top entrainment 20 rate, and ice crystal fall speed influence the INP reservoir and ice crystal budgets. Furthermore, this study investigates the competitive ice nucleation dynamics in mixed aerosol environments and provides a process-level quantification of the INP budget terms, which directly controls ice crystal budgets. For all studied case scenarios, the aerosol types and associated particle size distributions significantly impact INP and $N_i$, and the choice between a time-dependent and a deterministic freezing description yields orders-of-magnitude differences in the predicted INP and $N_i$ over the 10 h simulation time, 25 reflecting typical cloud lifetimes. Our results show that the influence of cloud cooling, INP entrainment, and sedimentation varies significantly depending on the chosen freezing parameterization. These findings underscore the critical need for robust IMF parameterizations and precise cloud system observations to enhance the accuracy of models in predicting mixed-phase cloud structure and evolution.



## 1 Introduction

In the Arctic regions, the rate of warming has been at least twice as large as the global mean since pre-industrial times (Holland et al., 2003; Serreze et al., 2011; Gulev et al., 2021), a phenomenon known as "Arctic amplification" (Hahn et al., 2021; Morice et al., 2021) that coincides with a drastic decrease in the cover of sea ice (Stroeve et al., 2012; Richter-Menge, 2018). Stratus,
a predominant cloud type in polar regions (Shupe et al., 2008; Shupe, 2011; Andronache, 2018; Lubin et al., 2020), plays a crucial role in the surface and top-of-atmosphere radiative budget due to its extensive prevalence (eg., Dong et al., 2003; Zuidema et al., 2005). These clouds control the development of precipitation through their liquid and ice phases (eg., Field et al., 2015; Mülmenstädt et al., 2015; Korolev et al., 2017; Silber et al., 2021). Arctic stratus clouds are predominantly mixed-phase clouds (MPCs) (Curry et al., 2000; Korolev et al., 2003; Shupe, 2011), characterized by the presence of at least one
supercooled liquid water layer where ice crystals form and sediment subsequently (Shupe et al., 2006; Morrison, 2012; Silber et al., 2021). The persistence of this mixed-phase state is governed by a delicate interplay between cloud dynamics and microphysics, with the ambient ice concentration determining whether the cloud remains liquid-bearing or glaciates (Khain et al., 2022).

In these Arctic MPCs, ice formation typically occurs from the supercooled liquid phase through heterogeneous ice nucleation.
Heterogeneous ice nucleation can commence via several modes including immersion freezing (IMF), where freezing is initiated on the surface of an ice-nucleating particle (INP) that is immersed in a supercooled aqueous solution droplet (Vali et al., 2015). Recent studies have indicated that IMF is the dominant pathway of primary ice formation in MPCs (Ansmann et al., 2008; Prenni et al., 2009; Hoose et al., 2010; De Boer et al., 2011; Hoose et al., 2012b; Murray et al., 2012; Westbrook et al., 2013; Kanji et al., 2017; Silber et al., 2021; Burrows et al., 2022).
To accurately predict the number concentration of ice nucleating particles (INPs) from aerosol particles, IMF parameterizations are needed. Over the past decades, IMF parameterizations representing specific particle types have been developed (Bauer et al., 2008; Murray et al., 2012; Niemand et al., 2012; Knopf et al., 2013; Demott et al., 2015; Liu et al., 2016; China et al., 2017; Kanji et al., 2017; Knopf et al., 2018; Mccluskey et al., 2018; Penner et al., 2018; Alpert et al., 2022; Burrows et al., 2022; Knopf et al., 2023a). Typically applied freezing parameterizations include deterministic schemes based on the singular
hypothesis (Levine, 1950; Vali, 1971; Vali, 2014), and stochastic schemes, such as those based on classic nucleation theory (CNT) (Pruppacher et al., 2010; Knopf and Alpert, 2023a).

Deterministic IMF parameterizations are founded on the assumption that ice nucleation active sites (INAS) initiate ice nucleation instantaneously when a specific temperature is reached for each INAS on an immersed particle surface. This freezing efficiency can therefore be described as a function of temperature and particle type including particle number
concentration and size. The deterministic description is thus time independent, thereby neglecting the cooling rate dependence in ice nucleation observations (Bigg, 1953; Pruppacher and Klett, 2010; Alpert et al., 2016; Knopf et al., 2020; Arabas et al., 2025).





The CNT-based parameterizations make no claim of specific INAS initiating freezing but assume that ice nucleation proceeds stochastically on the INP surface, and increasing INP surface area will therefore increase the freezing rate (Pruppacher and Klett, 2010; Knopf et al., 2020; Knopf and Alpert, 2023a). As such, CNT-based parameterizations are time dependent. As a consequence, at constant supersaturation with respect to ice, as time progresses, more freezing events are observed in accordance with experimental studies (e.g., Biermann et al., 1996; Koop et al., 1997; Alpert and Knopf, 2016; Knopf et al., 2020; Deck et al., 2022).

Mineral dust, organic, and sea spray aerosol (SSA) particles are common aerosol types in the Arctic region (Udisti et al., 2020; Schmale et al., 2021). At temperatures below -20 °C, mineral dust has a particularly high ice nucleating efficiency (Kanji et al., 2017) and is assumed to be the best understood source of INPs (Burrows et al., 2022). The variability of atmospheric INP concentrations can, in cases, be attributed to the change in number concentration of long-range transported dust (Demott et al., 2003; Chou et al., 2011; Boose et al., 2016; Burrows et al., 2022; Shi et al., 2022; Kawai et al., 2023). Organic aerosol (OA) particles can be directly emitted from fossil fuel combustion and biomass burning, also termed primary organic aerosol (POA). The condensation of oxidized volatile organic compounds (VOCs) can yield secondary organic aerosol (SOA) particles (Hallquist et al., 2009; Shrivastava et al., 2017; Bergman et al., 2022; Srivastava et al., 2022). The organic matter (OM) associated with OA particles that initiate the ice nucleation in the atmosphere is still not well understood (Knopf et al., 2018). OM can exhibit various phase states at the same atmospheric thermodynamic conditions, resulting in different ice nucleation pathways and ice nucleation rates (Knopf et al., 2018). Through recent field measurements and laboratory investigation, it has been pointed out that particulate OM can contribute to atmospheric ice nucleation (Knopf et al., 2010; Wang et al., 2012a; Wang et al., 2012b; Hiranuma et al., 2013; Knopf et al., 2014; Alpert and Knopf, 2016; Knopf et al., 2020; Xue et al., 2024). Several studies have demonstrated that in remote marine regions far from the influence of continental INP sources, SSA particles can act as a significant INP source (Burrows et al., 2013; Wilson et al., 2015; Vergara-Temprado et al., 2017; Huang et al., 2018; Mccluskey et al., 2018; Mccluskey et al., 2019; Zhao et al., 2021; Alpert et al., 2022; Raatikainen et al., 2022; Xue et al., 2024). Soot particles, which have also been discussed as a potential source of anthropogenic INPs. Under polluted conditions, relatively high black carbon fraction has been found in both cloud residuals as well as simulated results (Savre et al., 2015). However, recent laboratory studies reveal that soot exhibits minimal efficacy in immersion-mode freezing (Friedman et al., 2011; Schill et al., 2018; Kanji et al., 2020; Schill et al., 2020). Its contribution to atmospheric ice formation at temperatures above −38°C is likely not significant (Kanji et al., 2020).

In addition to ambient particle types, typical aerosol particle size distributions (PSDs) present in the Arctic are relevant to ice formation rates. The PSDs determine the particle number and surface area concentration, a necessary input for the IMF parameterizations, thus influencing the number concentration of INPs and, ultimately, ice crystals. In this study, we apply the PSDs derived from Indirect and Semi-Direct Aerosol Campaign (ISDAC) (Earle et al., 2011; Hiranuma et al., 2013) and the International Chemistry Experiment in the Arctic LOwer Troposphere (ICEALOT) (Quinn et al., 2017), both carried out in the Arctic region. Detailed information can be found in the methods section.



Our study is motivated by the fact that the inferred strength of ice production within Arctic stratus MPCs has often been challenging to reproduce by models applying a prognostic INP treatment, i.e., a treatment where the budget of INP is tracked (Harrington et al., 2001; Morrison et al., 2005; Fridlind et al., 2007; Fan et al., 2009; Avramov et al., 2011; Fridlind et al., 2012). Such a treatment contrasts with simpler diagnostic treatments of INP that enable physically unrealistically high ice

formation rates to persist indefinitely (e.g., Knopf et al., 2023b), and our previous work indicates that the type of IMF parameterization (deterministic or CNT-based) plays an important role in the strength of ice formation rates under Arctic stratus conditions (Knopf et al., 2023b; Arabas et al., 2025). In general, when assuming a deterministic freezing scheme, INP depletion from a turbulently mixed stratus layer occurs within a short period of time, on the order of the mixing time for a well-mixed cloud-topped boundary layer. This rapid depletion results from the INP exposure (via turbulent transport) to

conditions that result in their instantaneous activation at the coldest conditions within the cloud layer (within cloud tops). Subsequently, cloud-top entrainment or cooling as drivers of INP activation have often appeared too weak to explain observed ice crystal loading (Fridlind et al., 2012; Westbrook and Illingworth, 2013). Other processes may contribute to sustaining INP under some specific conditions, such as entrainment of INP from below the turbulently mixed layer if the cloud-containing layer is decoupled from the surface (Avramov et al., 2011) or restoration of some of the activated INP via complete ice crystal

sublimation within a turbulently mixed layer if there is an ice-subsaturated layer (at the base) that is deep enough to enable that (Solomon et al., 2015). Under the simplest conditions without such processes possible, Fridlind et al. (2012) demonstrated how unrealistically high concentrations of INPs from the free troposphere were needed to sustain inferred ice production via cloud-top entrainment when applying a deterministic IMF parameterization. Unfortunately, direct constraints on INP budgets have not been possible from available data sets owing at least in part to a lack of INP data within cloud layers, and inferring

INP budgets from ice crystal number concentrations invokes an additional set of large uncertainties emerging from all of the factors that influence ice crystal PSDs in Arctic MPCs that must also be parameterized (Fridlind et al., 2007; Avramov et al., 2011; Fridlind et al., 2012; Ovchinnikov et al., 2014; Morrison et al., 2020).

In most previous modelling studies, prognostic ice nucleation has been parameterized with deterministic IMF schemes (Morrison et al., 2005; Avramov et al., 2011; Fridlind et al., 2012; Solomon et al., 2015; Tully et al., 2023). It has been pointed

out that such deterministic parameterizations may not be applicable to all atmospherically relevant conditions (temperature and humidity) (eg., Niemand et al., 2012; Hiranuma et al., 2014; Ullrich et al., 2017), partly due to instrument limitations and the limited amount of data collected (Burrows et al., 2022), and this may include slowly continuing ice formation processes that may sustain continuous ice crystal production (Westbrook and Illingworth, 2013; Yang et al., 2013). Several studies have implemented CNT-based parameterizations (Savre and Ekman, 2015; Raatikainen et al., 2022; Shi et al., 2022) to investigate

the effects of different INPs on Arctic MPCs. Savre and Ekman (2015) found that the application of an evolving α-PDF scheme (introducing a contact-angle distribution function that scale the freezing efficacy of the INPs) can support continuous in-cloud ice production controlled mostly by the competition between cloud cooling, cloud-top entrainment, and ice sedimentation in three simulated cases. Raatikainen et al. (2022) showed that the relative significance of marine INP emissions and accounting for INP recycling was crucial for maintaining MPCs in their simulations over water rather than ice surfaces. Knopf et al.



(2023b) pointed out another possible explanation that the selection of IMF parameterizations determines the size of the INP reservoir in Arctic stratus, with the CNT-based parameterizations producing a reservoir that is several orders of magnitude larger than deterministic parameterizations, thereby becoming the dominant factor for sustained strength of ice crystal formation.

In this study we employ a simplistic 1D aerosol-cloud (1D-AC) model informed by observationally-constrained LES results

to examine the INP reservoir dynamics of Arctic stratus within a well-mixed cloud-topped boundary layer over sea ice (Knopf et al., 2023b). Our 1D-AC model simulates an evolving vertical column that advects with the mean horizontal wind, thereby approximating a Lagrangian framework to study the INP reservoir in long-lived Arctic clouds. A key simplification of the 1D-AC model is that the liquid phase is taken as a fixed quantity. This approach decouples ice microphysics from liquid-phase feedbacks, enabling a direct attribution of changes in the ice crystal budget solely to the immersion freezing parameterization

under investigation (Knopf et al., 2023b). This idealization is physically justified by observations of long-lived Arctic MPCs, which are often characterized by a quasi-steady liquid water path. This state is maintained by very weak ice precipitation rates that only minimally desiccate the cloud, a feature identified in both specific cases (eg., Fridlind et al., 2012) and statistically (eg., Silber et al., 2021). The objective of 1D-AC is to predict the co-evolution of the size- and composition-distributed aerosol acting as INP and the activated $N_i$ profiles in a simplified framework as a function of specified values of the boundary layer

turbulent mixing time scale, cloud-top entrainment rate, cloud cooling rate (CCR), and number-weighted ice crystal fall speed (cf. Fridlind et al., 2012). Knopf et al. (2023b) demonstrate how these specified values exhibit markedly differing controls on ice formation depending on which type of ice nucleation scheme is implemented. Different IMF parameterizations derived from the same freezing experiments will scale differently when predicting INPs for conditions other than those of the experiment (Knopf et al., 2021).

Here we extend the Knopf et al. (2023b) study by investigating the impact of different aerosol types and associated PSDs on INPs, IMF parameterizations, and various microphysical and model parameters on the evolution of INP and $N_i$. To account for the diversity of IMF parameterizations, including the ice nucleation number (INN)-based parameterization (Demott et al., 2010; Demott et al., 2015), INAS parameterization (Niemand et al., 2012; China et al., 2017) and water-activity-based IMF model (ABIFM) parameterization derived from CNT (Knopf and Alpert, 2013; Alpert and Knopf, 2016). We further initialize

the model with three different aerosol particle types and respective PSDs, including mineral dust, organic (humic-like substances) aerosol particles, and SSA particles, guided by observations in the Arctic regions. Each of these parameterizations requires different sets of parameter inputs, which will be discussed in detail in the methods section. The effect of varying aerosol number concentration and specified cloud system parameters on the INP reservoir and $N_i$ are also assessed. Furthermore, to assess the model's capability in handling more complex atmospheric compositions and to explore the

competitive interplay of different INP types, MPC scenarios incorporating co-existing externally-mixed mineral dust, organic, and sea spray aerosol populations are also investigated. The paper is organized as follows. In Sect. 2 we describe our implementation of the 1D-AC model and all simulation cases in detail. In Sect. 3 we present the sensitivity simulation results, also the findings from combined aerosol scenarios and a detailed INP budget analysis, followed by a discussion in Sect. 4.



Finally, the key findings are summarized and discussed in Sect. 5. Appendix A provides a detailed description of the IMF
parameterizations. Appendix B details the INP array calculation in our model for deterministic approaches. The governing
equations are presented in Appendix C. For a comprehensive list of symbols and abbreviations, refer to Appendices D and E,
respectively.

## 2 Methods

The 1D-AC model is currently based on cloud system conditions observed in a well-mixed cloud-topped boundary layer over
sea ice during the Surface Heat Budget of the Arctic campaign (SHEBA) (Knopf et al., 2023b). In this section, we briefly
describe the SHEBA case study. Then we introduce the applied aerosol types and corresponding PSDs and the governing
equations of the 1D-AC model. Lastly, we summarize the different simulations considered in this study.

### 2.1 The SHEBA case study

As the initial setup in our 1D-AC model, we use an Arctic MPC case study that was well observed during the SHEBA campaign
(Curry et al., 2000), and that has been widely used for advancing our understanding of MPC conditions (Morrison et al., 2011;
Fridlind et al., 2012). The campaign was carried out at roughly 76°N, 165°W in the Beaufort Sea, Arctic Ocean, approximately
570 km northeast of Prudhoe Bay, Alaska. This case study was characterized by widespread, long-lived, shallow MPCs
coupled to an underlying pack ice with cloud tops of approximately 600 to 400 m as indicated by Millimeter Wavelength
Cloud Radar (MMCR) echoes (Morrison et al., 2011; Fridlind et al., 2012). Rawinsondes provided profiles of temperature and
180 relative humidity (RH), indicating a well-mixed liquid cloud-topped boundary layer. Two-dimensional cloud optical array
probes (2D-C), two-dimensional precipitation optical array probes (2D-P) and the Cloud Particle Imager (CPI) yielded
reconstructed cloud particle shapes and sizes (Fridlind et al., 2012).

Compared to conditions observed during the Indirect and Semi-Direct Aerosol Campaign (Mcfarquhar et al., 2011) or the
Mixed-Phase Arctic Cloud Experiment (Verlinde et al., 2007), this SHEBA case provides a simple starting point owing to ice
supersaturated conditions from liquid cloud top to the near-surface as shown in Fig. 1, low liquid water path, high droplet
number concentration around 200 cm⁻³, an absence of liquid-phase precipitation or active ice aggregation, and relatively sparse
concentrations of unrimed and nondendritic ice crystals (Fridlind et al., 2018). These conditions are not expected to support
secondary ice production (SIP), which is commonly assumed to proceed by rime splintering (Hallett et al., 1974) or freezing
fragmentation (Lauber et al., 2018; Keinert et al., 2020), but is currently debated for its relative role in ice production process
in MPCs (Phillips et al., 2017; Luke et al., 2021; Waman et al., 2023; Zhao et al., 2023). Robustly observing SIP in both field
and laboratory studies is challenging (Korolev et al., 2020b; Silber, 2023), and large uncertainty remains in simulating SIP
processes (Phillips et al., 2018; Korolev et al., 2020a; Miltenberger et al., 2021; Burrows et al., 2022; Waman et al., 2023).
The apparent occurrence of SIP may be less than 10% in slightly supercooled (warmer than -10°C) Arctic clouds (Luke et al.,
2021) and is expected to be negligible under conditions for this SHEBA case (Fridlind et al., 2012) in contrast to other cases



(Fridlind and Ackerman, 2018). Our setup is therefore chosen to isolate the role of primary ice formation via IMF, which precedes any subsequent SIP in non-seeded Arctic stratus.

## 2.2 Aerosol particle size distributions

To evaluate the impact of different aerosol particle types on the INP reservoir, $N_i$ and ice crystal formation rate, we examine the effects of mineral dust, organic, and SSA particles. During the SHEBA field campaign composition-resolved aerosol PSDs were not measured. Chosen particle types and PSDs, thus, may not represent the ambient aerosol during SHEBA, though all aerosol PSDs are chosen based on measurements performed over Arctic regions. In this sensitivity study, the aim is not having the most realistic representation of the aerosol population for a particular case but to establish if and how different Arctic aerosol types could impact the INP reservoir and $N_i$. For each aerosol type, the applied aerosol particle size distributions (PSDs) are polydisperse consisting of two or three lognormal modes (Table 1). In addition to Aitken and accumulation modes, this framework includes a larger accumulation mode for aged aerosols and a source-specific SSA mode (Quinn et al., 2017).

**Table 1: Summary of the applied particle size distribution parameters for different aerosol types.**

| Aerosol type | $D_1$ (µm) | $D_2$ (µm) | $D_3$ (µm) | $D_{min}$ (µm) | $D_{max}$ (µm) | $\sigma_1$ | $\sigma_2$ | $\sigma_3$ | $N_1^{aer}$ (cm$^{-3}$) | $N_2^{aer}$ (cm$^{-3}$) | $N_3^{aer}$ (cm$^{-3}$) |
|---|---|---|---|---|---|---|---|---|---|---|---|
| Dust | 0.20 | 0.71 | | 0.01 | 17.32 | 1.47 | 2.44 | | 3.47 | 0.33 | |
| Organic | 0.20 | 0.71 | | 0.01 | 17.32 | 1.47 | 2.44 | | 55.20 | 0.50 | |
| SSA | 0.04 | 0.17 | 0.24 | 0.0022 | 5.64 | 1.60 | 1.55 | 2.35 | 38.48 | 119.70 | 12.83 |

The mineral dust and organic PSDs are based on aerosol samples measured by aircraft for single-layer stratocumulus with below-cloud aerosol concentrations less than 250 cm$^{-3}$ (clean cases) as reported in Earle et al. (2011). We use the same lognormal distribution parameters (geometric mean diameter ($D$), geometric standard deviation ($\sigma$)) for mineral dust and organic particles as provided by Savre and Ekman (2015). The aerosol number concentrations for each mode of mineral dust and organic aerosol particles are derived from micro-spectroscopic single particle analysis of ambient particles that were collected by aircraft (Hiranuma et al., 2013). We average particle types from Flight 30-Substrate 6 (F30-S6) during ISDAC to derive respective particle number concentrations. The applied SSA PSD is based on measurements during the International Chemistry Experiment in the Arctic Lower Troposphere (ICEALOT) (Quinn et al., 2017). SSA particles were sampled 18 m above the sea surface. Lastly, to reflect a more realistic aerosol population we combine the mineral dust, organic, and SSA PSD (composite PSD). Figure 1 displays the lognormal PSDs of the different aerosol particle types and the composite PSD, derived from the modal parameters specified in Table 1.



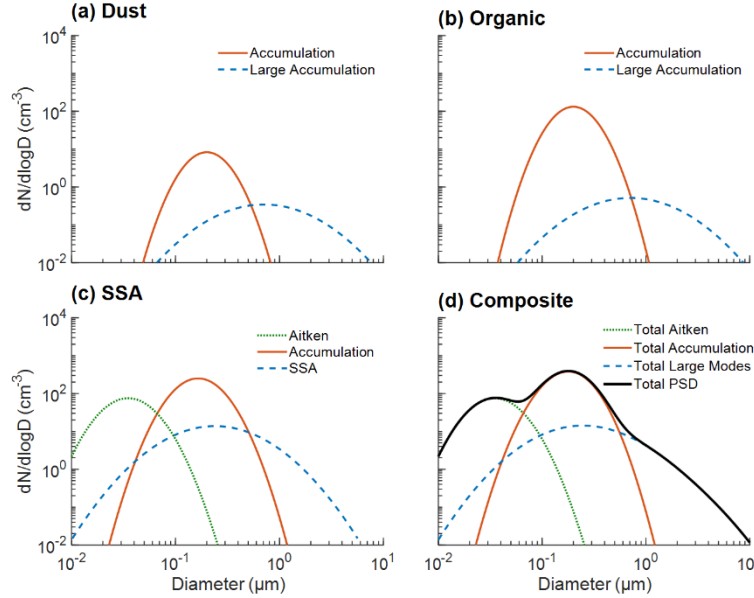

**Figure 1: Aerosol particle size distributions for mineral dust (a), organic particles (b), sea spray aerosol (c) and composite particle size distribution (d) following PSD parameters given in Table 1. Different colors represent different modes for respective aerosol PSDs.**

## 2.3 1D-AC model setup

LES baseline results for the SHEBA case study (Fridlind et al., 2012) using the Distributed Hydrodynamic Aerosol and Radiative Modeling Application (DHARMA) code (Ackerman et al., 2000; Stevens et al., 2002) serve as the source of the cloud layer conditions for our model setup. As noted above, 1D-AC maintains a time-invariant thermodynamic profile, including temperature, relative humidity, and liquid water path. We set the domain height to 390 m with the mixed phase cloud layer located from 262.5 to 390 m determined by thermodynamic conditions obtained from SHEBA LES simulation results (see Figure 2). The temperature decreases roughly linearly with height from -16.2 to -19.8 ℃. In this cloud case, the ice supersaturation ratio is larger than 1 throughout nearly the entire domain (see Figure 2), so ice sublimation of sedimenting ice crystals is negligible. The governing model baseline parameter values of mixing time scale, entrainment rate, number-weighted ice fall speed, simulation time step, and simulation vertical resolution applied in the model are shown in Table 2. IMF parameterizations for respective aerosols are summarized in Table 3 and a detailed description of these three IMF parameterizations is given in Appendix A. Above liquid cloud base, all INPs are assumed to be within liquid cloud droplets. To evaluate freezing at subsaturated conditions, i.e., at RH > 90% in this case (i.e., proceeding from a layer just below the cloud base), assuming the INP is engulfed by an aqueous solution below cloud base, we also include a fourth variation, termed ABIFM*, which permits nucleation to occur in subsaturated conditions (Knopf and Alpert, 2013). The calculation of INP arrays for application of the deterministic freezing parameterizations can be found in Appendix B. Prognostic treatments are used for the number concentrations of activatable INPs (the reservoir of INP that can actually be activated within a given cloud





layer, refer to $N_{\text{INP}}$) and $N_i$ and detailed prognostic equations are given in Appendix C. The stability of our simulation calculation is evaluated by insuring that results are insensitive to a doubling of vertical resolution (5 m) and a much shorter

time step (1 s) (Figure S1). The model framework allows for the simultaneous tracking of multiple aerosol types, each with distinct PSDs and ice nucleation properties, facilitating the External simulations described in Section 2.5. Furthermore, the prognostic equations (Appendix C) enable the diagnosis of individual budget terms for INP number concentration, facilitating the process-level analysis.

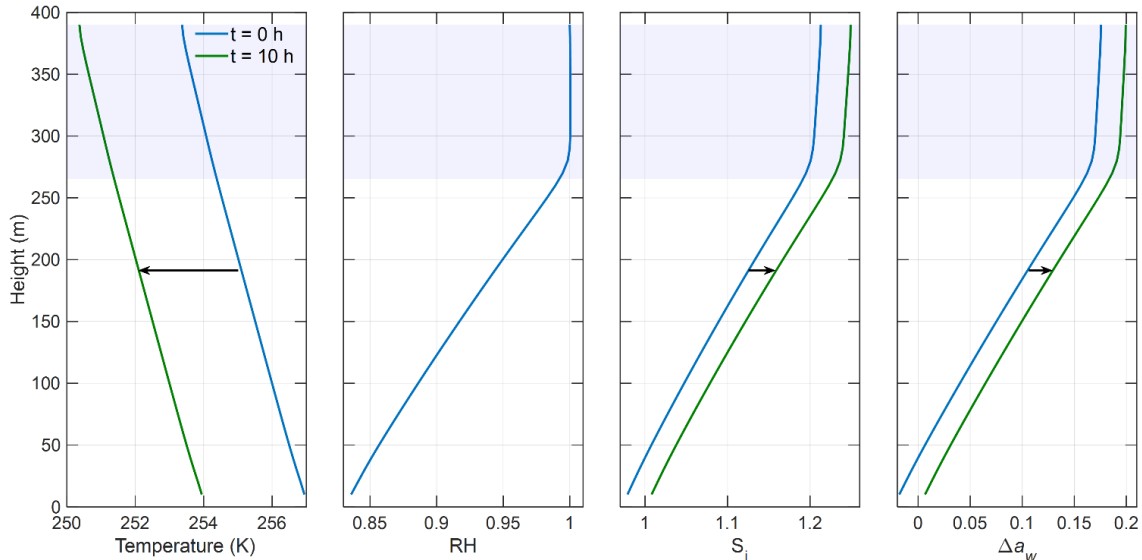

**Figure 2: Thermodynamic conditions applied in the minimalistic 1D aerosol-cloud model. From left to right: The temperature ($T$), the relative humidity (RH), supersaturation with respect to the ice ($S_{\text{ice}}$) and $\Delta a_w$. Blue and green lines represent the initial (t = 0 h) thermodynamic conditions and the thermodynamic conditions with cloud cooling rate of 0.3 °C h$^{-1}$ after 10 h, respectively. The blue shaded area denotes the cloud layer.**

**Table 2: 1D aerosol-cloud model baseline parameters.**

| Model parameters | Control run values |
|---|---|
| Mixing time scale, $\tau_{\text{mix}}$ | 1800 s |
| Entrainment rate, $w_e$ | 0.1 cm s$^{-1}$ |
| Sedimentation rate, $v_f$ | 30 cm s$^{-1}$ |
| Simulation time step, $\delta t$ | 10 s |
| Simulation resolution (vertical), $\delta z$ | 10 m |



**Table 3: Immersion freezing parameterizations used in this study. Input parameters are aerosol number concentration for particles larger than 0.5 μm, $N_{\mathrm{aer}>0.5\mu m}$, aerosol number concentration, $N_{\mathrm{aer}}$, aerosol surface area, $A_{\mathrm{aer}}$, temperature, $T$, and relative humidity, RH. INP array stores the number concentration of activatable INPs using different data structures for different immersion freezing parameterizations. The INP array dimensions include $z_i$, height at grid cell index i, t, time, $d$, INP diameter array, $T$, INP temperature array.**

| Immersion freezing parameterizations | Reference | Types of parameterization | Aerosol type | Input | INP array |
|---|---|---|---|---|---|
| D2010 | *DeMott et al.*, 2010, Eq. 1 | Deterministic (INN) | Organic, SSA | $N_{\mathrm{aer}>0.5\mu m}, T$ | $z_i, t, T$ |
| D2015 | *DeMott et al.*, 2015, Eq. 2 | Deterministic (INN) | Mineral dust | $N_{\mathrm{aer}>0.5\mu m}, T$ | $z_i, t, T$ |
| N2012 | *Niemand et al.*, 2012, Eq. 5 | Deterministic (INAS) | Mineral dust | $N_{\mathrm{aer}}, A_{\mathrm{aer}}, T$ | $z_i, t, d, T$ |
| C2017 | *China et al.*, 2017, Fig. 4b | Deterministic (INAS) | Organic | $N_{\mathrm{aer}}, A_{\mathrm{aer}}, T$, RH | $z_i, t, d, T$ |
| A2022 | *Alpert et al.*, 2022, Fig. 5a | Deterministic (INAS) | SSA | $N_{\mathrm{aer}}, A_{\mathrm{aer}}, T$, RH | $z_i, t, d, T$ |
| ABIFM | *Alpert and Knopf*, 2016, Table 2; *China et al.*, 2017, Fig. 4b; *Alpert et al.*, 2022 Fig. 5a | CNT (ABIFM) | Mineral dust, Organic and SSA | $N_{\mathrm{aer}}, A_{\mathrm{aer}}, T$, RH | - |

## 2.4 Sensitivity case studies

We investigate the sensitivity of $N_{\mathrm{INP}}$, $N_i$ and ice crystal formation rate ($dN_i/dt$) with respect to three immersion parameterizations and several key cloud parameters. A key distinction between the schemes, which governs their fundamentally different behaviors, is the realtionship of the INP reservoir ($N_{\mathrm{INP}}$) to the total aerosol population in its composition class ($N_{\mathrm{aer}}$). For instance, mineral dust, a canonical INP class, is highly efficient at nucleating ice but often represents a numerically negligible fraction of $N_{\mathrm{aer}}$ compared to more abundant types like organics or sea spray aerosol. In deterministic schemes, $N_{\mathrm{INP}}$ represents a limited subset of $N_{\mathrm{aer}}$ that are in a composition class based on the temperature at which they may be activated. In contrast, for the CNT-based schemes, all aerosol particles in an INP composition class are treated as potential INPs, making the INP reservoir effectively equivalent to the total aerosol population in that composition class, $N_{\mathrm{aer}}$. We vary the number concentration of aerosols in each composition class ($N_{\mathrm{aer}}$), cloud cooling rate (CCR), cloud-top entrainment rate ($w_e$) and ice crystal fall velocity ($v_f$) while applying three aerosol particle types including mineral dust, organic aerosol, and SSA PSDs.

We examine various simulation setups as given in Table 4. The thermodynamic conditions and cloud parameters from the LES baseline results of the SHEBA case study serve as the control run (hereafter referred to as CTRL) while applying the three different aerosol PSDs. We note that the CTRL setup differs slightly from the one used in our previous study (Knopf et al., 2023b). Detailed information on these modifications can be found in Appendix B.





**Table 4: Parameter choices of the different 1D aerosol-cloud model simulation. Imm_CTRL: The CTRL run with the baseline settings for all IMF parameterizations, no perturbations, used as a reference. h_res_t and h_res_z: Simulation applying higher resolution with doubly refined vertical resolution (5 m) and much smaller time step (1 s).**

| Simulations | Description |
|---|---|
| Control Run | |
| Imm_CTRL | With the original settings (see Table 2) |
| Higher Resolution Tests | |
| h_res_t | $\delta t$ = 1 s |
| h_res_z | $\delta z$ = 5 m |
| Sensitivity Tests | |
| $N_{aer}$ x 10 | Increasing / decreasing the $N_{aer}$ (number concentration of |
| $N_{aer}$ x 01 | aerosols) by a factor of 10 |
| CCR = 0.3 | Cloud cooling rate (CCR) = 0.3 ℃ h$^{-1}$ |
| $w_e$ = 1.0 | Entrainment rate ($w_e$) = 1.0 cm s$^{-1}$ |
| NO_ENTRAIN | Entrainment rate ($w_e$) = 0 cm s$^{-1}$ |
| $v_f$ = 1.0 | Ice crystal fall velocity ($v_f$) = 1.0 m s$^{-1}$ |
| External | External mixture of mineral dust, organic, and SSA particles |

To examine the sensitivity of cloud properties ($N_{INP}$, $N_i$ and $dN_i/dt$) towards the parameters $N_{aer}$, CCR, $w_e$ and $v_f$, a series of simulations for each IMF parameterization are carried out by repeating the CTRL case with adjusting the targeted parameters (see Table 4). A detailed summary of all simulations is included in Table 5.

**Table 5: Summary of 1D-AC model simulations.**

| Case | Name | IMF parameterization applied (INN, INAS and ABIFM, respectively) | Description of simulation setup | Number of simulated cases |
|---|---|---|---|---|
| 1 | $Imm_{MD}^{CTRL}$ | D2015, N2012, ABIFM_dust | Baseline settings for all IMF parameterizations, no perturbations, the CTRL run, used as a reference. | 4 (IMF parameterizations) x 3 (particle types): 12 |
| 2 | $Imm_{Org}^{CTRL}$ | D2010, C2017, ABIFM_organics | | |
| 3 | $Imm_{SSA}^{CTRL}$ | D2010, A2022, ABIFM_SSA | | |





| | | | | |
|---|---|---|---|---|
| 4 | $\mathrm{Imm}_{MD}^{N_{aer} \times 10}$ | Same as num. 1 | $N_{aer}$ is increased or decreased by a factor of 10 compared to the CTRL run. | 4 (IMF parameterizations) x 6 (particle types): 24 |
| 5 | $\mathrm{Imm}_{Org}^{N_{aer} \times 10}$ | Same as num. 2 | | |
| 6 | $\mathrm{Imm}_{SSA}^{N_{aer} \times 10}$ | Same as num. 3 | | |
| 7 | $\mathrm{Imm}_{MD}^{N_{aer} \times 01}$ | Same as num. 1 | | |
| 8 | $\mathrm{Imm}_{Org}^{N_{aer} \times 01}$ | Same as num. 2 | | |
| 9 | $\mathrm{Imm}_{SSA}^{N_{aer} \times 01}$ | Same as num. 3 | | |
| 10 | $\mathrm{Imm}_{MD}^{CCR = 0.3}$ | Same as num. 1 | The temperature profiles are modified to linearly decrease by 3 ℃ over the 10 h simulation. | 4 (IMF parameterizations) x 3 (particle types): 12 |
| 11 | $\mathrm{Imm}_{Org}^{CCR = 0.3}$ | Same as num. 2 | | |
| 12 | $\mathrm{Imm}_{SSA}^{CCR = 0.3}$ | Same as num. 3 | | |
| 13 | $\mathrm{Imm}_{MD}^{w_e = 1.0}$ | Same as num. 1 | $w_e$ is increased from 0.1 cm s$^{-1}$ to 1.0 cm s$^{-1}$ compared to the CTRL runs. | 4 (IMF parameterizations) x 3 (particle types): 12 |
| 14 | $\mathrm{Imm}_{Org}^{w_e = 1.0}$ | Same as num. 2 | | |
| 15 | $\mathrm{Imm}_{SSA}^{w_e = 1.0}$ | Same as num. 3 | | |
| 16 | $\mathrm{Imm}_{MD}^{w_e = 0}$ | Same as num. 1 | $w_e$ is switched off compared to the CTRL runs. | 4 (IMF parameterizations) x 1 (particle types): 4 |
| 17 | $\mathrm{Imm}_{MD}^{v_f = 1.0}$ | Same as num. 1 | $v_f$ is increased from 0.3 m s$^{-1}$ to 1.0 m s$^{-1}$ compared to the CTRL runs. | 4 (IMF parameterizations) x 3 (particle types): 12 |
| 18 | $\mathrm{Imm}_{Org}^{v_f = 1.0}$ | Same as num. 2 | | |
| 19 | $\mathrm{Imm}_{SSA}^{v_f = 1.0}$ | Same as num. 3 | | |
| 20 | Imm_External | Same as num. 1, 2, 3 | External mixture of mineral dust, organic, and SSA particles each assigned a unique, observationally-based particle size distribution | 3 (IMF parameterizations): 3 |

We multiply the $N_{aer}$ by a factor of 10 and $10^{-1}$ for CTRL termed experiments $N_{aer}$ x 10 and $N_{aer}$ x 01, respectively. A total of 36 cases are obtained, exploring three different aerosol PSDs (mineral dust, organic and SSA particles), four different IMF

freezing parameterizations (INN, INAS, ABIFM and ABIFM*) and three sets of experiments (CTRL, $N_{aer}$ x 10, $N_{aer}$ x 01) (see Table 5).

In order to assess the sensitivity of cloud properties to CCR changes, the temperature profiles for CTRL are modified by applying a cooling rate of 0.3 ℃ h$^{-1}$ to each layer (see Figure S2). This means that after 10 hours, the entire layer is 3 ℃ cooler than at the start of the simulation, termed experiment CCR = 0.3 (see Table 4). This results in 24 cases consisting of three

aerosol PSDs, four freezing parameterizations and two sets of cases (CTRL for deterministic IMF parameterizations and CNT-based IMF parameterizations and CCR = 0.3) (see Table 5). The evolution of the temperature profiles for CTRL, and CCR = 0.3 are presented in Fig. S2.





Additionally, we investigate the responses of the INPs and $N_i$ evolution to changes in $w_e$ by repeating the CTRL cases while increasing $w_e$ from 0.1 to 1.0 cm s$^{-1}$, termed experiments $w_e = 1.0$ which contain 24 cases including three aerosol PSDs, four

IMF freezing parameterizations and two sets of experiments (CTRL, $w_e = 1.0$) (see Table 5).

Lastly, the response of INPs and $N_i$ to changes in $v_f$ is evaluated by changing the $v_f$ from 0.3 to 1.0 m s$^{-1}$, termed experiments $v_f = 1.0$ containing 24 cases in total, consisting of three aerosol PSDs, four IMF parameterizations, and two sets of experiments (CTRL, $v_f = 1.0$) (see Table 5).

**2.5 Prognostic Evaluation of IMF Parameterizations in the Presence of Different Aerosol PSDs**

To move beyond idealized, single aerosol type studies and demonstrate the model's capability to simulate aerosol populations more representative of the Arctic atmosphere, we conduct simulations representing an external mixture of mineral dust, organic, and SSA particles each assigned a unique, observationally-based particle size distribution (Table 1) (Riemer et al., 2019). We term this simulation as "External". This approach ensures that particles from different sources are treated as physically separate entities while coexisting in the same simulation. The primary goal is to prognostically evaluate how

competing aerosol types and associated IMF parameterizations impact $N_{INP}$ and $N_i$, with freezing initiated by either INN, INAS, or ABIFM as described in Appendix A.

**3 Results**

**3.1 Influence of Immersion Freezing Parameterization on INP and Ice Crystal Evolution**

The choice of IMF parameterization fundamentally dictates the simulated temporal evolution of $N_{INP}$, $N_i$, $dN_i/dt$. Figure 3

illustrates the 10-hour domain-averaged time series of $N_{INP}$ for control (CTRL) simulations under different IMF schemes and aerosol types, with Figure S3, S4, S5 providing a detailed view of the initial 0.1 hours. The significant differences in predicted INP concentrations when applying different IMF parameterizations to identical particle size distributions are further illustrated in Figure S6, which demonstrates how parameterization choice can lead to orders-of-magnitude variations in INP predictions. For deterministic IMF schemes (INN, INAS), where INP activation is treated as instantaneous once temperature criteria are

met, a rapid decrease in $N_{INP}$ is observed, typically by over 90% from initial values within the first hour (Fig. 3, Table 6, and Fig. S3).






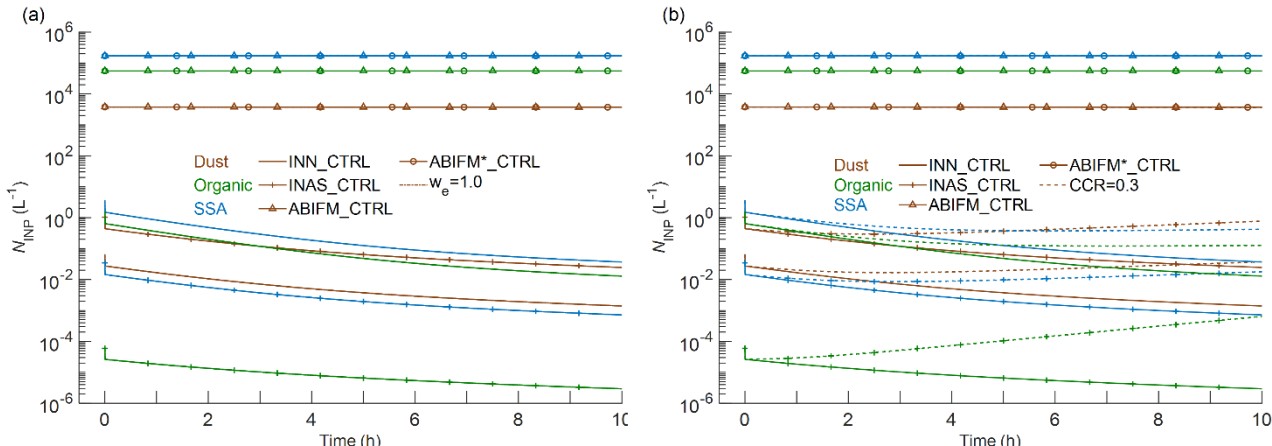

**Figure 3. Time series of simulated domain-averaged activatable INP number concentration ($N_{INP}$ in L$^{-1}$) when changing the cloud-top entrainment rate (a) and cloud cooling rate (b). Simulations are initialized with different aerosol PSDs (dust, organic, and SSA particles), immersion freezing parameterizations (INN, INAS, ABIFM, ABIFM*) and cloud parameters (cloud cooling rate, cloud-top entrainment rate). Brown, green, and blue lines represent the application of aerosol PSDs of dust, organic, and SSA particles, respectively. Simulation results represent different immersion freezing parameterizations: INN (no symbols), INAS (cross), ABIFM (triangle), and ABIFM* (circle). In both panels, the thin solid lines indicate results with the original, unperturbed cloud parameters (CTRL). The dashed lines denote results with the cloud cooling rate (CCR) of 0.3 °C h$^{-1}$ (CCR = 0.3) and the dash-dotted lines show the results with the cloud-top entrainment rate ($w_e$) of 1 cm s$^{-1}$ ($w_e$ = 1.0).**

**Table 6: The temporal evolution of the domain-averaged number concentration of activatable ice-nucleating particles ($N_{INP}$ in L$^{-1}$) for given aerosol PSDs (dust, organics and SSA described in Table 1) normalized by the initially activatable INP number concentration, that is $N_{INP}/N_{INP}^{t=0}$, using deterministic ice nucleation number based (INN), the deterministic ice nucleation active sites (INAS), the classical nucleation theory (CNT) water activity based immersion freezing model (ABIFM) and ABIFM enabling subsaturated freezing (ABIFM*). $N_{INP}$ is given for model simulations described in Table 4 for model times 0.5, 5, and 10 h.**

|  | Dust | | | Organic | | | SSA | | |
|---|---|---|---|---|---|---|---|---|---|
|  | 0.5h | 5h | 10h | 0.5h | 5h | 10h | 0.5h | 5h | 10h |
| INN_CTRL | 0.32 | 0.06 | 0.02 | 0.30 | 0.03 | 0.01 | 0.30 | 0.03 | 0.01 |
| INAS_CTRL | 0.33 | 0.06 | 0.02 | 0.37 | 0.11 | 0.05 | 0.32 | 0.06 | 0.02 |
| ABIFM_CTRL | 1.00 | 0.99 | 0.98 | 1.00 | 1.00 | 1.00 | 1.00 | 1.00 | 1.00 |
| ABIFM*_CTRL | 1.00 | 0.99 | 0.98 | 1.00 | 1.00 | 1.00 | 1.00 | 1.00 | 1.00 |
| INN_CCR=0.3 | 0.36 | 0.30 | 0.57 | 0.31 | 0.08 | 0.08 | 0.32 | 0.10 | 0.11 |
| INAS_CCR=0.3 | 0.36 | 0.35 | 0.73 | 0.46 | 1.77 | 10.66 | 0.35 | 0.28 | 0.51 |
| ABIFM_CCR=0.3 | 1.00 | 0.99 | 0.97 | 1.00 | 1.00 | 1.00 | 1.00 | 1.00 | 1.00 |
| ABIFM*_CCR=0.3 | 1.00 | 0.98 | 0.97 | 1.00 | 1.00 | 1.00 | 1.00 | 1.00 | 1.00 |
| INN_$w_e$=1.0 | 0.32 | 0.06 | 0.02 | 0.30 | 0.03 | 0.01 | 0.30 | 0.03 | 0.01 |
| INAS_$w_e$=1.0 | 0.33 | 0.06 | 0.02 | 0.37 | 0.11 | 0.05 | 0.32 | 0.06 | 0.02 |
| ABIFM_$w_e$=1.0 | 1.00 | 0.99 | 0.98 | 1.00 | 1.00 | 1.00 | 1.00 | 1.00 | 1.00 |
| ABIFM*_$w_e$=1.0 | 1.00 | 0.99 | 0.98 | 1.00 | 1.00 | 1.00 | 1.00 | 1.00 | 1.00 |



This rapid consumption of the initial, limited INP reservoir in deterministic schemes directly impacts ice formation, as shown by the time series of domain-averaged $N_i$ in Figure 4. An initial sharp peak in $N_i$ occurs around 0.2 hours, followed by a substantial decrease. For the CTRL cases, $N_i$ decreases by 86-98% from this peak value after 10 hours.

**Figure 4: Temporal evolution of the domain-averaged ice crystal number concentration ($N_i$ in L$^{-1}$) in response to different cloud system parameters. The nine panels are organized by aerosol type in rows (mineral dust, top; organic, middle; and sea spray aerosol (SSA), bottom) and by sensitivity experiment in columns. The columns from left to right represent simulations with an applied cloud cooling rate (CCR), an enhanced entrainment rate ($w_e$), and an increased ice crystal fall speed ($v_f$), respectively. Within each panel, different line styles and colors represent the four immersion freezing (IMF) parameterizations, with legends and styling identical to those used in Figure 3.**




The $dN_i/dt$ follows a similar trajectory, as depicted in Figure 5. After an initial burst, the formation rate for deterministic

schemes decreases by 3-4 orders of magnitude over the 10-hour period, demonstrating the rapid decline in ice production rates

following depletion of the initial INP reservoir.

**Figure 5: Temporal evolution of the domain-averaged ice crystal formation rate ($dN_i/dt$ in L$^{-1}$ h$^{-1}$). The six panels are organized by aerosol type in rows (mineral dust, top; organic, middle; and sea spray aerosol (SSA), bottom) and by sensitivity experiment in columns. The columns from left to right show the response to an applied cloud cooling rate (CCR), and enhanced entrainment rate ($w_e$). Legends and line styles for the four immersion freezing (IMF) parameterizations are identical to those used in Figure 4.**





To provide a more quantitative summary of these trends, the evolution of $N_i$ is normalized and presented in Table 7. Unlike $N_{INP}$, which has a defined value at time zero, $N_i$ only begins to form after the first time step when nucleation processes are activated. Therefore, Table 7 normalizes $N_i$ by its value at the first model output time (t=10 s), providing detailed values that confirm the dramatic decrease observed in the figures.

**Table 7: The domain-averaged number concentration of ice crystals ($N_i$ in L⁻¹) normalized by the initial ice crystal number concentraton at the first model output time, that is $N_i/N_i^{t=10\,s}$ following Table 6.**

| | Dust | | | Organic | | | SSA | | |
|---|---|---|---|---|---|---|---|---|---|
| | 0.5h | 5h | 10h | 0.5h | 5h | 10h | 0.5h | 5h | 10h |
| INN_CTRL | 0.44 | 0.05 | 0.02 | 0.42 | 0.03 | 0.01 | 0.43 | 0.04 | 0.01 |
| INAS_CTRL | 0.45 | 0.05 | 0.02 | 0.50 | 0.09 | 0.04 | 0.45 | 0.05 | 0.02 |
| ABIFM_CTRL | 98.79 | 71.32 | 59.39 | 114.90 | 118.56 | 118.56 | 112.84 | 116.27 | 116.25 |
| ABIFM*_CTRL | 97.11 | 68.00 | 56.21 | 114.70 | 118.34 | 118.34 | 111.05 | 114.34 | 114.32 |
| INN_CCR=0.3 | 0.50 | 0.28 | 0.53 | 0.44 | 0.08 | 0.08 | 0.45 | 0.10 | 0.11 |
| INAS_CCR=0.3 | 0.52 | 0.34 | 0.71 | 0.69 | 1.98 | 12.10 | 0.51 | 0.27 | 0.49 |
| ABIFM_CCR=0.3 | 102.68 | 117.95 | 159.11 | 129.92 | 709.88 | 4422.28 | 118.52 | 237.06 | 490.69 |
| ABIFM*_CCR=0.3 | 100.98 | 112.55 | 150.87 | 129.70 | 708.67 | 4415.27 | 116.67 | 233.29 | 483.03 |
| INN_$w_e$=1.0 | 0.55 | 0.16 | 0.13 | 0.51 | 0.13 | 0.11 | 0.52 | 0.13 | 0.11 |
| INAS_$w_e$=1.0 | 0.56 | 0.16 | 0.13 | 0.64 | 0.23 | 0.18 | 0.55 | 0.15 | 0.13 |
| ABIFM_$w_e$=1.0 | 99.27 | 74.94 | 65.36 | 114.90 | 118.56 | 118.56 | 112.84 | 116.28 | 116.26 |
| ABIFM*_$w_e$=1.0 | 97.55 | 71.54 | 62.01 | 114.70 | 118.34 | 118.34 | 111.05 | 114.35 | 114.33 |
| INN_$v_f$=1.0 | 0.10 | 0.01 | 0.01 | 0.09 | 0.01 | 0.00 | 0.09 | 0.01 | 0.00 |
| INAS_$v_f$=1.0 | 0.10 | 0.01 | 0.01 | 0.12 | 0.02 | 0.01 | 0.10 | 0.01 | 0.01 |
| ABIFM_$v_f$=1.0 | 28.08 | 20.24 | 16.90 | 34.07 | 34.07 | 34.06 | 33.24 | 33.23 | 33.23 |
| ABIFM*_$v_f$=1.0 | 27.45 | 19.20 | 15.92 | 33.99 | 33.99 | 33.99 | 32.57 | 32.56 | 32.56 |

Correspondingly, since ice crystal formation rates are only meaningful after nucleation begins, Table 8 normalizes the $dN_i/dt$ values using the initial rate at t=10 s as the reference. This allows for a clear, quantitative assessment of the rapid decline in
ice production rates over the simulation period.






**Table 8: The domain-averaged ice crystal formation rate ($dN_i/dt$ in $L^{-1}$ $h^{-1}$) normalized by the initial ice crystal number concentraton at the first model output time, that is $dN_i/dt$ / $dN_i/dt^{t=10\,s}$ following Table 6.**

| | Dust | | | Organic | | | SSA | | |
|---|---|---|---|---|---|---|---|---|---|
| | 0.5h | 5h | 10h | 0.5h | 5h | 10h | 0.5h | 5h | 10h |
| INN_CTRL | 0.00 | 0.00 | 0.00 | 0.00 | 0.00 | 0.00 | 0.00 | 0.00 | 0.00 |
| INAS_CTRL | 0.00 | 0.00 | 0.00 | 0.00 | 0.00 | 0.00 | 0.00 | 0.00 | 0.00 |
| ABIFM_CTRL | 0.84 | 0.61 | 0.51 | 1.00 | 1.00 | 1.00 | 1.00 | 1.00 | 1.00 |
| ABIFM*_CTRL | 0.85 | 0.59 | 0.49 | 1.00 | 1.00 | 1.00 | 1.00 | 1.00 | 1.00 |
| INN_CCR=0.3 | 0.00 | 0.00 | 0.00 | 0.00 | 0.00 | 0.00 | 0.00 | 0.00 | 0.00 |
| INAS_CCR=0.3 | 0.00 | 0.00 | 0.01 | 0.00 | 0.01 | 0.09 | 0.03 | 0.02 | 0.03 |
| ABIFM_CCR=0.3 | 0.89 | 1.03 | 1.39 | 1.20 | 6.40 | 39.91 | 1.08 | 2.09 | 4.33 |
| ABIFM*_CCR=0.3 | 0.90 | 1.00 | 1.35 | 1.20 | 6.41 | 39.92 | 1.08 | 2.09 | 4.34 |
| INN_$w_e$=1.0 | 0.00 | 0.00 | 0.00 | 0.00 | 0.00 | 0.00 | 0.00 | 0.00 | 0.00 |
| INAS_$w_e$=1.0 | 0.00 | 0.00 | 0.00 | 0.00 | 0.00 | 0.00 | 0.00 | 0.00 | 0.00 |
| ABIFM_$w_e$=1.0 | 0.85 | 0.64 | 0.56 | 1.00 | 1.00 | 1.00 | 1.00 | 1.00 | 1.00 |
| ABIFM*_$w_e$=1.0 | 0.85 | 0.62 | 0.54 | 1.00 | 1.00 | 1.00 | 1.00 | 1.00 | 1.00 |

In contrast, simulations employing CNT-based IMF parameterizations (ABIFM, ABIFM*), where all aerosol particles are potential INPs and activate via a continuous, time dependent freezing process, exhibits a substantially larger and more stable $N_{INP}$ reservoir. $N_{INP}$ remain within ~2% of its initial value at t=0 throughout the 10-hour simulation (Fig. 3a). Table 6 presents these values normalized by their initial value at t=0. Consequently, $N_i$ in CNT-based simulations reach a quasi-stable plateau after an initial increase and remain orders of magnitude higher compared to the case of deterministic schemes (Fig. 4 and Table

7). Similarly, $dN_i/dt$ for CNT schemes is maintained at significantly higher levels (Fig. 5 and Table 8). Simulations with ABIFM* produce domain-averaged $N_{INP}$ and $N_i$ trends broadly similar to the ABIFM cases (Fig. 3 and 4).

**3.2 Impact of Aerosol Type and Number Size Distribution**

While the choice of parameterization establishes the overarching framework for INP and ice evolution, the specific type and size distribution of aerosol particles further modulate ice nucleation efficiency within these frameworks. The impact of aerosol

identity (mineral dust, organic, SSA) on $N_{INP}$, $N_i$, and $dN_i/dt$ varies significantly depending on whether INP activation is parameterized based on INN, INAS, or ABIFM.

For the INAS parameterization, which ties ice nucleation to the ice-active site density ($n_s$) of particles (Eq. A4-A7), aerosol composition is paramount. Despite mineral dust having the lowest total aerosol number concentration ($N_{aer}$) among the considered types (Table 1), it consistently yields the highest $N_{INP}$ and subsequently $N_i$ and $dN_i/dt$ in CTRL simulations (see

Figs. 3, 4, 5, green lines vs. black/blue for INAS crosses). This is directly attributable to the significantly higher $n_s$ values prescribed for mineral dust compared to organic and SSA particles at the relevant cloud temperatures (e.g., Eq. A5 vs. Eqs. A6 and A7). Conversely, for the INN parameterization, which primarily depends on the number concentration of aerosol





particles larger than 0.5 µm ($N_\text{aer} > 0.5$ µm, Eqs. A1-A2), the aerosol type itself (beyond its contribution to $N_\text{aer} > 0.5$ µm) plays a lesser role. Consequently, SSA, with the highest $N_\text{aer} > 0.5$ µm in our PSDs (Table 1), results in the largest initial $N_\text{INP}$

for INN.

Under the CNT-based ABIFM scheme, where all aerosol particles are potential INPs ($N_\text{INP} = N_\text{aer}$), the initial $N_\text{INP}$ primarily reflects the total aerosol loading of each type. However, the subsequent ice formation rates are governed by the heterogeneous ice nucleation rate coefficient ($J_\text{het}$), which is aerosol-specific (Eqs. A10-A12). Mineral dust, with its generally higher $J_\text{het}$ values, tends to produce more ice crystals and at a faster rate compared to organic or SSA particles for a given $N_\text{aer}$ and

thermodynamic condition (Figs. 4 and 5, ABIFM triangles). Thus, also when applying a time-dependent freezing process, aerosol identity remains a key factor in determining the intensity of ice production. The differing  PSDs associated with each aerosol type (Fig. 2 and Table 1) inherently influence these outcomes by determining the total number and surface area of particles available for nucleation, regardless of the parameterization.

Beyond these inherent differences between aerosol types, we test the system's sensitivity by scaling the total aerosol loading

by factors of 10 and 0.1. The results (see Fig. S7) show a straightforward, linear response: the concentrations of $N_\text{INP}$ and $N_\text{i}$, as well as $dN_\text{i}/dt$, scale proportionally with the initial aerosol loading for all parameterizations.

### 3.3 Sensitivity to Cloud System Parameters

Beyond the intrinsic properties of aerosols and the choice of freezing parameterization, the dynamic and thermodynamic environment of the cloud system—specifically CCR, $w_\text{e}$, and $v_\text{f}$—exerts significant control over the evolution of INPs and ice

crystals.

### 3.3.1 Cloud Cooling Rate (CCR): A Powerful Driver of Ice Production Intensity

Continuous radiative cooling concentrated at cloud top supports turbulent mixing and leads to progressively declining temperatures throughout the cloud domain, which can drive further INP activation by lowering cloud temperatures (Morrison et al., 2011). The impact of an applied CCR of 0.3 °C h⁻¹ is most pronounced for deterministic parameterizations (Figs. 3b, 4,

and 5). For these parameterizations, cooling increases the number concentration of activatable INP by lowering the ambient temperature to meet the fixed activation thresholds of a progressively larger and colder subset of the total INP population. (Eqs. A1, A4 and Fig. S2). For example, with INAS, the $N_\text{INP}$ initially declines due to activation as shown in Fig. 3b (lines with cross signs). However, under continuous cooling (Fig. 3b, dashed lines with cross signs), it can recover and even exceed its initial value after several hours, especially for organic aerosols which exhibits high sensitivity of $N_\text{INP}$ to temperature

changes. Table 6 shows that the organic INAS_CCR=0.3 case reaches over 10 times of initial $N_\text{INP}$ after 10h. This translates to a large increase in $N_\text{i}$ (up to a factor of ~300 for organic INAS, see Table 7) and sustains, or even increases, $dN_\text{i}/dt$ compared to CTRL runs where $N_\text{i}$ rapidly depletes (Figs. 4 and 5).





For CNT-based approaches, the impact of CCR on the already vast $N_{\mathrm{INP}}$ reservoir is minimal (see Table 6, $N_{\mathrm{INP}}$ changes by < 3% for ABIFM). However, lower temperatures significantly enhance $J_{\mathrm{het}}$, leading to substantially increased $N_{\mathrm{i}}$ (e.g., $N_{\mathrm{i}}$

increases by factors of ~3-4 for ABIFM with CCR, Table 7) and $dN_{\mathrm{i}}/dt$ (Figs. 4 and 5). This highlights that while the potential INP reservoir in CNT schemes is less sensitive to cooling-induced expansion, the actual rate of ice formation remains highly sensitive to temperature.

### 3.3.2 Cloud-Top Entrainment Rate ($w_{\mathrm{e}}$): A Critical INP Source for Depletion-Prone Deterministic Schemes

Cloud-top entrainment of free-tropospheric air provides a mechanism for replenishing INPs. In our simulations with

instantaneous activation for deterministic schemes, entrained INPs are immediately converted to ice crystals. Therefore, while the standing $N_{\mathrm{INP}}$ concentration within the cloud layer does not show a sustained increase due to entrainment (Fig. 3a, deterministic cases, $w_{\mathrm{e}} = 1.0$ cm s⁻¹ vs. CTRL), this process acts as a continuous source flux ($S_{\mathrm{ent}}$, Fig. 10) sustaining ice production. This is evident in the significantly higher $N_{\mathrm{i}}$ and $dN_{\mathrm{i}}/dt$ observed in deterministic scheme simulations with enhanced entrainment ($w_{\mathrm{e}} = 1.0$ cm s⁻¹) compared to CTRL runs, particularly after the initial in-cloud INP reservoir is depleted

(Figs. 4 and 5). For instance, $N_{\mathrm{i}}$ for INAS with $w_{\mathrm{e}} = 1.0$ cm s⁻¹ can be 3-7 times higher than CTRL after 10 hours (see Table 7, comparing $w_{\mathrm{e}} = 1.0$ to CTRL for INAS).

For CNT-based approaches, entrainment directly adds to the total aerosol particle population ($N_{\mathrm{aer}}$), which constitutes the $N_{\mathrm{INP}}$ reservoir. However, minor effects of entrainment can be found. The impact on $N_{\mathrm{i}}$ and $dN_{\mathrm{i}}/dt$, while positive, is less significant in relative terms compared to deterministic schemes (e.g., $N_{\mathrm{i}}$ increases by ~2-3% for ABIFM with $w_{\mathrm{e}} = 1.0$ cm s⁻¹

after 10 h, Table 7), as the initial reservoir is already substantial and not the primary limiting factor for ice production.

### 3.3.3 Ice Crystal Fall Speed ($v_{\mathrm{f}}$): The Primary Sink for Ice Crystals

Ice crystal sedimentation is the sole loss mechanism for ice crystals in our 1D-AC model. Increasing the number-weighted ice fall speed ($v_{\mathrm{f}}$) from 0.3 m s⁻¹ (CTRL) to 1.0 m s⁻¹ leads to a more rapid removal of ice crystals from the cloud layer. This results in substantially lower $N_{\mathrm{i}}$ across all parameterizations and aerosol types (Fig. 5). For deterministic schemes, the initial

peak in $N_{\mathrm{i}}$ is sharper and the subsequent decline more pronounced, with $N_{\mathrm{i}}$ being roughly an order of magnitude lower after 10 hours with $v_{\mathrm{f}} = 1.0$ m s⁻¹ compared to CTRL (Table 7). Similarly, for CNT-based approaches, higher $v_{\mathrm{f}}$ leads to an approximate 70% reduction in $N_{\mathrm{i}}$ after 10 hours (Table 7). The efficiency of sedimentation thus plays a crucial role in modulating the standing $N_{\mathrm{i}}$ and the lifetime of ice within the mixed-phase cloud.

### 3.4 Vertical Profiles Reveal Contrasting Sensitivities

While domain-averaged properties provide a valuable overview of the cloud system's response, they can obscure critical height-dependent processes that govern cloud evolution. To deconstruct these mechanisms, we now analyze the vertical profiles of the time-averaged change in INP number concentration ($\Delta N_{\mathrm{INP}}(z)$), ice crystal number concentration ($\Delta N_{\mathrm{i}}(z)$), and



ice crystal formation rate ($\Delta dN_i/dt(z)$) in response to changes in CCR, $w_e$, and $v_f$. This approach allows us to pinpoint where in the cloud these forcings have their greatest impact and how that impact differs between IMF parameterizations.

### 3.4.1 INP Concentration Response ($\Delta N_{\mathrm{INP}}(z)$)

Figure 6 reveals that the INP reservoir responds in fundamentally different ways to cloud cooling and cloud-top entrainment. For the deterministic INN and INAS schemes, CCR acts as a powerful volume-wide forcing, increasing $N_{\mathrm{INP}}$ to values up to ~ 4.6 and ~ 21.4 times the original, respectively, primarily in the upper half of the cloud (Fig. 6a, b). In contrast, the impact of entrainment is sharply localized to the cloud top, providing a boundary-driven source that boosts $N_{\mathrm{INP}}$ by up to more than four times its original value. The CNT-based ABIFM scheme shows a different behavior: its large INP reservoir is virtually insensitive to both cooling and entrainment, with $\Delta N_{\mathrm{INP}}$ remaining less than 1% throughout the vertical column (Fig. 6c, d).

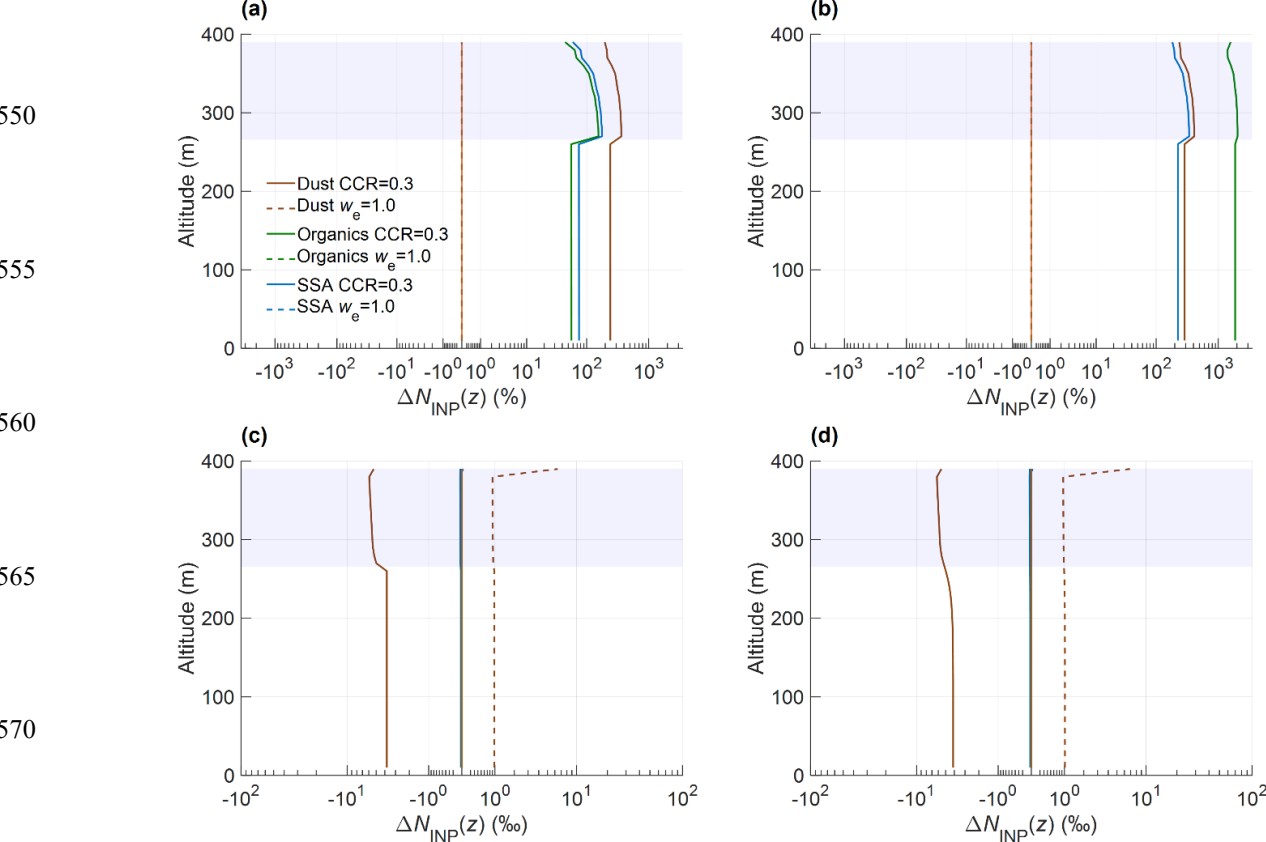

**Figure 6: Vertical profiles of the change in number concentration of activatable INP ($\Delta N_{\mathrm{INP}}(z)$ in % or ‰) averaged over entire 10 h of simulation time. $\Delta N_{\mathrm{INP}}$ differs compared to the respective CTRL runs due to the change of cloud parameters (cloud cooling rate, cloud-top entrainment rate) applying dust, organic and SSA particles, given as black, green, and blue lines, respectively. Different immersion freezing parameterizations are applied including (a) ice nucleation number based (INN), (b) ice-nucleation active sites (INAS), (c) water-activity based immersion freezing model (ABIFM), and (d) ABIFM enabling subsaturated freezing (d) ABIFM\*. Simulation results for changing cloud cooling rate (solid lines) and cloud-top entrainment rate (dashed lines) are shown. The blue shaded area denotes the cloud layer and the red line in the middle highlights the value of 0.**



### 3.4.2 Ice Crystal Concentration Response ($\Delta N_i(z)$)

The response of the ice crystal concentration ($\Delta N_i(z)$), shown in Figure 7, directly reflects these differing sensitivities. For the INAS scheme, CCR is the dominant factor, driving an enormous $\Delta N_i(z)$ of up to 20-fold (Fig. 7b). The ABIFM scheme is similarly dominated by cooling, which yields a $\Delta N_i(z)$ of up to tenfold (Fig. 7c). Conversely, the INN scheme is more sensitive

to entrainment, which causes a nearly five-fold increase in $N_i$ at the cloud top (Fig. 7a). Increasing the $v_f$ acts as a powerful, universal sink. This leads to a substantial negative change in $N_i$ across all parameterizations. As illustrated in Fig. 7 dotted lines, increasing the fall velocity reduces the ice concentration by over 90% throughout the domain.

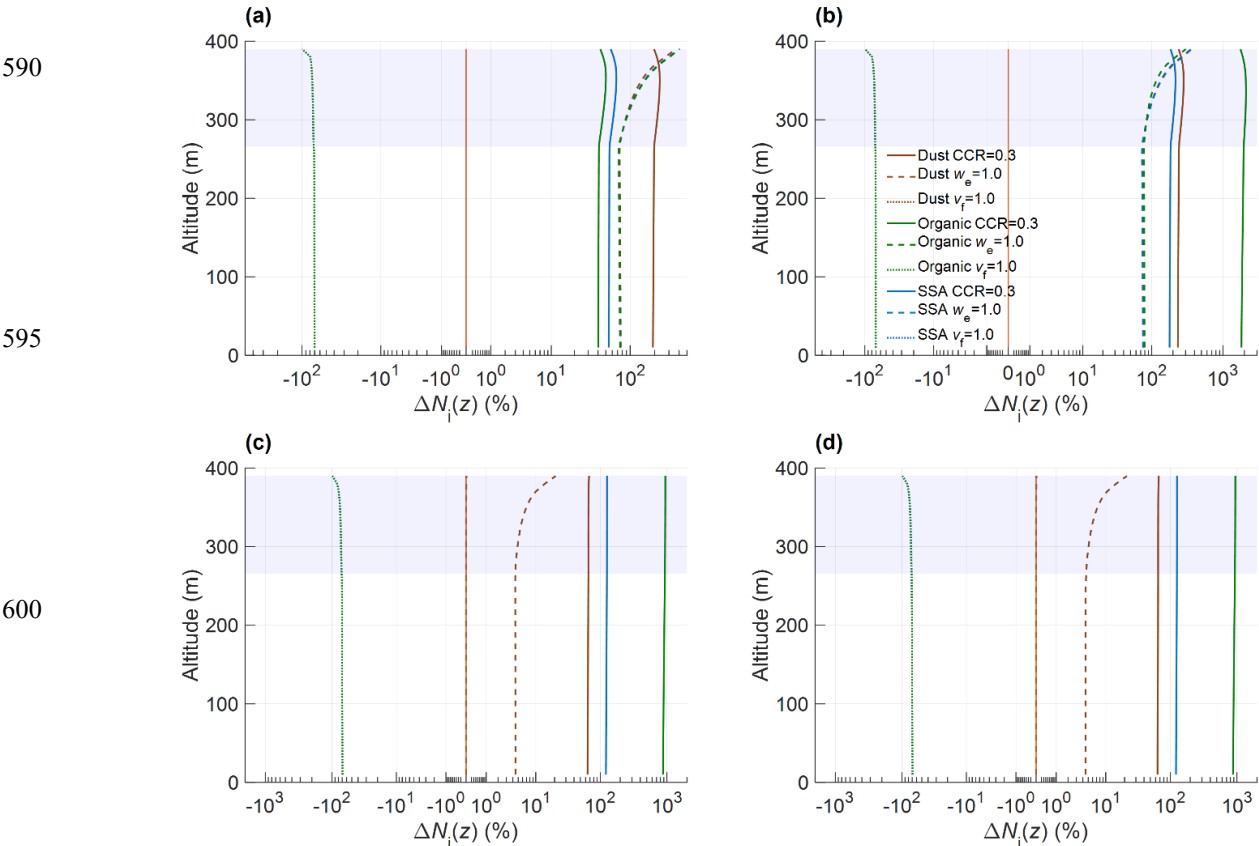

**Figure 7: As in Figure 6 but for the change in number concentration of ice crystals ($\Delta N_i(z)$ in %).**

### 3.4.3 Ice Crystal Formation Rate Response ($\Delta dN_i/dt(z)$)

The change in the ice crystal formation rate ($\Delta dN_i/dt(z)$) in Fig. 8 confirms these findings. Cloud cooling enhances the formation rate throughout the entire cloud layer for all parameterizations, with the most dramatic impact seen in the INAS





scheme, where the formation rate shows a peak increase over 25-fold near the cloud top (Fig. 8b). The effect of enhancing cloud-top entrainment is, again, almost exclusively a cloud-top phenomenon. It provides a significant boost to the formation rate for the deterministic schemes (increasing it more than fivefold for INN) but has a minor effect on the CNT-based scheme.

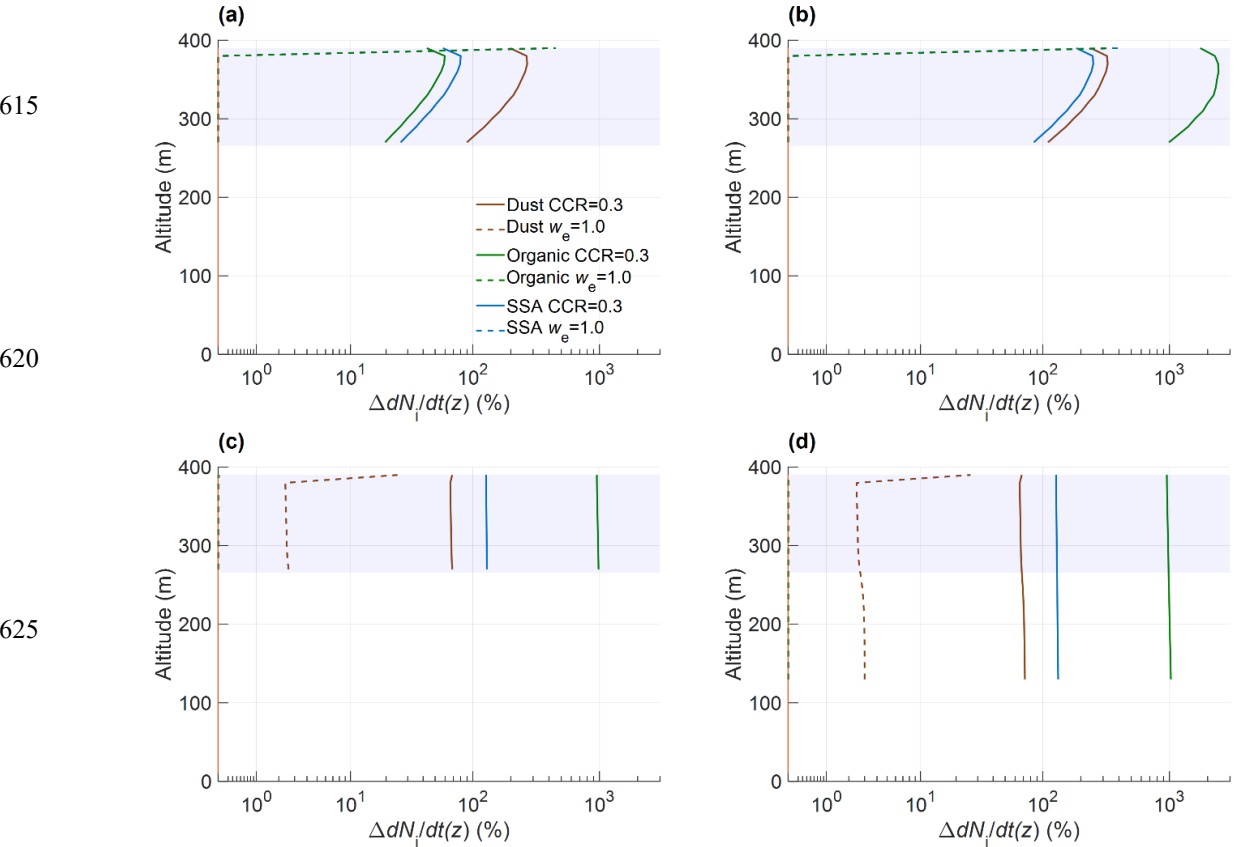

**Figure 8: As in Figure 6 but for the change in ice crystal formation rate ($\Delta dN_i/dt(z)$).**

## 3.5 Competing Aerosol in Cloud Ice Formation

The 1D-AC model is configured to simulate a more realistic scenario where mineral dust, organic, and SSA populations coexist, each with their distinct PSDs and INP parameterizations (External). This approach allows for an examination of the model's
ability to handle aerosol complexity and, more importantly, to investigate the relative contributions and potential dominance of different aerosol types (Fig. 9). The results show that the dominant source of ice can be governed by a combination of aerosol type and chosen freezing parameterization.







**Figure 9: Initial total aerosol population, activatable INP composition distribution, and resulting ice crystal number contributions ($N_i$) using different immersion freezing parameterizations. Panel (a) shows the external aerosol composition. Simulations were run with an aerosol population consisting of dust (brown), organic (green), and SSA (blue) particles. Panels b-d show the initial (t=0) fractional composition of the activatable INP number concentration for each parameterization. Panels e-g show the resulting domain-averaged $N_i$ over 10 hours and its contribution from each aerosol type. Each column represents a simulation applying INN_External (panels b and e), INAS_External (panels c and f), and ABIFM_External (panels d and g).**



Figure 9a shows the relative contribution of compositionally segregated subpopulations (mineral dust, organic, and sea spray aerosol particles) to the total aerosol number concentration. The aerosol population for these simulations is numerically

dominated by organic and SSA particles, which together comprise over 96% of the total particle number concentration. Mineral dust, in contrast, is a minor component by number.

The simulation INN_External identifies 5.32 L$^{-1}$ activatable INPs from particles larger than 0.5 μm, with SSA and organic particles comprising >98% of the initial activatable INPs (Fig. 9b). In this case, the INP concentration, determined solely by temperature and aerosol number concentration, is controlled by the most abundant particles throughout the simulation. SSA

serves as the primary ice source (Fig. 9e), with the combined ice concentration decreasing by approximately two orders of magnitude over 10 hours, from ~1 to 10$^{-2}$ L$^{-1}$.

In contrast, the INAS_External produces an initial activatable INP concentration of only 1.08 L$^{-1}$, composed almost exclusively (>96%) of mineral dust particles (Fig. 9c). Despite dust representing < 2% of the total aerosol population, its $n_s$ exceeds that of organic aerosols and SSA by several orders of magnitude, thereby dominating ice formation. Accordingly, dust particles

control the subsequent ice crystal formation (Fig. 9f). The ice crystal population is composed almost entirely of crystals formed on mineral dust, with negligible contribution from the more abundant but less efficient organic and SSA particles.

The ABIFM_External yields the highest activatable INP concentration (2.31 × 10$^5$ L$^{-1}$), equal to the total aerosol population (Figs. 9a and 9d). Ice production under ABIFM exhibits markedly different dynamics. Despite starting with the fewest activatable INPs, dust ultimately emerges as the dominant source of ice crystals during the 10-hour simulation, followed by

SSA and then organic particles (Fig. 9g). Ice crystal number concentration reaching ~2.3 L$^{-1}$ after 10 hours. This is two to three orders of magnitude higher than the final concentrations in the deterministic frameworks .

## 3.6 Dominant Processes in the INP Budget

To quantitatively diagnose the controlling factors of the simulated INP budget, a comparative analysis is performed for the deterministic (INAS) and CNT-based (ABIFM) parameterizations, specifically for the CTRL simulations initialized with the

mineral dust aerosol PSD. All budget terms are expressed as domain-averaged rates. The time-averaged budget balances (Fig. 10a, c) provide a concise summary of the relative magnitudes of INP sources (entrainment) and sinks (activation) over the 10-hour simulation, while the corresponding temporal evolution plots (Fig. 10b, d) reveal the dynamic mechanisms responsible for these balances.

For the deterministic (INAS) parameterization, the time-averaged domain-averaged budget (Fig. 10a) shows a net sink, with

the activation term ($S_{act} \approx$ -0.06 L$^{-1}$ h$^{-1}$) being an order of magnitude larger than the entrainment source ($S_{ent} \approx$ 0.005 L$^{-1}$ h$^{-1}$). The temporal evolution of $S_{ent}$ and $S_{act}$ (Fig. 3b) reveals the mechanism behind this imbalance: following an initial rapid activation phase that depletes available INPs within the first hour, the system transitions to an entrainment-limited regime. By approximately 2 hours, the entrainment source and activation sink converge to nearly equal magnitudes (~0.03 L$^{-1}$ h$^{-1}$), establishing a quasi-steady state where ice formation is controlled by the rate at which new INPs are supplied through

entrainment.




In stark contrast, the CNT (ABIFM) parameterization exhibits fundamentally different dynamics. The time-averaged budget terms are two orders of magnitude larger, with a massive activation sink ($S_{act} \approx$ -7.4 L$^{-1}$ h$^{-1}$) continuously overwhelming the substantial entrainment source ($S_{ent} \approx$ 0.5 L$^{-1}$ h$^{-1}$) (Fig. 10c). The temporal evolution (Fig. 10d) demonstrates that this imbalance is established instantaneously and persists throughout the simulation. Unlike the deterministic case, the activation

rate never becomes entrainment-limited but instead maintains relatively high values (~7.7 L$^{-1}$ h$^{-1}$), reflecting the continuous availability of potential INPs from the large aerosol reservoir characteristic of the stochastic freezing approach. If, however, this reservoir were to be significantly depleted (e.g., in an aerosol-poor environment or over much longer timescales), we would expect the activation rate to eventually become source-limited by entrainment, converging in principle with the behavior of the deterministic case.


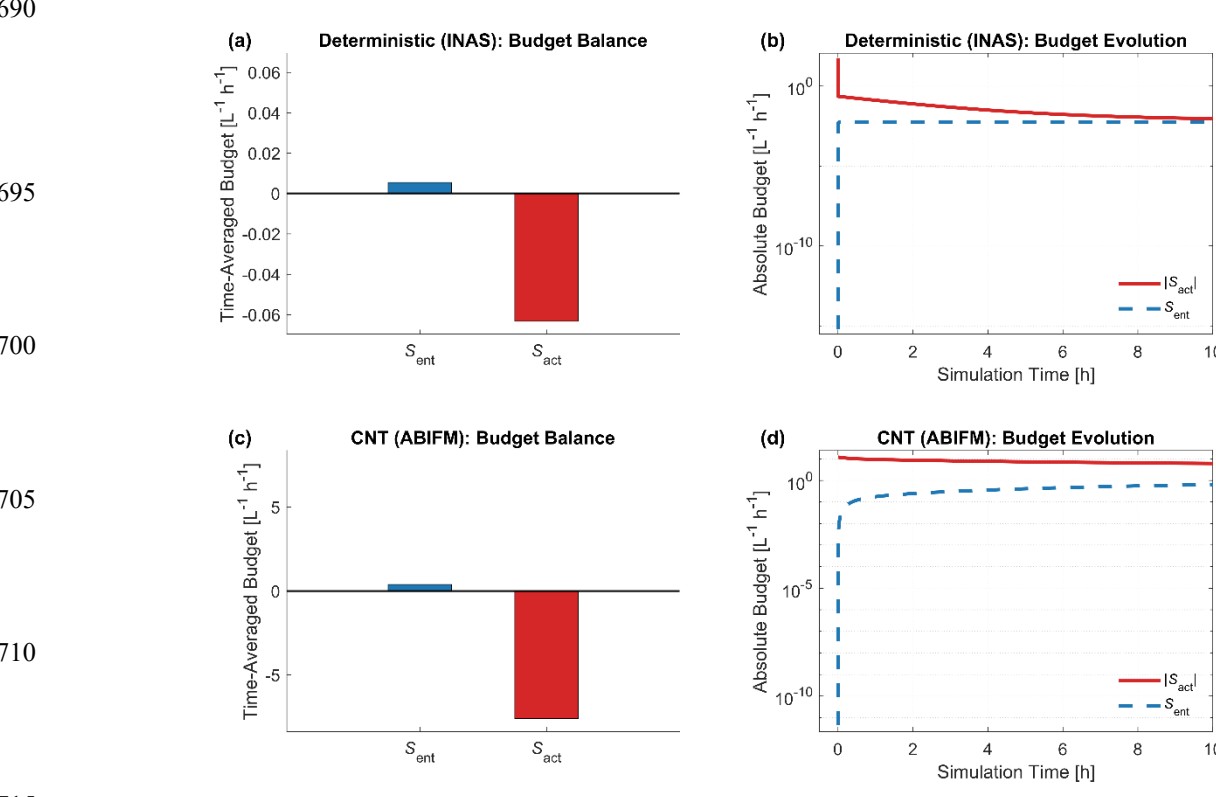






**Figure 10: A comparative budget analysis for the control (CTRL) simulation with the mineral dust PSD, illustrating the fundamentally different controls on INP reservoirs applying either deterministic parameterization (INAS) (top row) and the CNT-based parameterization (ABIFM) (bottom row). Panels a and c provide a quantitative summary of the time-averaged domain INP**
**budget balance. Panels b and d reveal the underlying dynamic evolution of the absolute magnitudes of these budget terms over the 10-hour simulation. Note the logarithmic y-axis in panels (b) and (d). All budget terms represent domain-averaged rates in units of L$^{-1}$ h$^{-1}$.**





## 4 Discussion

### 4.1 Parameterization Choice Dominates Simulated Ice Phase Evolution

The most striking outcome of this study is the profound impact of the chosen IMF parameterization on the simulated lifecycle of INPs and ice crystals in Arctic mixed-phase stratus over all composition classes and PSDs (Figs. 3, 4, and 5). The orders-of-magnitude difference in the predicted INP reservoir size and $N_i$ between deterministic and CNT-based approaches highlights a fundamental structural uncertainty in representing ice nucleation. Deterministic schemes, by defining INPs as a limited, instantaneously activating subset of aerosols (e.g., Demott et al., 2010; Niemand et al., 2012; Demott et al., 2015),

inherently lead to more rapid INP depletion and a more transient ice phase, rendering them highly dependent on continuous replenishment mechanisms. This aligns with previous modeling studies that have shown the necessity of INP replenishment mechanisms to sustain ice in such frameworks (Fridlind et al., 2012; Fridlind and Ackerman, 2018; Knopf et al., 2023b, and references therein).

Conversely, CNT-based approaches, which treat all aerosol particles in their composition class as potential INPs activating

continuously over time (e.g., Koop et al., 2000; Knopf and Alpert, 2013, 2023a), maintain a vast INP reservoir. This results in more persistent and significantly higher $N_i$, suggesting that Arctic MPCs could sustain a more vigorous ice phase if nucleation follows such time-dependent kinetics. The implication is that the perceived need for extremely efficient INP replenishment or additional ice production mechanisms may be reduced if CNT-like processes are dominant. The similarity between ABIFM and ABIFM* results further suggests that, for the simulated conditions, once saturation is achieved, in-cloud nucleation

processes appear to be more critical than slow freezing in subsaturated layers below the cloud base for these simulations and conditions.

### 4.2 Aerosol Characteristics as Modulators of Ice Formation

Within each parameterization framework, aerosol type and PSD further refine ice nucleation predictions. The strong dependence of INAS results on aerosol type (Figs. 3, 4, 5) underscores the importance of accurate $n_s$ parameterizations

(Niemand et al., 2012; China et al., 2017; Alpert et al., 2022). The dominance of mineral dust governing $N_i$, despite lower $N_{aer}$, is consistent with laboratory findings of its higher $n_s$ values compared to many organic species or SSA at typical MPC temperatures (Hoose et al., 2012a; Kanji et al., 2017). The INN scheme's primary reliance on $N_{aer} > 0.5\mu m$ makes it more sensitive to the aerosol number concentration (Table 1) than to aerosol type differences, a simplification that might not capture the full spectrum of INP behavior (Demott et al., 2010; Knopf et al., 2021). For CNT schemes, while all particles are potential

INPs, the aerosol-specific $J_{het}$ values (Alpert and Knopf, 2016) ensure that composition and particle size play a role in determining the rate of ice formation, with dust again often showing higher efficacy.

### 4.3 Interpretation of Cloud System Parameter Sensitivities

#### 4.3.1 Cloud Cooling and Entrainment: Critical Forcing Mechanisms in Deterministic schemes





The heightened sensitivity of deterministic schemes to CCR and $w_e$ (Figs. 3, 4, and 5; Tables 6 and 7) reinforces their reliance

on external drivers to mitigate the rapid depletion of INPs. Our vertical profile analysis (Figs. 6-8) further illustrates these sensitivities by pinpointing how and where different forcings, cloud cooling, cloud-top entrainment impact the cloud properties. These results highlight a fundamental limitation: deterministic schemes cannot sustain ice crystal production on a par with CNT schemes without continuous external forcing, resulting in significant differences in ice production within persistent mixed-phase clouds.

In deterministic schemes, CCR increase the number concentration of activatable INP by lowering temperatures to the activation thresholds of more INPs (Ullrich et al., 2017). This explains the significant increase in ice formation across heights observed in the vertical profiles (Figs. 7 and 8). In contrast, entrainment becomes a direct flux of new INPs that are immediately converted to ice at the cloud top, rather than replenishing a standing INP reservoir. This is why $N_{INP}$ itself (Fig. 3a) shows no direct change with $w_e$, while $N_i$ is substantially enhanced, leading to a much higher final $N_i$ than in the CTRL run (Table 7),

with the increase in $N_i$ sharply localized at the cloud top (Figs. 7 & 8).

CNT-based schemes exhibit markedly different behavior. Due to the immense INP reservoir, their response is governed by nucleation kinetics. Cloud cooling therefore drives a strong, layer-wide increase in ice production by enhancing the nucleation rate, while the minor addition of particles from cloud-top entrainment has a negligible impact on the final ice crystal concentration (Table 7).

The different responses to cloud forcing are a direct consequence of how the INP reservoir is defined. Deterministic schemes, which assume a scarce INP reservoir derived from a subset of aerosols, are critically dependent on cloud cooling and cloud-top entrainment to mitigate their rapid depletion. In contrast, CNT-based schemes, which treat the entire aerosol population as an abundant potential reservoir, are highly responsive to cooling that accelerates freezing but are largely indifferent to the minor perturbation of entrainment. This foundational difference explains why for the deterministic approach, even strong,

continuous CCR and cloud-top entrainment can only mitigate depletion, not prevent it - a direct consequence of its initial limited INP reservoir.

### 4.3.2 Sedimentation Controlling Ice Sink

Sedimentation, controlled by $v_f$, is the primary removal mechanism for ice crystals across all tested IMF parameterizations (Table 7). Our vertical analysis provides confirmation of this unifying role.

Regardless of the IMF parameterization or the specific forcing applied , the vertical profiles of the change in ice concentration ($\Delta N_i(z)$) consistently show their greatest magnitude—whether positive or negative—in the upper portion of the cloud, with the effect diminishing toward the cloud base (Fig. 7). This robust vertical structure is the clear signature of gravitational removal acting on a distributed, in-cloud source of ice. The highest rates of ice production occur in the coldest, uppermost cloud layers. At the same time, sedimentation continuously removes ice crystals from all levels, transporting them downward.



This interplay between in-cloud ice production and removal creates a natural vertical gradient. The impact of any process perturbation is therefore strongest near the region of most active ice formation and is progressively diminished by sedimentation at lower altitudes.

These results demonstrate that ice crystal fall speed emerges as a critical parameter governing cloud ice distribution, regardless of the specific microphysical processes involved. Consequently, accurate representation of ice particle fall speeds becomes

essential in models.

### 4.4 Competitive Ice Nucleation: Insights from Externally-Mixed Aerosol Simulations

Each IMF parameterization uses different physical assumptions that determine how aerosol particles contribute to the activatable INP population, creating distinct relationships between the initial aerosol population, the activatable INP reservoir, and the resulting $N_i$.

INN parameterization rely on aerosol  number concentration. Since SSA and organic aerosols dominate the number concentration in this size range, they predominantly comprise the initial activatable INP pool (Fig. 9b). Consequently, the resulting ice crystal population reflects this abundance pattern, with SSA serving as the primary contributor to ice formation (Fig. 9e).

In contrast, the INAS parameterization is based on aerosol type and surface area. The freezing efficiency for each aerosol type

is described by $n_s$. For the examined external aerosol, our simulations demonstrate that the $n_s$ term governs the ice nucleation response. This is because, the $n_s$ values for mineral dust exceed those of SSA and organic particles by several orders of magnitude. This substantial difference in intrinsic ice nucleation efficiency enables the relatively sparse dust particles to dominate the activatable INP reservoir (Fig. 9c), despite their lower number concentrations. Accordingly, dust particles control the subsequent ice crystal formation (Fig. 9f).

The CNT-based ABIFM parameterization assumes a time-dependent freezing process. Hence, all aerosol particles serve as potential INPs, such that the initial activatable INP reservoir mirrors the total aerosol number distribution, dominated by SSA and organic particles (Fig. 9c). However, the final $N_i$ is dominated by dust  (Fig. 9g). Similar to INAS, this is due to the larger nucleation rate coefficient ($J_{het}$) of dust compared to SSA and organic particles. Over the 10-hour simulation period, the substantially higher nucleation rates of dust particles enable more rapid ice crystal production, ultimately outcompeting other

aerosol types.

This study demonstrates our model's capacity to prognostically evaluate competing ice microphysics parameterizations using realistic aerosol PSDs. By simulating competitive nucleation between multiple aerosol populations, we move beyond single-type sensitivity tests toward conditions that better represent atmospheric complexity. These results address a fundamental question in ice microphysics: Is atmospheric ice formation controlled by rare but highly efficient particles, or by the collective

contribution of abundant aerosols? Our modeling framework provides the quantitative tools needed to test these competing paradigms against field observations, ultimately constraining which parameterizations most accurately reproduce observed cloud microphysical properties.



### 4.5. Process-Level Controls on INP Evolution: A Budget Perspective

The budget analysis reveals contrasting behaviors between the INAS and CNT (ABIFM) parameterizations: although their
time-averaged activation-to-entrainment ratios are similarly around~10:1 (Fig. 10a,c), the absolute magnitudes differ by two
orders of magnitude, and the temporal evolution diverges markedly (Fig. 10b,d). The deterministic parameterization
exemplifies source-limited behavior. Following rapid depletion of the INPs (Fig. 10b), ice production becomes entirely
controlled by cloud-top entrainment of INPs. The modest long-term budget balance (Fig. 10a) reflects this supply limitation,
where $S_{ent}$ dictates ongoing ice formation, independent of the freezing parameterization.
In contrast, the CNT parameterization exhibits process-rate-limited dynamics. The sustained, high-magnitude activation sink
(Fig. 10d) demonstrates nucleation acting on an abundant aerosol reservoir, with $S_{act}$ vastly exceeding $S_{ent}$ (Fig. 10c). Here,
ice production is governed by the intrinsic freezing kinetics under ambient thermodynamic conditions, enabling rapid
glaciation potential absent in the deterministic framework.
This process-level comparison demonstrates that choice of freezing parameterization represents a selection between
fundamentally different hypotheses for primary ice formation. The stark differences in baseline dynamics (Fig. 10) underscore
how this choice directly impacts simulation results.

### 5 Summary and Conclusions

The objective of this study is to dissect the key processes and uncertainties that govern ice formation in Arctic mixed-phase
clouds. First, we explore structural uncertainty in primary ice formation by evaluating how three different immersion freezing
ice nucleation parameterizations influence the microphysical evolution of mixed-phase Arctic stratus in the SHEBA MPC
case, using three aerosol types with distinct particle size distributions. Second, we assess the sensitivity of INP reservoir
dynamics — specifically $N_{INP}$ and $N_i$ — to variations in aerosol number concentration ($N_{aer}$), and key cloud and atmospheric
parameters, including cloud cooling rate (CCR), entrainment velocity ($w_e$), and ice crystal fall speed ($v_f$). Furthermore, we
investigate competitive nucleation among different aerosol types in a more realistic scenario using externally-mixed aerosol
populations and conduct a detailed INP budget analysis to diagnose and quantify the dominant processes controlling the INP
reservoir. We address these objectives using a version of the 1D-AC model extended to include multiple aerosol types and
their specific freezing parameterizations, informed by LES of the SHEBA case (Knopf et al., 2023b). The main findings can
be briefly summarized as follows.

1. When treating INPs prognostically as done in this study, the choice of IMF parameterizations can yield contrasting
concentrations and trends of simulated $N_{INP}$, $N_i$, and $dN_i/dt$. Compared to deterministic approaches, a time-
dependent description (CNT) yields an orders of magnitude larger INP reservoir, a finding consistent with previous
work (Knopf et al., 2023b). This leads to ice crystal formation at a substantially higher rate without significant INP
depletion , resulting in a simulated ice phase that is more persistent and not limited by the same replenishment fluxes
as the deterministic schemes.



2. Different aerosol types and associated PSDs significantly modify the magnitude of the simulated $N_{INP}$, $N_i$, and $dN_i/dt$ but do not alter their trends observed in the control run. The relative roles of aerosol types and associated PSDs are primarily dependent on the choice of IMF parameterizations.

3. Among all microphysical parameters tested, changing the $N_{aer}$ elicits a directly proportional response in the simulated $N_{INP}$, $N_i$, and $dN_i/dt$ across all parameterizations. For example, increasing $N_{aer}$ by a factor of 10 leads to a corresponding ten-fold increase in these variables, and vice-versa.

4. Changing the $N_{aer}$ has a directly proportional impact on $N_{INP}$, $N_i$, and $dN_i/dt$. In contrast, cloud cooling (CCR), $w_e$, and $v_f$ have non-linear effects, with the dominant process for ice production depending on the chosen parameterization. For deterministic approaches, where the INP reservoir is limited, ice production is critically dependent on the $w_e$ for replenishment. For CNT-based approaches with an abundant INP reservoir, CCR becomes the dominant control; it can reverse the overall decline in ice crystal concentration and, specifically for organic aerosols, drive $N_i$ to levels that substantially exceed the initial peak.

5. Initiating IMF under subsaturated conditions (when using ABIFM) leads to a more gradual trend in vertically resolved cloud properties around the cloud base compared to in-cloud only nucleation. These trends do not significantly impact the conclusions of the cases investigated in this study. However, we cannot fully rule out the potential importance of parameterization choices in accounting for IMF under subsaturated conditions more generally.

The 1D-AC model simulation results support the following conclusions. Our findings underscores the importance of the IMF parameterization selection in modeling mixed-phase Arctic stratus. Although each examined IMF parameterization was derived from laboratory experiments, the significant differences, particularly between deterministic and CNT-based approaches, when simulating $N_{INP}$, $N_i$, and $dN_i/dt$ from aerosol PSDs highlight the profound uncertainty in extrapolating these frameworks to atmospheric conditions and the necessity to constrain them with in-situ observations. Considering that primary ice formation is a ubiquitous phenomenon in Arctic MPCs, characterized by a low liquid water path (Silber et al., 2021), there is an urgent need for more laboratory and field research to better assess the scaling of IMF parameterizations under realistic MPC conditions (Knopf et al., 2020; Knopf et al., 2021; Burrows et al., 2022; Knopf and Alpert, 2023a).

The starkly different model behaviors emerging from these two different freezng parameterizations have significant implications for climate modeling. The CNT framework predicts sustained ice production without the same dependency on strong, continuous replenishment fluxes seen in the deterministic framework. As a consequence, interpretation of the evolution of a mixed-phase cloud including cloud lifetime and radiative properties will be different and will crucially depend on which freezing description is more representative of nature. A separate, practical advantage of the CNT approach is its computational simplicity. CNT-based methods do not require the INP array at all. This simplification removes the necessity of temperature-dependent INP arrays, reducing the complexity of the computational model. By eliminating the need to track additional variables for particle uniqueness, the model becomes more streamlined. Consequently, the time-dependent approach, which





provides freezing rates, enhances the feasibility of incorporating prognostic capabilities into global models with size-resolved aerosol modules (Bauer et al., 2008; Liu et al., 2016).

We evaluated three typical aerosol particle types: mineral dust, organic and SSA. Other potentially significant INP types (such as biological particles or biomass-burning aerosols) could also impact ice formation rates in Arctic MPCs. In future studies, the impact of other atmospheric INPs on cloud glaciation should be investigated. Continued work is needed to develop accurate ice nucleation parameterizations that cover the temperature range of realistic MPC conditions.

The observed non-linear responses of $N_{\mathrm{INP}}$, $N_{\mathrm{i}}$, and $dN_{\mathrm{i}}/dt$ to changes in CCR, $w_{\mathrm{e}}$, and $v_{\mathrm{f}}$ highlight the importance of detailed

cloud microphysical and dynamical measurements. Additional profiling observations of MPC temperatures, number concentration of free-troposphere aerosol particles, and the characterization of ice crystal fall velocity are essential for robustly evaluating the representation of INP reservoirs in mixed-phase Arctic stratus.



# APPENDIX A

**Immersion freezing parameterizations**

The deterministic INN parameterization of mineral dust particles acting as INPs is given by (Demott et al., 2015, D2015)

$$\sum_{T=T_{min}}^{T_{INP}} N_{INP}^{sing\,(INN)}(T) = cf\left(N_{aer>0.5\mu m}\right)^{1.25} e^{(0.46(-T_{INP})-11.6)}. \tag{A1}$$

$N_{INP}^{sing\,(INN)}(T)$ is the number concentration of all activatable INPs which are particles can serve as INPs and potentially be activated (hereafter referred to as INPs) at temperature $T$, $cf$ is the calibration factor with a value of 3 suggested when applying this parameterization to atmospheric data, $N_{aer>0.5\mu m}$ is the total number of aerosol particles with diameters larger than 0.5 μm, and $T$ is the temperature in the unit of Celsius.

A similar deterministic INN parameterization of ambient aerosol particles acting as INPs is given by (Demott et al., 2010, D2010)

$$\sum_{T=T_{min}}^{T_{INP}} N_{INP}^{sing\,(INN)}(T) = 0.0000594(-T)^{3.33}\left(N_{aer>0.5\mu m}\right)^{(0.0264(-T_{INP})+0.0033)}. \tag{A2}$$

We applied this IMF parameterization (Eq. A2) for the case of SSA and organic particles while noting that this does likely not reflect SSA and organic particles but represents ambient atmospheric particles in general.

In the 1D-AC model, the INN parameterizations are implemented as follows:

$$N_{INP}^{Imm}(z_i, t, k_*^{Imm}) = N_{INP}^{sing\,(INN)}(z_i, t, T_{INP})H(T(z_i, t)), \tag{A3}$$

where $N_{INP}^{sing\,(INN)}$ is multiplied by a Heaviside function $H(T(z_i, t))$, which equals to 1 when $T(z_i, t) > T_{INP}$ and 0 otherwise. This function ensures that INP activation occurs only when the temperature is lower than the activation temperature of INPs. In the model configuration used in this study, the array of $T_{INP}$ designates the activation temperature of INPs per temperature bin using geometric progression bins.

The INAS parameterization of natural dust particles acting as immersion INPs is given as (Niemand et al., 2012, N2012):

$$\sum_{T=T_{min}}^{T_{INP}} N_{INP}^{sing\,(INAS)}(T, d) = N_{aer}(d)(1 - e^{-A_{aer}(d)n_s(T_{INP})}) \tag{A4}$$

$$n_s^{dust}(T_{INP}) = e^{(-0.517(-T_{INP})+8.934)}. \tag{A5}$$

$N_{INP}^{sing\,(INAS)}(T, d)$ is the number concentration of all activatable INPs in temperature bin $T$ and diameter bin $d$, $N_{aer}$ is the total number concentration of aerosol particles, $A_{aer}$ is the individual particle surface area, $n_s$ is the INAS density in units of m$^{-2}$.

For organic particles (humic-like substances) and SSA, the $n_s$ is given by China et al. (2017) (C2017) and (Alpert, 2022) (A2022), respectively, as

$$n_s^{organic}(T_{INP}) = 10^{(66.90259(\Delta a_w)-12.322)} \tag{A6}$$

$$n_s^{SSA}(T_{INP}) = 10^{(24.02526(\Delta a_w)-2.26105)}, \tag{A7}$$





where $\Delta a_w$ is the water activity criterion, given below.

In this model, the INAS parameterizations are implemented as follows:

$$N_{\text{INP}}^{\text{Imm}}(z_i, t, k_*^{\text{Imm}}) = N_{\text{INP}}^{\text{sing (INAS)}}(z_i, t, d, T_{\text{INP}}) H\big(T(z_i, t)\big). \tag{A8}$$

For the INAS case, the $T_{\text{INP}}$ array is set same as in the INN case, while the $d$ array is defined using 50 bins, with a bin to bin mass ratio of 1.5. This implies that each subsequent bin has 1.5 times as much the mass as the previous one.

ABIFM parameterizes the heterogeneous ice nucleation rate coefficient $J_{het}$, as a function of the water activity criterion, $\Delta a_w$ (Knopf and Alpert, 2013). Derived from the water activity at the ice melting temperature, $a_w^i$, and water activity at the freezing temperature, $a_w$, $\Delta a_w$ is used in ABIFM to describe IMF as a function of ambient $T$ and RH:

$$\Delta a_w(T) = a_w(T) - a_w^i(T). \tag{A9}$$

Notice that, for IMF from water droplets, $a_w(T) = 1$, while under subsaturated conditions, $a_w(T) < 1$. The equation for $a_w^i(T)$ is given in Koop et al. (2009). To evaluate freezing at subsaturated conditions, i.e., at RH > 90%, we conducted additional model simulations termed ABIFM*.

ABIFM and ABIFM* derive $J_{het}$ in units of cm$^{-2}$ s$^{-1}$ for mineral dust, organic particles, and SSA particles in the following way (Alpert and Knopf, 2016; China et al., 2017; Alpert et al., 2022), respectively:

$$\log_{10}\left(J_{\text{het}}^{\text{dust}}\big(\Delta a_w(T)\big)\right) = 22.62\Delta a_w(T) - 1.35 \tag{A10}$$

$$\log_{10}\left(J_{\text{het}}^{\text{organic}}\big(\Delta a_w(T)\big)\right) = 66.90259\Delta a_w(T) - 13.40148 \tag{A11}$$

$$\log_{10}\left(J_{\text{het}}^{\text{SSA}}\big(\Delta a_w(T)\big)\right) = 26.6132\Delta a_w(T) - 3.9346. \tag{A12}$$

In the CNT case, the number concentration of activatable INPs, $N_{\text{INP}}^{\text{CNT (ABIFM)}}$, is given by

$$N_{\text{INP}}^{\text{CNT (ABIFM)}}(d) = N_{\text{aer}}(d), \tag{A13}$$

where $N_{\text{aer}}(d)$ is the number concentration of the applied aerosol particle type per bin.

In the 1D-AC model, the CNT-based IMF parameterizations are implemented as follows:

$$N_{\text{INP}}^{\text{Imm}}(z_i, t, k_*^{\text{Imm}}) = N_{\text{INP}}^{\text{CNT (ABIFM)}}(z_i, t, d). \tag{A14}$$

The $d$ array has the same characteristics as for the INAS case.





APPENDIX B

**INP array calculation for deterministic approaches**

In deterministic approaches, the model employs a decoupled framework that separates the intrinsic INP activation spectrum from its environmental expression. This design ensures consistent initialization across simulations while allowing dynamic evolution based on atmospheric conditions.

**Temperature array construction**

The temperature array is constructed to ensure finer resolution at lower temperatures, where ice nucleation is more sensitive.

This array is generated using a geometric progression, with the temperature bins increasing exponentially. The freezing array begins at a minimum temperature $T_{\min}$ = -38 °C and the temperature increments are determined by an initial temperature step $\Delta T_0$ (the $\Delta T$ between the first and second temperature bin edges) and an exponential factor $dT_{\exp}$ (the ratio of $\Delta T$ between consecutive bins).

The progression continues, but the array stops when the next temperature value is greater than or equal to the maximum

temperature $T_{\max}$ = -5 °C, meaning the last value in the array may not be reached at exactly $T_{\max}$. The temperature array is constructed as:

$$T_{\text{array}} = \left[ T_{min}, T_{min} + \Delta T_0, T_{min} + \Delta T_0 \cdot dT_{\exp}^{(1)}, T_{min} + \Delta T_0 \cdot dT_{\exp}^{(2)}, \dots \right] \text{ while } T < T_{max} \tag{B1}$$

**INP reservoir initialization**

Using established parameterizations (Appendix A), the model calculates the cumulative INP concentration active at or below

each temperature, then discretizes this into a differential activation spectrum $N_{\text{INP}}(T)$ representing INPs that activate specifically within each temperature bin. This creates the baseline INP reservoir array: $\text{INP}_{\text{array}}(z, T, t)$.

**Dynamic evolution and filtering**

During simulation, the INP reservoir evolves through entrainment of new INPs from external sources, permanent removal through activation when environmental temperature drops to or below a bin's activation temperature (conversion to ice crystals),

and vertical redistribution via mixing without changing total INP numbers.

At each timestep, the model applies a dynamic temperature mask based on the minimum column temperature to determine the potentially activatable INP concentration:

$$\text{INP}_{\text{totarray}}(z, t) = \sum_{T \geq T_{\text{cloud top}}} \text{INP}_{\text{array}}(z, T, t) \tag{B2}$$

This approach ensures that only INPs with activation temperatures warmer than current conditions remain in the activatable

reservoir, while consumed INPs are permanently removed. The decoupled framework provides consistent comparisons across simulations while capturing realistic INP dynamics.



APPENDIX C

**Governing equations**

Based on the following equations, the simplified 1D aerosol cloud model predicts the budgets for ice-nucleating particles (INPs) and ice particles (Knopf et al., 2023b):

$$\frac{dN_{\text{INP}}^{\text{Imm}}(z_i, t, k_*^{\text{Imm}})}{dt} = -S_{\text{act}}(z_i, t, k_*^{\text{Imm}}) + S_{\text{ent}}(z_i, t, k_*^{\text{Imm}}) + S_{\text{mix}}(z_i, t, k_*^{\text{Imm}}) \tag{C1}$$

$$\frac{dN_{\text{i}}(z_i, t)}{dt} = \sum_{k_*^{\text{Imm}}} S_{\text{act}}(z_i, t, k_*^{\text{Imm}}) - S_{\text{i}_{\text{sed}}}(z_i, t) + S_{\text{i}_{\text{mix}}}(z_i, t), \tag{C2}$$

where $N_{\text{INP}}$ presents the number concentration of activatable INPs and $N_{\text{i}}$ denotes the number concentration of ice crystals. The $z_i$ signifies the height at grid cell with index $i$ counted from bottom layer (1, surface) to the top layer (m, PBL top). Notice that the superscript Imm refers to the specific type of IMF parameterization used (INN, INAS, ABIFM or ABIFM*), and $k_*^{\text{Imm}}$ represents additional variable dimensions that depend on the chosen IMF parameterizations. For example, for application of INN, $k_*^{\text{Imm}}$ is $T_{\text{INP}}$ (reflecting the INP temperature arrays), for INAS, $k_*^{\text{Imm}}$ is a combination of $T_{\text{INP}}$ and $d$ (reflecting a 2D array consisting of INP temperature and diameter arrays), while for ABIFM and ABIFM*, $k_*^{\text{Imm}} = d$. Moreover, the model utilizes several variables to describe the budget terms. $S_{\text{act}}$, $S_{\text{ent}}$ and $S_{\text{mix}}$ represent the INP activation, cloud-top entrainment, and turbulent mixing budget terms, respectively. For ice particles, $S_{\text{i}_{\text{sed}}}$ and $S_{\text{i}_{\text{mix}}}$ denote the ice sedimentation and mixing budget terms. It is essential to note that ice crystals do not retain information about their associated INPs. Therefore, the activation budget is summed over the full $T_{\text{INP}}$ array (in INN), the $T_{\text{INP}}$ and $d$ arrays (in INAS) and the $d$ array (in ABIFM and ABIFM*).

In the model configuration employed in this study, we utilize a time splitting approach to calculate the budgets for ice-nucleating particles (INPs) and ice particles at every time step. The implicit solution for $S_{\text{act}}$ is given as

$$S_{\text{act}}(z_i, t, k_*^{\text{Imm}}) = \frac{N_{\text{INP}}^{\text{Imm}}(z_i, t, k_*^{\text{Imm}})}{\delta t + \tau_{\text{act}}}, \tag{C3}$$

where the time step length denoted by $\delta t$, is set by default to 10 s. For CNT-based parameterizations (ABIFM, ABIFM*), $\tau_{act}$ is calculated as $\tau_{\text{act}} = \frac{1}{J_{\text{het}}(T(z_i), a_{\text{w}}(z_i))\pi d^2}$. For deterministic parameterizations (INN, INAS) in this study, INP activation is treated as instantaneous once temperature conditions are met. This is implemented by directly converting the number of activatable INPs (determined by the respective deterministic parameterization at the given temperature) to ice crystals within the model timestep, effectively bypassing the explicit $\tau_{act}$-dependent formulation of $S_{\text{act}}$ shown in Eq. (C3) which is used for CNT schemes.The implicit Euler method (Hoffman et al., 2001), is used to ensure numerical stability, where $\delta t \ll \tau_{act}$ relevant for CNT schemes), the change in concentration is proportional to $\frac{-\delta t}{\delta t + \tau_{\text{act}}}$, which becomes increasingly similar to the explicit solution, where the change in concentration is proportional to $\frac{-\delta t}{\tau_{\text{act}}}$. However, this implicit Euler method ensures that the loss never exceeds the initial value.





To ensure numerical stability and avoid violating the Courant-Friedrichs-Lewy (CFL) condition, $S_{\text{ent}}$ is computed implicitly:

$$S_{\text{ent}}(z_m, t, k_*^{\text{Imm}}) = \frac{N_{\text{INP}}^{\text{Imm}}(z_m, t, k_*^{\text{Imm}}) + N_{\text{INP,FT}}^{\text{Imm}}(k_*^{\text{Imm}})}{\left(\delta t + \frac{\delta t}{w_e}\right)}, \tag{C4}$$

where $w_e$ is the entrainment rate and $N_{\text{INP,FT}}^{\text{Imm}}$ refers to the free-troposphere immersion freezing INP concentration, which is equal to the initial domain INP size distribution. Notice that $z_m$ represents the height of the domain top. Following Fridlind et al. (2012), $w_e$ is assumed to be 0.1 cm s⁻¹ based on LES results.

The mixing terms for INPs and ice particles are calculated as the following:

$$S_{\text{mix}}(z_i, t, k_*^{\text{Imm}}) = \frac{1}{\tau_{\text{mix}}}(\overline{N_{\text{INP}}^{\text{Imm}}(t, k_*^{\text{Imm}})} - N_{\text{INP}}^{\text{Imm}}(z_i, t, k_*^{\text{Imm}})) \tag{C5}$$

$$S_{\text{i}_{\text{mix}}}(z_i, t) = \frac{1}{\tau_{\text{mix}}}\left(\overline{N_i(t)} - N_i(z_i, t)\right), \tag{C6}$$

where $\tau_{\text{mix}}$ is the PBL mixing time scale, $\overline{N_{\text{INP}}^{\text{Imm}}}$ is the PBL mean activatable INP number concentration averaged over the whole domain, and $\overline{N_i(t)}$ denotes the PBL mean $N_i$.

Lastly, the sedimentation term for ice crystals is determined as follows:

$$S_{\text{i}_{\text{sed}}}(z_i, t) = \frac{v_f}{\delta z}\left(N_i(z_{i+1}, t) - N_i(z_i, t)\right), \tag{C7}$$

where $v_f$ is the number-weighted ice sedimentation rate (ice crystal fall velocity), maintained as a constant value based on LES estimates of circa 30 cm s⁻¹ (Fridlind et al., 2012). To prevent potential numerical instability, we always set the $v_f \cdot \delta t$ to be smaller than $\delta z$, thus leading to CFL < 1. Also, note that in Eq. (C7), the $i$ ranges from 1 to $m$-1 (index of layer below the PBL top) instead of from 1 to m.

As shown above, we account for the loss and gain of INPs and ice crystals, thus, treating the INPs and ice crystals prognostically.





APPENDIX D

Nomenclature

| | |
|---|---|
| $a_\mathrm{w}$ [-] | Water activity |
| $a_\mathrm{w}^\mathrm{i}$ [-] | Ice melting point as a function of water activity |
| $d$ [µm] | INP diameter array |
| $\Delta a_\mathrm{w}$ [-] | Water activity criterion |
| $\delta t$ [s] | Time step length |
| $\delta z$ [m] | Model grid cell thickness (vertical resolution) |
| $J_\mathrm{het}$ [cm$^{-2}$ s$^{-1}$] | Heterogeneous ice nucleation rate coefficient |
| $k_*^\mathrm{Imm}$ [°C], [°C µm], or [µm] | Additional variable dimensions depending on the applied parameterization |
| $N_\mathrm{aer}$ [cm$^{-3}$] | Aerosol number concentration |
| $N_{\mathrm{aer}>0.5\mu m}$ [cm$^{-3}$] | Aerosol number concentration for particles larger than 0.5 µm |
| $N_\mathrm{i}$ [L$^{-1}$] | Ice crystal number concentration |
| $\overline{N_\mathrm{i}(t)}$ [L$^{-1}$] | PBL mean ice crystal number concentration |
| $N_\mathrm{INP}$ [L$^{-1}$] | Activatable INP number concentration |
| $\overline{N_\mathrm{INP}}$ [L$^{-1}$] | PBL mean activatable INP number concentration |
| $N_\mathrm{INP}^\mathrm{Imm}$ [L$^{-1}$] | Activatable INP number concentration for given immersion freezing parameterization |
| $N_\mathrm{INP,FT}$ [L$^{-1}$] | Free-troposphere activatable INP concentration |
| $N_\mathrm{INP}(T)$ [L$^{-1}$] | The number concentration of INPs that activate specifically within temperature bin |
| $n_\mathrm{s}$ [cm$^{-2}$] | Ice nucleation active sites density |
| RH [%] | Relative humidity expressed in percent |
| $S_\mathrm{act}$ [m$^{-3}$ s$^{-1}$] | INP activation budget term |



| | |
|---|---|
| $S_{\text{act}}^{\text{Imm}}$ [m$^{-3}$] | INP activation budget term calculated based on the selected immersion freezing parameterizations |
| $S_{\text{act}}^{\text{sing (INN)}}$ [m$^{-3}$] | INP activation budget term calculated based on the INN |
| $S_{\text{act}}^{\text{sing (INAS)}}$ [m$^{-3}$] | INP activation budget term calculated based on the INAS |
| $S_{\text{act}}^{\text{CNT (ABIFM)}}$ [m$^{-3}$] | INP activation budget term calculated based on the ABIFM |
| $S_{\text{ent}}$ [m$^{-3}$ s$^{-1}$] | Cloud-top entrainment budget term |
| $S_{\text{ice}}$ [-] | Supersaturation with respect to ice |
| $S_{i_{\text{mix}}}$ [m$^{-3}$ s$^{-1}$] | Ice mixing budget term |
| $S_{i_{\text{sed}}}$ [m$^{-3}$ s$^{-1}$] | Ice sedimentation budget term |
| $S_{\text{mix}}$ [m$^{-3}$ s$^{-1}$] | Turbulent mixing budget term |
| $t$ [s] | Model time step |
| $\tau_{\text{act}}$ [s] | Activation time scale |
| $T_{\text{INP}}$ [°C] | INP temperature array |
| $\tau_{\text{mix}}$ [s] | PBL mixing time scale |
| $v_{\text{f}}$ [m s$^{-1}$] | ice sedimentation rate |
| $w_{\text{e}}$ [cm s$^{-1}$] | Entrainment rate |
| $z_i$ [m] | Height at grid cell index $i$ |
| $z_m$ [m] | Height at domain top (grid cell index $m$) |
| INP$_{\text{array}}$ [m$^{-3}$] | Three-dimensional array that stores calculated INP values. |
| INP$_{\text{totarray}}$ [m$^{-3}$] | The sum of all INPs with activation temperatures warmer than the current cloud conditions that remain available for ice nucleation. |



APPENDIX E

List of Abbreviations

| | |
|---|---|
| 2D-C | Two-dimensional cloud optical array probes |
| 2D-P | Two-dimensional precipitation optical array probes |
| ABIFM | The water-activity-based immersion freezing model |
| ABIFM$^*$ | The water-activity-based immersion freezing model which enables immersion freezing commencing under subsaturated conditions |
| CCR | Cloud cooling rate |
| CFL | Courant–Friedrichs–Lewy condition |
| CNT | Classical nucleation theory |
| CPI | Cloud Particle Imager |
| DHARMA | Distributed Hydrodynamic Aerosol and Radiative Modeling Application |
| ICEALOT | International Chemistry Experiment in the Arctic Lower Troposphere |
| INAS | Ice nucleation active sites-based parameterization |
| INN | Ice nucleation number-based parameterization |
| INP | Ice nucleating particle |
| ISDAC | Indirect and Semi-Direct Aerosol Campaign |
| LES | Large-eddy simulation |
| MMCR | Millimeter Wavelength Cloud Radar |
| M-PACE | Mixed-Phase Arctic Cloud Experiment |
| MPC | Mixed-phase cloud |
| PBL | Planetary boundary layer |
| PSD | Particle size distribution |
| SHEBA | The Surface Heat Budget of the Arctic campaign |
| SSA | Sea spray aerosol |



*Code and data availability.* The 1D model version used in this manuscript is available on Zenodo at
https://doi.org/10.5281/zenodo.7108690 (Silber et al., 2022). The most current version of the model is available on Zenodo
at https://doi.org/10.5281/zenodo.16414825 (Sun et al., 2025a). The model output data is available on Zenodo at
https://doi.org/10.5281/zenodo.16413525 (Sun et al., 2025b). The data analysis and plotting scripts for the manuscript is
available on Zenodo at https://doi.org/10.5281/zenodo.16414282 (Sun et al., 2025c).

*Author contributions.* YS modified the Python code of the 1D-AC model with guidance from IS, performed all sensitivity
tests, and wrote the first draft of the paper, DAK supervised the project. DAK, IS, NR, and AMF assisted in methodology
and results validation. All authors discussed interpretation of the data and contributed to writing and reviewing of the paper.

*Competing interests.* The contact author has declared that none of the authors has any competing interests.

*Disclaimer.* Publisher's note: Copernicus Publications remains neutral with regard to jurisdictional claims in published maps
and institutional affiliations.

*Acknowledgements.* This study was supported by the U.S. Department of Energy's Atmospheric System Research, an Office
of Science Biological and Environmental Research program, under Grant DE-SC0021034.

*Financial support.* This study was supported by the U.S. Department of Energy's Atmospheric System Research, an Office
of Science Biological and Environmental Research program, Grant DE-SC0021034.




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
