# Peer review of "Prognostic simulations of mixed-phase clouds with model 1D-AC v1.0: The impact of freezing parameterizations on ice crystal budgets"

_EGUsphere, 2025_

## Referee Comment (RC1)

(egusphere-2025-3620)

This is a modeling study that focuses on Arctic mixed-phase stratus and their sensitivity to different formulations of the immersion freezing mechanism (IMF). Sensitivity simulations are performed with a simplified 1D aerosol-cloud model that investigates a lot of interesting aspects, such as the different impact of using deterministic and CNT IMF schemes on the cloud ice crystal budget and their response to variations in aerosol load, microphysical properties (terminal velocity) and thermodynamic parameters (cloud-cooling and cloud-top entrainment rate). Overall, it is an interesting study that highlights how sensitive is the representation of the complex interactions between primary ice production and thermodynamic/aerosol processes to the choice of IMF parameterization. However my main worries concern whether the findings are truly representative of the Arctic mixed-phase cloud conditions

**MAIN COMMENTS:**

(A) The case study is constructed using thermodynamic measurements from SHEBA and aerosol inputs from ISDAC and ICEALOT campaigns. While ISDAC and ICEALOT occurred in spring, it is not clarified to which season the SHEBA case corresponds to. The Arctic aerosol composition exhibits seasonal and spatial variability with long-range transport of dust and anthropogenic aerosols peaking in late winter—spring ("Arctic haze") and marine organic/sea-spray sources dominating in summer. This seasonal variability is also reflected in the INP composition/origin (Creamean et al. 2022).

Here, the prescribed dust load appears low, yet multiple observational studies show that dust intrusions into the Arctic can be important during certain periods, often linked to springtime transport from Asian or Saharan sources. Similarly, sea-spray emissions depend on open-water fraction and wind-driven surface conditions, which vary seasonally and geographically. Therefore, a short discussion on how the chosen PSDs and thermodynamic properties align in season and location would be useful, along with clarifications on the Arctic conditions that are represented by this case.

- (B) The authors mainly show results related to the activatable INPs and ice crystal number. I think it would be very useful to show results related to activated INPs and compare to the vast literature that has been recently published on Arctic INPs (e.g. Wex et al. 2019, Creamean et al. 2022, etc). I suspect that the CNT formulations might predict INP concentrations that are outside (above) the observed range, suggesting that these two parameterizations are not suitable for Arctic conditions.
- (C) I think that the highly idealized nature of these simulations is underdiscussed. Mixed-phase microphysical processes are complex and the impact of processes like WBF, riming and aggregation can largely affect the ice crystal size distribution and eventually the available ice crystal budget. Taking into account these processes could likely change the relative contribution of each IMF parameterization to the ice number and affect their interactions with other processes, like sedimentation. Moreover, there is increasing evidence that secondary ice production (SIP) is important in Arctic

stratocumulus. Accounting for SIP can also impact how the ice crystal budget responds to different IMF parameterizations (and whether the choice of IMF remains that important). All these uncertainties should be discussed in the final section. Considering a cloud that is unaffected by these processes suggests limited representativeness of the real Arctic conditions

**MINOR COMMENTS:**

Line 185: you probably refer to Figure 2. What is the liquid water path range?

Line 216: Here it is mentioned that SSA PSDs are based on measurements above sea, while the SHEBA case corresponds to pack-ice conditions (see main comment A)

Line 218: composite. Figure 1 should be 2

Section 2.3: Are primary ice production and sedimentation the only microphysical processes accounted in the model? This should clearly stated that other important processes are ignored (e.g. WBF, riming, etc) Also are there any aerosol processes accounted for?

Line 278: Could you provide references for dust being negligible in the Arctic? There are many studies that do not support this claim (e.g. Boo et al. 2023; Creamean et al 2022)

Section 2.4: The description of sensitivity simulations is a bit confusing. E.g. CCR=0.3 is listed as sensitivity test, while based on the caption of figure 2, I would assume that the same CCR is applied in the CTRL simulation. If CCR is zero in CTRL simulation then this should be listen Table 4. If green profiles in Figure 2 concern only the sensitivity test and not CTRL case, this should be clearly explained in the caption

Line 285-287: Also it should be explicitly stated that CTRL simulation is run with a single aerosol type

Line 308: also shown in Figure 2(?)

Line 310: clarify that each PSD corresponds to different aerosol type

Line 503: Why organic INPs are more sensitive to temperature changes?

Line 576: black instead of brown

Fig 7 is confusing. There is a light solid brown line in panels a and b, not include in the legend. Also dashed blue line is not visible in panel a

Fig 8 is confusing. Is it the logarithmic scale that inhibits the demonstration of the whole vertical profile?

Lines 663-666: why does this happen?

Lines 737-738: I would rephrase. For example, regarding the need of additional ice production mechanisms, this is not simply a theoretical perception; observations indicate the occurrence of such mechanisms in Arctic clouds. Observing INP recycling of course is not possible. Moreover, please compare the CNT INP predictions to Arctic measurements (see main comment B). This way we might get an idea of whether CNT-like processes can be truly dominant in the Arctic, which is generally known as a low-INP region.

Lines 815-817: You can include an important test by including INP measurements from the literature (again comment B)

**REFERENCES**

Creamean, J.M., Barry, K., Hill, T.C.J. *et al.* Annual cycle observations of aerosols capable of ice formation in central Arctic clouds. *Nat Commun* 13, 3537 (2022). https://doi.org/10.1038/s41467-022-31182-x

Wex, H., Huang, L., Zhang, W., Hung, H., Traversi, R., Becagli, S., Sheesley, R. J., Moffett, C. E., Barrett, T. E., Bossi, R., Skov, H., Hünerbein, A., Lubitz, J., Löffler, M., Linke, O., Hartmann, M., Herenz, P., and Stratmann, F.: Annual variability of icenucleating particle concentrations at different Arctic locations, Atmos. Chem. Phys., 19, 5293–5311, https://doi.org/10.5194/acp-19-5293-2019, 2019.

---

## Author Comment (AC1)

**Response to Referee #1**

We thank reviewer 1 for thoughtful and constructive comments that helped us to improve the clarity and quality of our manuscript. Below we have provided a point-by-point response to each of the points raised by reviewer 1. The reviewer's comments are given in *italic black font* and our response in green with quotes in *italic font* style. The line numbers refer to the revised and highlighted manuscript file. Before addressing the specific points in detail, we would like to give a **general response** to comments given by both reviewers referring to the idealized nature of this study.

This work introduces the AC-1D model, which provides a novel framework to prognostically treat INP and ice crystal budgets while explicitly accounting for polydisperse aerosol inputs. A key capability of this model is the flexibility to apply different immersion freezing (IMF) parameterizations, including deterministic and time-dependent (CNT) schemes, to different aerosol types with unique particle size distributions.

By purpose, we chose to focus on INPs and ice crystal formation, and the role of supercooled water droplets in mixed-phase clouds is implicit to the model initialization of the thermodynamic state in the examined scenario. Cloud modeling has not yet been able to accurately determine the strength of the primary or secondary ice production (PIP or SIP) pathways. We believe this is in part due to the models' complexity trying to account for all different processes proceeding in the cloud. Here, we aim to assess the strength of the PIP in mixed-phase clouds since everything else follows from this. If we cannot constrain this aspect of the cloud sufficiently the remaining processes stay ambiguous.

During the development of this prognostic model, novel concepts and notions emerged including the concept of "activatable" INPs which reflect the maximum number of INPs available to form ice under given cloud conditions (not all activatable INPs necessarily form ice though) and the notion of an INP reservoir which is available to the cloud and, as we show, is defined by choice of freezing parameterization. This extends previous modeling studies.

In short, this bottom-up approach links specific polydisperse aerosol particle size distributions (PSDs) to their respective freezing behaviors, establishing the model as a robust testbed for investigating structural model uncertainties for user-provided cloud conditions (e.g., LES informed etc.). Specifically, the model architecture facilitates:

1. **Comprehensive prognostic treatment** of aerosol, INP, and ice crystal budgets, explicitly accounting for the loss of each property (e.g., via activation or sedimentation) to capture their temporal evolution.
2. **Flexible initialization** using realistic, polydisperse, and multi-component aerosol composition rather than simplified monodisperse inputs, as well as the ability to easily switch between prognostic and diagnostic modes or compare different freezing parameterizations simultaneously.

3. **Process-level diagnosis**, enabling the user to quantify detailed INP and ice crystal budgets (e.g., activation vs. sedimentation vs. entrainment) in response to user-defined perturbations in thermodynamic profiles, cooling rates, and microphysical parameters.

We developed this simplified model to isolate and quantify the structural uncertainty introduced by the choice of immersion freezing (IMF) parameterization. While we acknowledge that Arctic clouds are complex systems influenced by SIP and liquid-ice feedbacks (such as the Wegener-Bergeron-Findeisen process driving ice growth), including these processes in this initial study would obscure the primary signal we aim to investigate: the foundational uncertainty in ice crystal number concentrations due to different freezing parameterizations.

We make the following changes to the text to make these points clearer.

Abstract:

Line 15: We add the following statement:

"*We developed one-dimensional aerosol-cloud (AC-1D) model, which provides a novel framework to prognostically treat INP and ice crystal budgets while explicitly accounting for polydisperse and multicomponent aerosol that activate INPs following different freezing parameterizations.*"

We modified the text on line 155:

*"Building on these identified uncertainties in primary ice production (PIP), we now focus in detail on the broader characteristics of Arctic aerosol. In this study, we employ the AC-1D model as a prognostic tool designed to isolate the structural uncertainties in PIP. The impact of liquid-ice feedbacks (such as the Wegener-Bergeron-Findeisen process driving ice growth) and secondary ice production (SIP) on the ice crystal budget depends crucially on the accurate description of the PIP. A key feature of this framework is its ability to conduct comprehensive sensitivity analyses by coupling polydisperse and multicomponent aerosol inputs directly to the INP and ice budgets. This setup allows for the simultaneous, prognostic evaluation of fundamentally distinct IMF parameterizations, while permitting user-defined adjustments to thermodynamic profiles and cloud system parameters. By generating detailed process-level data, such as explicit INP and ice crystal budgets, the model serves as a robust testbed to determine how the choice of parameterization dictates the PIP and evolution of the INP reservoir in Arctic mixed-phase clouds (Knopf et al., 2023; Arabas et al., 2025)."*

We thank the reviewer for the constructive criticism of our work. The comments provided valuable perspective that helped us significantly improve the manuscript, particularly regarding the contextualization of our aerosol inputs and the comparison of our model results with observational INP data.

**MAIN COMMENTS:**

*(A) The case study is constructed using thermodynamic measurements from SHEBA and aerosol inputs from ISDAC and ICEALOT campaigns. While ISDAC and ICEALOT occurred in spring, it is not clarified to which season the SHEBA case corresponds to. The Arctic aerosol composition exhibits seasonal and spatial variability with long-range transport of dust and anthropogenic aerosols peaking in late winter– spring ("Arctic haze") and marine organic/sea-spray sources dominating in summer. This seasonal variability is also reflected in the INP composition/origin (Creamean et al. 2022).*
*Here, the prescribed dust load appears low, yet multiple observational studies show that dust intrusions into the Arctic can be important during certain periods, often linked to springtime transport from Asian or Saharan sources. Similarly, sea-spray emissions depend on open-water fraction and wind-driven surface conditions, which vary seasonally and geographically. Therefore, a short discussion on how the chosen PSDs and thermodynamic properties align in season and location would be useful, along with clarifications on the Arctic conditions that are represented by this case.*

We appreciate the reviewer's insight regarding the seasonal and spatial variability of Arctic aerosol composition.

We have rewritten the text in Section 2.2 (Lines 245-274 of the original manuscript) to explicitly justify the composite aerosol initialization based on microphysical consistency with the SHEBA case while referring to the reviewer comment details. The revised sections now read:

*"To evaluate the impact of different aerosol types on the INP reservoir, $N_i$ and ice crystal formation rate, we examine the effects of mineral dust, organic, and SSA particles. While the thermodynamic profile is derived from the specific SHEBA case study (early May) to represent a typical Arctic mixed-phase boundary layer, the aerosol initialization represents a composite of spring Arctic conditions derived from ISDAC and ICEALOT (April).*

*Since size-resolved aerosol measurements were not available for the specific SHEBA case study, we approximated the total aerosol inputs based on the observed cloud droplet number concentration ($N_d \approx 200$ $cm^{-3}$; Fridlind et al., 2012). Our model assumes that all in-cloud aerosols are activated.*

*To reconstruct a physically consistent aerosol population, we utilized data from the ISDAC campaign (for mineral dust and organic fractions) and the ICEALOT campaign (for SSA). The*

*ISDAC campaign is an appropriate proxy because it exhibited cloud microphysical properties very similar to SHEBA case, with measured $N_d$ values ranging from 185 to 205 cm$^{-3}$ (Savre et al., 2015).*

*To achieve quantitative alignment between the prescribed aerosol load and the SHEBA target ($N_d \approx 200$ cm$^{-3}$), we selected aerosol inputs corresponding to the "clean case" defined in Earle et al. (2011). This case shows number concentrations of aerosol ($N_{aer}$) less than 250 cm$^{-3}$, while $N_d$ is approximately $135 \pm 34$ cm$^{-3}$ (Earle et al., 2011). Critically, this case exhibits a high activation efficiency, validating our model assumption that all in-cloud aerosols are activated .Within this total number constraint, the relative compositional fractions of mineral dust and organic aerosol were derived from single particle analyses performed during the same field campaign (Hiranuma et al., 2013). Although the PSDs are derived from the subsequent flight (Flight 31, April 27), Earle et al. (2011) classify both dates as the same meteorological regime, justifying this composite initialization.*

*While mineral dust is a major component of Arctic INPs during long-range transport events (Creamean et al., 2022; Böö et al., 2023), the background conditions defined for this specific sensitivity study maintain low dust number concentration compared to organic and SSA particles to match the microphysical constraints of the SHEBA case. High-load scenarios, e.g., increase in dust load, are explored separately in sensitivity runs. Similarly, while the SHEBA case occurred over pack ice where local SSA emission is suppressed, we include the ICEALOT SSA distribution (Quinn et al., 2017) to account for potential transport from open water leads and to establish a robust background state for testing competitive nucleation processes.*

*For each aerosol type, the applied aerosol particle size distributions (PSDs) are polydisperse consisting of two or three lognormal modes (Table 1). In addition to Aitken and accumulation modes, this framework includes a larger accumulation mode for aged aerosols and a source-specific SSA mode. Lastly, to reflect a more realistic aerosol population we combine the mineral dust, organic, and SSA PSD (composite PSD). Figure 1 displays the lognormal PSDs of the different aerosol particle types and the composite PSD, derived from the modal parameters specified in Table 1."*

*(B) The authors mainly show results related to the activatable INPs and ice crystal number. I think it would be very useful to show results related to activated INPs and compare to the vast literature that has been recently published on Arctic INPs (e.g. Wex et al. 2019, Creamean et al. 2022, etc). I suspect that the CNT formulations might predict INP concentrations that are outside (above) the observed range, suggesting that these two parameterizations are not suitable for Arctic conditions.*

We appreciate this excellent suggestion. To address this, we have added a new figure (Fig. 3) to the beginning of Section 3.1, comparing our model's predicted INP concentration against a comprehensive set of recent Arctic field observations.

[Figure]

**Figure 3. Predicted activated INP concentrations from immersion freezing parameterizations are shown for mineral dust (black lines), SSA (blue lines), and organic (green lines) aerosols, calculated with their respective PSD from Table 1. Lines indicate the different schemes: INN (solid), INAS (dashed), and ABIFM (dotted). Note that the INN schemes use DeMott et al. (2015) for mineral dust and DeMott et al. (2010) for SSA and organic aerosol particles. Model predictions are compared against a composite of Arctic field observations from Wex et al. (2019) (colored shaded regions reflect Alert, Utqiaġvik, Ny-Ålesund, Villum) and Creamean et al. (2022) (vertical orange bars). The vertical pink background shading indicates the temperature range in the simulation domain.**

We have inserted the following text to validate the parameterizations against observations before discussing the simulation results at line 492:

*"As a crucial step for model evaluation, we first place the chosen IMF parameterizations into an observational context. This provides a baseline for interpreting the prognostic simulations that follow. To this end, we compare the activated INP concentrations predicted by each IMF parameterization (initialized with the specific aerosol PSDs from Table 1) against recent Arctic field observations.*

*Figure 3 compares these model predictions against a composite of Arctic data. The shaded regions represent the annual variability observed at four ground-based Arctic stations (Alert, Utqiaġvik, Ny-Ålesund, and Villum Research Station) utilizing filter samples for ice nucleation*

*experiments analyzed with a cooling rate of 1 °C min⁻¹ (Wex et al., 2019). Additionally, we show INP measurements recorded during the Multidisciplinary drifting Observatory for the Study of Arctic Climate (MOSAiC) expedition onboard the research vessel Polarstern (Creamean et al., 2022). These particle samples were analyzed using a cooling rate of 0.33 °C min⁻¹. The vertical bars indicate the minimum and maximum INP concentrations observed during the full annual cycle at -17.5, -20, and -22.5 °C, selected to match the temperature range of our simulated cloud layer. For the time-dependent ABIFM (CNT) parameterization, a nucleation time period is required to derive the cumulative activated INP number concentration. We apply  t = 1 min (following Alpert et al. (2022)) to approximate the nucleation time scales for the experiments.*

*As shown in Figure 3, the agreement depends on both the parameterization choice and the assumed aerosol type. The deterministic INAS scheme, when applied to SSA and organic aerosol PSDs, shows good agreement within the range of field observations. The ABIFM (CNT) predictions for organic aerosol also align well with the annual range observed during the MOSAiC campaign. While the deterministic INN scheme and dust parameterizations tend to predict concentrations at the upper end of the observational range, they generally fall within the total variability spanning the different stations and seasons. Except for a pure dust case, INAS and CNT parameterizations predict INP concentrations similar to the ones observed in the Arctic regions."*

[The original text starting at Line 521: *"The choice of IMF parameterization fundamentally dictates..."* follows here.]

*(C) I think that the highly idealized nature of these simulations is underdiscussed. Mixed-phase microphysical processes are complex and the impact of processes like WBF, riming and aggregation can largely affect the ice crystal size distribution and eventually the available ice crystal budget. Taking into account these processes could likely change the relative contribution of each IMF parameterization to the ice number and affect their interactions with other processes, like sedimentation. Moreover, there is increasing evidence that secondary ice production (SIP) is important in Arctic*
We acknowledge that Arctic mixed-phase clouds are complex systems where processes such as the Wegener-Bergeron-Findeisen process, riming, and SIP can play significant roles. However, the simplification of the AC-1D model, specifically the exclusion of liquid-phase feedbacks and SIP, is an intended design choice to isolate the role of PIP. Please see also our general response above.

The fundamental objective of this work is to quantify the uncertainty in INP and ice crystal budgets arising directly from the choice of immersion freezing parameterization. If the representation of PIP remains uncertain, the simulation of secondary processes that depend on this primary ice become highly speculative.

Therefore, this study provides a necessary bottom-up investigation, treating INPs prognostically based on the polydisperse aerosol population. This allows us to establish a baseline of uncertainty for PIP before adding the complexity of secondary processes.

It is important to note that not every mixed-phase cloud is prone to SIP as discussed in Fridlind et al. (2018) (chapter7, section 2.5 and 5.3). For example, a 6-year ground-based remote sensing study in the Arctic found that SIP events occurred in less than 10% of observed slightly supercooled clouds (temperatures > -10°C), even though this temperature range is considered optimal for the Hallett-Mossop rime splintering process (Luke et al., 2021). For the specific SHEBA case, observations indicated a low liquid water path, no precipitation, and sparse unrimed ice crystals (Fridlind et al., 2012), conditions under which SIP and riming were not dominant factors. We have expanded the Introduction and Discussion sections to explicitly frame this model as a robust prognostic framework designed to disentangle the fundamental structural uncertainties of primary ice formation. By offering a transparent environment to track the INP budget, this tool serves as an essential prerequisite step to more complex microphysical simulations.

We have revised the manuscript in three specific locations to explicitly frame the model as a prognostic testbed for PIP and to justify the exclusion of secondary processes for this specific case study.

First, in the Introduction (Line 155 at Page 5), we revised the text:

*"Building on these identified uncertainties in primary ice production (PIP), we now focus in detail on the broader characteristics of Arctic aerosol. In this study, we employ the AC-1D model as a prognostic tool designed to isolate the structural uncertainties in PIP. The impact of liquid-ice feedbacks (such as the Wegener-Bergeron-Findeisen process driving ice growth) and secondary ice production (SIP) on the ice crystal budget depends crucially on the accurate description of the PIP. A key feature of this framework is its ability to conduct comprehensive sensitivity analyses by coupling polydisperse and multicomponent aerosol inputs directly to the INP and ice budgets. This setup allows for the simultaneous, prognostic evaluation of fundamentally distinct IMF parameterizations, while permitting user-defined adjustments to thermodynamic profiles and cloud system parameters. By generating detailed process-level data, such as explicit INP and ice crystal budgets, the model serves as a robust testbed to determine how the choice of parameterization dictates the PIP and evolution of the INP reservoir in Arctic mixed-phase clouds (Knopf et al., 2023; Arabas et al., 2025)."*

Second, in the methods section (Section 2.3), We added the following paragraph at lines 331:

*"In this setup, PIP and ice crystal sedimentation are the dominant ice microphysical processes tracked. Processes such as the Wegener-Bergeron-Findeisen (WBF) process, riming, and SIP are deliberately excluded to isolate the uncertainty in INP and ice crystal budgets arising directly from the choice of the immersion freezing parameterization."*

Third, we added the following text at the end of section 4.1 (line 924):

*"Finally, the distinct ice crystal evolution trends observed here highlight the sensitivity of the system to the representation of PIP. By disentangling this process from liquid-phase feedbacks and SIP, we demonstrate that the choice of freezing parameterization constitutes a foundational source of uncertainty in the simulated ice crystal budget."*

**MINOR COMMENTS:**

*Line 185: you probably refer to Figure 2. What is the liquid water path range?*

We have corrected the reference to Figure 2. We have also added the liquid water path range observed for this case, which is approximately 5–10 g m$^{-2}$ (Fridlind et al., 2012), to Section 2.1 (Line 214).

*Line 216: Here it is mentioned that SSA PSDs are based on measurements above sea, while the SHEBA case corresponds to pack-ice conditions (see main comment A)*

We acknowledge the distinction pointed out by the reviewer. As discussed in detail in our response to Main Comment A, the SHEBA case indeed occurred over pack ice where local sea spray emission is suppressed. However, we intentionally included the ICEALOT SSA distribution to represent a "complete" Arctic background state that accounts for potential transport from open water leads or the marginal ice zone. This allows the model to serve as a robust testbed for competitive nucleation.

Please refer to Response to Comment A for the full revision of Section 2.2. Specifically addressing this concern, we have added the following clarification to the text in Section 2.2 (Line at 267):

*"Similarly, while the SHEBA case occurred over pack ice where local SSA emission is suppressed, we include the ICEALOT SSA distribution (Quinn et al., 2017) to account for potential transport from open water leads and to establish a robust background state for testing competitive nucleation processes."*

*Line 218: composite. Figure 1 should be 2*

*Corrected. We also corrected the figure numbering. To ensure figures appear in the order they are mentioned in the text, we have renumbered the first two figures. The thermodynamic profiles figure (discussed in Section 2.1) is now Figure 1, and the figure showing the aerosol particle size distributions (discussed in Section 2.2) is now Figure 2. The text was updated to reflect changes.*

*Section 2.3: Are primary ice production and sedimentation the only microphysical processes accounted in the model? This should clearly stated that other important processes are ignored (e.g. WBF, riming, etc) Also are there any aerosol processes accounted for?*

That is correct. We have rewritten the text in Section 2.3 (Lines 331–336) to clearly define the active and excluded processes.

*"In this setup, PIP and ice crystal sedimentation are the dominant ice microphysical processes tracked. Processes such as the Wegener-Bergeron-Findeisen (WBF) process, riming, and SIP are deliberately excluded to isolate the uncertainty in INP and ice crystal budgets arising directly from the choice of the immersion freezing parameterization. Regarding aerosol physics, the model treats the aerosol population prognostically, explicitly accounting for the depletion of INPs via activation and changes due to transport. However, internal aerosol microphysical processes (e.g., coagulation, condensation growth) are not simulated."*

*Line 278: Could you provide references for dust being negligible in the Arctic? There are many studies that do not support this claim (e.g. Boo et al. 2023; Creamean et al 2022)*

We have revised this statement to avoid generalization. We do not claim dust is universally negligible in the Arctic; rather, we clarify that in the specific background PSDs selected for this sensitivity study as given in response to general comments A and B, mineral dust number concentrations are low compared to organic and sea spray aerosols. This was based on the study by Earle et al. (2011)) where dust number concentrations are low compared to organic and SSA particles. We have added references (including Creamean et al. (2022)) to acknowledge that dust can be a major component during transport events, but maintaining the low-dust case to align with droplet number concentration of the SHEBA case study.

*Section 2.4: The description of sensitivity simulations is a bit confusing. E.g. CCR=0.3 is listed as sensitivity test, while based on the caption of figure 2, I would assume that the same CCR is applied in the CTRL simulation. If CCR is zero in CTRL simulation then this should be listen Table 4. If green profiles in Figure 2 concern only the sensitivity test and not CTRL case, this should be clearly explained in the caption*

We have clarified the description in Table 4 and Section 2.4. The CTRL run has a Cloud Cooling Rate (CCR) of 0. The CCR = 0.3 case is a specific sensitivity experiment. The caption for Figure

1 has been updated to explicitly state that the green profiles represent the evolved conditions under the specific sensitivity test (CCR = 0.3) after 10 hours, while the blue profiles represent the initial/CTRL conditions:

*Table 4: Parameter choices of the different aerosol-cloud 1D model simulation. Imm_CTRL: The CTRL run with the baseline settings for all IMF parameterizations, no perturbations, used as a reference. h_res_t and h_res_z: Simulation applying higher resolution with doubly refined vertical resolution (5 m) and much smaller time step (1 s).*

*Figure 1: Thermodynamic conditions applied in the minimalistic 1D aerosol-cloud model. From left to right: The temperature (T), the relative humidity (RH), supersaturation with respect to the ice ($S_{ice}$) and $\Delta a_w$. Blue and green lines represent the initial (t = 0 h) thermodynamic conditions and the thermodynamic conditions with cloud cooling rate of 0.3 ℃ h$^{-1}$ (CCR = 0.3 ℃ h$^{-1}$ sensitivity run) after 10 h, respectively. The blue shaded area denotes the cloud layer.*

*Line 285-287: Also it should be explicitly stated that CTRL simulation is run with a single aerosol type*

We have revised the text in Section 2.4 (Lines 446–449) to clarify this point:
*"We examine various simulation setups as given in Table 4. The thermodynamic conditions and cloud parameters from the LES baseline results of the SHEBA case study serve as the control run (hereafter referred to as CTRL) while applying the three different aerosol PSDs individually. Note that the baseline CTRL simulations are initialized with single aerosol types to isolate their specific freezing behaviors."*

*Line 308: also shown in Figure 2(?)*
The reviewer is correct. We have revised the text in Section 2.4 (Page 16, Line 473) to include this reference (note the change in figure number): *"The evolution of the temperature profiles for CTRL, and CCR = 0.3 are presented in Figure 1 and Fig. S2."*

*Line 310: clarify that each PSD corresponds to different aerosol type*
We have revised the text in Section 2.4 (Page 16, Lines 471–472 of the original manuscript) to read:
*"This results in 24 cases consisting of three aerosol PSDs (mineral dust, organic, and SSA respectively), four freezing parameterizations and two sets of cases..."*

*Line 503: Why organic INPs are more sensitive to temperature changes?*

We have added an explanation to Section 3.3.1 (Page 28, Lines 681–683) to clarify this mechanism:
*"...especially for organic aerosols which exhibit high sensitivity of $N_{INP}$ to temperature changes. This heightened sensitivity arises because the INAS density formulation for organic particles*

*(China et al., 2017) exhibits a significantly steeper slope with respect to temperature compared to the mineral dust parameterization in this temperature range."*

*Line 576: black instead of brown*
We have corrected the color descriptions in the text and captions to accurately match the figures (Page31, Line 756): *"applying dust, organic and SSA particles, given as brown, green, and blue lines, respectively."*

*Fig 7 is confusing. There is a light solid brown line in panels a and b, not include in the legend. Also dashed blue line is not visible in panel a*
We have redesigned Figure 7 (now Fig. 8) to improve legibility. We ensured that all plotted lines correspond to the legend, and improved the visibility of the dashed lines (representing entrainment).

[Figure]

**Figure 8: Vertical profiles of the change in number concentration of activatable INP ($\Delta N_{\mathrm{INP}}(z)$ in %) averaged over entire 10 h of simulation time. $\Delta N_{\mathrm{INP}}$ differs compared to the respective CTRL runs due to the change of cloud parameters (cloud cooling rate, cloud-top entrainment rate) applying dust, organic and SSA particles, given as black, green, and blue lines, respectively. Different immersion freezing parameterizations are applied including (a) ice nucleation number based (INN),**

(b) ice-nucleation active sites (INAS), (c) water-activity based immersion freezing model (ABIFM), and (d) ABIFM enabling subsaturated freezing (d) ABIFM*. Simulation results for changing cloud cooling rate (solid lines) and cloud-top entrainment rate (dashed lines) are shown. The blue shaded area denotes the cloud layer and the red line in the middle highlights the value of 0.

*Fig 8 is confusing. Is it the logarithmic scale that inhibits the demonstration of the whole vertical profile?*

Figure 8, now Fig. 9 in revised manuscript, shows ice crystal formation rate which can only proceed in the cloud layer when assuming saturated conditions. Except for panel (d) where freezing is allowed to proceed at slightly subsaturated conditions, i.e., from diluted aqueous solutions, a formation rate exists for below the cloud layer.

To prevent confusion, we have updated the figure caption to explicitly state this physical constraint. (Note: We have also updated the figure to make it clearer as discussed in the previous comment).

[Figure]

**Figure 9: As in Figure 7 but for the change in ice crystal formation rate ($\Delta dN_i/dt(z)$).**

**References**

Alpert, P. A., Kilthau, W. P., O'Brien, R. E., Moffet, R. C., Gilles, M. K., Wang, B., Laskin, A., Aller, J. Y., and Knopf, D. A.: Ice-nucleating agents in sea spray aerosol identified and quantified with a holistic multimodal freezing model, Sci. Adv., 8, eabq6842, https://doi.org/10.1126/sciadv.abq6842, 2022.

Arabas, S., Curtis, J. H., Silber, I., Fridlind, A. M., Knopf, D. A., West, M., and Riemer, N.: Immersion Freezing in Particle-Based Aerosol-Cloud Microphysics: A Probabilistic Perspective on Singular and Time-Dependent Models, J Adv Model Earth Sy, 17, e2024MS004770, https://doi.org/https://doi.org/10.1029/2024MS004770, 2025.

Böö, S., Ekman, A. M. L., Svensson, G., and Devasthale, A.: Transport of Mineral Dust Into the Arctic in Two Reanalysis Datasets of Atmospheric Composition, Tellus B: Chemical and Physical Meteorology, https://doi.org/10.16993/tellusb.1866, 2023.

Creamean, J. M., Barry, K., Hill, T. C. J., Hume, C., DeMott, P. J., Shupe, M. D., Dahlke, S., Willmes, S., Schmale, J., Beck, I., Hoppe, C. J. M., Fong, A., Chamberlain, E., Bowman, J., Scharien, R., and Persson, O.: Annual cycle observations of aerosols capable of ice formation in central Arctic clouds, Nat. Commun., 13, 3537, https://doi.org/10.1038/s41467-022-31182-x, 2022.

DeMott, P. J., Prenni, A. J., Liu, X., Kreidenweis, S. M., Petters, M. D., Twohy, C. H., Richardson, M. S., Eidhammer, T., and Rogers, D. C.: Predicting global atmospheric ice nuclei distributions and their impacts on climate, Proc. Nat. Acad. Sci., 107, 11217-11222, https://doi.org/10.1073/pnas.0910818107, 2010.

DeMott, P. J., Prenni, A. J., McMeeking, G. R., Sullivan, R. C., Petters, M. D., Tobo, Y., Niemand, M., Möhler, O., Snider, J. R., Wang, Z., and Kreidenweis, S. M.: Integrating laboratory and field data to quantify the immersion freezing ice nucleation activity of mineral dust particles, Atmos. Chem. Phys., 15, 393-409, https://doi.org/10.5194/acp-15-393-2015, 2015.

Earle, M. E., Liu, P. S. K., Strapp, J. W., Zelenyuk, A., Imre, D., McFarquhar, G. M., Shantz, N. C., and Leaitch, W. R.: Factors influencing the microphysics and radiative properties of liquid-dominated Arctic clouds: Insight from observations of aerosol and clouds during ISDAC, J. Geophys. Res.-Atmos., 116, 16, https://doi.org/10.1029/2011jd015887, 2011.

Fridlind, A. M. and Ackerman, A. S.: Simulations of Arctic Mixed-Phase Boundary Layer Clouds: Advances in Understanding and Outstanding Questions, in: Mixed-Phase Clouds, edited by: Andronache, C., Elsevier, 153-183, https://doi.org/10.1016/b978-0-12-810549-8.00007-6, 2018.

Hiranuma, N., Brooks, S. D., Moffet, R. C., Glen, A., Laskin, A., Gilles, M. K., Liu, P., Macdonald, A. M., Strapp, J. W., and McFarquhar, G. M.: Chemical characterization of individual particles and residuals of cloud droplets and ice crystals collected on board research aircraft in the ISDAC 2008 study, J. Geophys. Res.-Atmos., 118, 6564-6579, https://doi.org/10.1002/jgrd.50484, 2013.

Knopf, D. A., Silber, I., Riemer, N., Fridlind, A. M., and Ackerman, A. S.: A 1D Model for Nucleation of Ice From Aerosol Particles: An Application to a Mixed-Phase Arctic Stratus Cloud Layer, J. Adv. Model. Earth Sys., 15, e2023MS003663, https://doi.org/10.1029/2023MS003663, 2023.

Luke, E. P., Yang, F., Kollias, P., Vogelmann, A. M., and Maahn, M.: New insights into ice multiplication using remote-sensing observations of slightly supercooled mixed-phase clouds in the Arctic, Proc. Nat. Acad. Sci., 118, e2021387118, https://doi.org/10.1073/pnas.2021387118, 2021.

Quinn, P. K., Coffman, D. J., Johnson, J. E., Upchurch, L. M., and Bates, T. S.: Small fraction of marine cloud condensation nuclei made up of sea spray aerosol, Nature Geosci., 10, 674-+, https://doi.org/10.1038/Ngeo3003, 2017.

Savre, J. and Ekman, A. M. L.: Large-eddy simulation of three mixed-phase cloud events during ISDAC: Conditions for persistent heterogeneous ice formation, J. Geophys. Res.-Atmos., 120, 7699-7725, https://doi.org/10.1002/2014jd023006, 2015.

Wex, H., Huang, L., Zhang, W., Hung, H., Traversi, R., Becagli, S., Sheesley, R. J., Moffett, C. E., Barrett, T. E., Bossi, R., Skov, H., Hünerbein, A., Lubitz, J., Löffler, M., Linke, O., Hartmann, M., Herenz, P., and Stratmann, F.: Annual variability of ice-nucleating particle concentrations at different Arctic locations, Atmos. Chem. Phys., 19, 5293-5311, https://doi.org/10.5194/acp-19-5293-2019, 2019.

---

## Author Comment (AC2)

**Response to Referee #2**

We thank reviewer 2 for thoughtful and constructive comments that helped us to improve the clarity and quality of our manuscript. Below we have provided a point-by-point response to each of the points raised by reviewer 2. The reviewer's comments are given in *italic black font* and our response in green with quotes in *italic font* style. The line numbers refer to the revised and highlighted manuscript file. Before addressing the specific points in detail, we would like to give a **general response** to comments given by both reviewers referring to the idealized nature of this study.

This work introduces the AC-1D model, which provides a novel framework to prognostically treat INP and ice crystal budgets while explicitly accounting for polydisperse aerosol inputs. A key capability of this model is the flexibility to apply different immersion freezing (IMF) parameterizations, including deterministic and time-dependent (CNT) schemes, to different aerosol types with unique particle size distributions.

By purpose, we chose to focus on INPs and ice crystal formation, and the role of supercooled water droplets in mixed-phase clouds is implicit to the model initialization of the thermodynamic state in the examined scenario. Cloud modeling has not yet been able to accurately determine the strength of the primary or secondary ice production (PIP or SIP) pathways. We believe this is in part due to the models' complexity trying to account for all different processes proceeding in the cloud. Here, we aim to assess the strength of the PIP in mixed-phase clouds since everything else follows from this. If we cannot constrain this aspect of the cloud sufficiently the remaining processes stay ambiguous.

During the development of this prognostic model, novel concepts and notions emerged including the concept of "activatable" INPs which reflect the maximum number of INPs available to form ice under given cloud conditions (not all activatable INPs necessarily form ice though) and the notion of an INP reservoir which is available to the cloud and, as we show, is defined by choice of freezing parameterization. This extends previous modeling studies.

In short, this bottom-up approach links specific polydisperse aerosol particle size distributions (PSDs) to their respective freezing behaviors, establishing the model as a robust testbed for investigating structural model uncertainties for user-provided cloud conditions (e.g., LES informed etc.). Specifically, the model architecture facilitates:

1. **Comprehensive prognostic treatment** of aerosol, INP, and ice crystal budgets, explicitly accounting for the loss of each property (e.g., via activation or sedimentation) to capture their temporal evolution.
2. **Flexible initialization** using realistic, polydisperse, and multi-component aerosol composition rather than simplified monodisperse inputs, as well as the ability to easily switch between prognostic and diagnostic modes or compare different freezing parameterizations simultaneously.

3. **Process-level diagnosis**, enabling the user to quantify detailed INP and ice crystal budgets (e.g., activation vs. sedimentation vs. entrainment) in response to user-defined perturbations in thermodynamic profiles, cooling rates, and microphysical parameters.

We developed this simplified model to isolate and quantify the structural uncertainty introduced by the choice of immersion freezing (IMF) parameterization. While we acknowledge that Arctic clouds are complex systems influenced by SIP and liquid-ice feedbacks (such as the Wegener-Bergeron-Findeisen process driving ice growth), including these processes in this initial study would obscure the primary signal we aim to investigate: the foundational uncertainty in ice crystal number concentrations due to different freezing parameterizations.

We make the following changes to the text to make these points clearer.

Abstract:

Line 15: We add the following statement:

"*We developed one-dimensional aerosol-cloud (AC-1D) model, which provides a novel framework to prognostically treat INP and ice crystal budgets while explicitly accounting for polydisperse and multicomponent aerosol that activate INPs following different freezing parameterizations.*"

We modified the text on line 155:

"*Building on these identified uncertainties in primary ice production (PIP), we now focus in detail on the broader characteristics of Arctic aerosol. In this study, we employ the AC-1D model as a prognostic tool designed to isolate the structural uncertainties in PIP. The impact of liquid-ice feedbacks (such as the Wegener-Bergeron-Findeisen process driving ice growth) and secondary ice production (SIP) on the ice crystal budget depends crucially on the accurate description of the PIP. A key feature of this framework is its ability to conduct comprehensive sensitivity analyses by coupling polydisperse and multicomponent aerosol inputs directly to the INP and ice budgets. This setup allows for the simultaneous, prognostic evaluation of fundamentally distinct IMF parameterizations, while permitting user-defined adjustments to thermodynamic profiles and cloud system parameters. By generating detailed process-level data, such as explicit INP and ice crystal budgets, the model serves as a robust testbed to determine how the choice of parameterization dictates the PIP and evolution of the INP reservoir in Arctic mixed-phase clouds (Knopf et al., 2023b; Arabas et al., 2025).*"

We thank the reviewer for carefully evaluating our manuscript. We think that our responses clarify the implementation and interpretation of the CNT framework within this model and its application. The comments allowed us to better communicate our approach taken.

**Major comments:**

*1.) My biggest issue with the study is that I have some conceptual problems with understanding how the time-dependency was done for the CNT parameterization scheme (for me, right now it seems not correct how the implementation for the CNT scheme is done, but that could be related to some missing information or misunderstanding on my side). The following points need more clarification in order to be able to understand the study:*

We appreciate the reviewer's detailed questions, which helped us realize that our description of the CNT implementation was not sufficiently clear in the original text.

We have added the following clarification to the **Introduction** (Line 58) to explicitly define the terminology and physical interpretation used in this study:

*"Singular (INAS-type) IMF parameterizations represent freezing as a cumulative function of temperature, assuming that a population of nucleation-active sites initiates freezing once sufficiently low temperatures are reached on an immersed particle surface. In this framework, the frozen fraction depends on temperature and aerosol properties (including particle number concentration and available surface area), and freezing is often treated as effectively instantaneous upon reaching the activation threshold. As a result, such schemes do not explicitly represent the observed dependence of freezing on cooling rate or time spent at a given temperature (Bigg, 1953; Pruppacher et al., 2010; Alpert et al., 2016; Knopf et al., 2020; Arabas et al., 2025).*

*CNT-based parameterizations, in contrast, represent immersion freezing as a rate-based process governed by aerosol-specific kinetic parameters (e.g., contact angle distributions or water-activity-dependent coefficients), rather than fixed active sites. In these formulations, the freezing rate scales with the available INP surface area, such that increasing INP surface area increases the rate of ice formation (Pruppacher and Klett, 2010; Knopf et al., 2020; Knopf et al., 2023a). Consequently, for constant thermodynamic conditions, the cumulative number of freezing events increases with elapsed time, consistent with laboratory observations (e.g., Biermann et al., 1996; Koop et al., 1997; Alpert and Knopf, 2016; Knopf et al., 2020; Deck et al., 2022).*

*Although IMF parameterizations are often categorized as deterministic/singular versus stochastic, it is useful to distinguish the microscopic interpretation of nucleation from the numerical implementation used in atmospheric models. At the molecular scale, heterogeneous nucleation is frequently described as a random process; however, most Eulerian models predict the mean evolution of ice formation using deterministic population equations. In this work, the*

*CNT-based ABIFM scheme is implemented as a rate-based (memoryless) freezing formulation, in which the frozen fraction over a model time step depends only on the current thermodynamic conditions and aerosol surface area, not on the prior residence time of individual particles. This distinction helps avoid conflating physical randomness with Monte Carlo sampling approaches used in particle-resolved models (Shima et al., 2020; Arabas et al., 2025), and clarifies how time dependence enters the governing equations across different model hierarchies.''*

*- CNT is a time-dependent scheme, but nowhere in the manuscript is it clearly explained how this time-dependency is reflected in the model (if there is a time-tracking or time counting for each INP etc).*

We did not explicitly explain the CNT implementation in the governing equation in the main text but moved this description to the appendices. The time dependency of the heterogeneous nucleation rate coefficient $J_{\text{het}}$ (cm$^{-2}$ s$^{-1}$) is indirectly given in the governing equations (originally Appendix C, Equation C3) via the characteristic activation time scale, $\tau_{act}$.

$N_{\text{INP}}^{\text{Imm}}$ represents the activatable INP number concentration for the cloud, also termed the INP reservoir. For the CNT case, this reservoir is the total available aerosol number concentration at the current time step, $N_{aer}(t)$. The activation source term, $S_{\text{act}}$, is solved implicitly to ensure numerical stability. Based on Equation C3:

$$S_{act} = \frac{N_{aer}(t)}{\delta t + \tau_{act}}$$

Where $\tau_{act}$ is the characteristic activation time scale. $\tau_{act}$ depends inversely on the heterogeneous nucleation rate coefficient, $J_{\text{het}}$:

$$\tau_{\text{act}} = \frac{1}{J_{\text{het}}(T, a_{\text{w}})\pi d^2} \tag{1}$$

Hence, the equation reflects the time-dependency of the CNT implementation directly through $\tau_{act}$. To determine the number of newly formed ice crystals ($\Delta N_i$) over the current model time step ($\delta t$), we multiply the source rate by $\delta t$:

$$\Delta N_i = S_{act} \cdot \delta t = N_{aer}(t) \cdot \left(\frac{\delta t}{\delta t + \tau_{act}}\right) \tag{2}$$

This equation automatically scales the frozen fraction based on $\delta t$. When the freezing process is slow relative to the model time step (i.e., $\delta t \ll \tau_{act}$), the denominator is dominated by $\tau_{act}$, and the equation simplifies to a linear form:

$$\Delta N_i \approx N_{aer}(t) \cdot \left(\frac{\delta t}{\tau_{act}}\right) = N_{aer}(t) \cdot (J_{het}\pi d^2) \cdot \delta t \tag{3}$$

Because the ice formation scales linearly with $\delta t$ in equation 3, the result (ice formation) is independent of the chosen time step. Thus, individual time-tracking of particles is not required.

The equation 1 automatically scales the frozen fraction based on $\delta t$. When the freezing process is slow relative to the model time step, then $\delta t \ll \tau_{act}$, equation C3 simplifies to a linear form change). Consequently, the result is independent of the chosen model time step.

Here we provide an example. We calculated $\tau_{act}$ for a mineral dust particle (1.0 $\mu$m diameter) at the average cloud temperature (-19.0 °C) in simulated cases:

Based on ABIFM parameterization for dust (Eq. A10):

$$\log_{10}\left(J_{\text{het}}^{\text{dust}}(\Delta a_{\text{w}}(T))\right) = 22.62\Delta a_{\text{w}}(T) - 1.35 \tag{4}$$

$J_{\text{het}} \approx 295$ cm$^{-2}$ s$^{-1}$ and $\tau_{act} \approx 1.08$ x $10^5$ s (around 30 hours).

Since $\tau_{act}$ is orders of magnitude larger than the time step $\delta t$ (10 s), the denominator ($\delta t + \tau_{act}$) is dominated by $\tau_{act}$, and the equation simplifies to the linear form expected by the reviewer. We note that even in the coldest parts of the simulated cloud (-23 °C), $\tau_{act}$ remains significantly larger than $\delta t$ ($J_{\text{het}} \approx 1572$ cm$^{-2}$ s$^{-1}$ and $\tau_{act} \approx 2.02$ x $10^4$ s (around 5.6 hours). Consequently, the equation simplifies to a linear form like equation 3.

To better communicate the time-dependency of the CNT implementation, we have added a new section 2.4 named as "Numerical Implementation of *Temperature-Threshold and Rate-Based Freezing*". We insert the following text at Line 407.

**"2.4 Numerical Implementation of Temperature-Threshold and Rate-Based Freezing**

*To accurately represent the different physical bases of the parameterizations, we employ distinct numerical implementations. In the following, we adopt the terminology of 'singular' (representing temperature-threshold schemes like INN and INAS) and 'CNT-based' (representing rate-based schemes like ABIFM). For singular schemes (INN, INAS), the number of activated INPs is determined instantaneously based on the current thermodynamic conditions (temperature, surface area, number concentration).*

*In contrast, the CNT-based scheme (ABIFM) describes a rate of freezing. To ensure numerical stability while capturing the time-dependence without computationally expensive Lagrangian particle tracking, we calculate the activation source term ($S_{act}$) using an implicit form:*

$$S_{act} = \frac{N_{aer}(t)}{\delta t + \tau_{act}} \tag{1}$$

*Where $N_{aer}(t)$ is the available aerosol reservoir, $\delta t$ is the model time step, and the $\tau_{act}$ is the characteristic activation time scale defined by the heterogeneous nucleation rate coefficient ($J_{het}$) and particle surface area ($\pi d^2$):*

$$\tau_{act} = \frac{1}{J_{het}(T, a_w)\pi d^2} \tag{2}$$

*The change in ice crystal number concentration ($\Delta N_i$) over one time step is the product of the source rate and the time step. Substituting the definition of $\tau_{act}$, this can be expressed as:*

$$\Delta N_i = S_{act} \cdot \delta t = N_{aer}(t) \cdot \left(\frac{\delta t}{\delta t + \tau_{act}}\right) \tag{3}$$

*In the conditions relevant to this study, the characteristic freezing time ($\tau_{act}$) is typically orders of magnitude larger than the model time step ($\delta t = 10$ s). For example, mineral dust at -19°C, $\tau_{act} \approx 30$ hours. Under these conditions ($\delta t \ll \tau_{act}$), the denominator is dominated by $\tau_{act}$, and the equation simplifies to a linear form:*

$$\Delta N_i \approx N_{aer}(t) \cdot \left(\frac{\delta t}{\tau_{act}}\right) = N_{aer}(t) \cdot (J_{het}\pi d^2) \cdot \delta t \tag{4}$$

*This allows for the prognostic evolution of the INP reservoir without the need to track the time history of individual aerosol particles.*

*Finally, to verify that defining the total aerosol population as the INP reservoir does not lead to unrealistic instantaneous depletion of the aerosol population, we calculated the cumulative frozen fraction over the simulation duration. As detailed in Table S1 (Supplement), for representative cloud conditions (-19.0 °C), only ~28% of the most active reservoir (mineral dust) activates after 10 hours, while organic and sea spray activated fractions remain negligible (<1%). This confirms that treating the total aerosol population as the potential reservoir is physically robust and does not result in premature exhaustion of INPs."*

*- It is assumed that the whole aerosol distribution is effectively available INPs (INP reservoir) at the start (e.g. line 281-284 and line 481-482). That makes only sense if the aerosol particles had been exposed to the conditions in the cloud for a longer time (= the case of a long-existing cloud is studied). I wonder after how much time the whole aerosol distribution is activated for each aerosol species, and if this assumption is realistic? Can you please add calculations for each aerosol type and some representative/average temperatures of the case how long it takes to activate the whole aerosol distribution?*

We agree that the wording "the whole aerosol distribution is available as INPs" can be misread as implying instantaneous activation. That is not what is assumed in our model. In the CNT/ABIFM formulation, all aerosol-containing droplets are treated as eligible candidates for freezing (the "reservoir"), but only a small fraction freezes per model time step according to a temperature-dependent freezing rate.

Specifically, for a droplet containing an INP of surface area $A_{aer}$, CNT provides a heterogeneous nucleation rate *coefficient ($J_{het}$)*, which yields a per-droplet freezing rate:

$$k(T) = J_{\text{het}}(T) \cdot A_{aer} . \tag{4}$$

The unfrozen population then evolves according to the first-order loss law:

$$\frac{dN_{\text{unfrozen}}}{dt} = -k(T) \cdot N_{\text{unfrozen}} , \tag{5}$$

So, over a model time step ($\delta t$) the frozen fraction is

$$f_{\text{frozen}} = 1 - \exp(-k(T)\Delta t) . \tag{6}$$

This formulation is memoryless in the sense that the freezing probability over $\delta t$ depends only on the current thermodynamic conditions, not on how long a droplet has already remained unfrozen. Therefore, it is appropriate to treat all droplets present at any given time (including newly entrained ones) as part of the eligible reservoir, while the realized freezing over time is controlled by $k(T)$ and depletion of $N_{\text{unfrozen}}$ .

To address the reviewer's request for characteristic timescales, we computed the activated (frozen) fraction as a function of time using representative conditions from the case (e.g., T=-19 °C) for each aerosol type. These calculations show that depletion is not instantaneous; rather, the frozen fraction grows gradually with the characteristic timescale $\tau_{act} = 1/k(T)$. We now include these results in Table S1 and clarify the "reservoir" wording in the manuscript.

Table S1: Activated fraction of the total aerosol reservoir over time at the average cloud temperature (-19.0 °C).

| Aerosol Type | $\tau_{act}$ | Activated Fraction (1 h) | Activated Fraction (10 h) |
|---|---|---|---|
| Mineral Dust | 1.08 x $10^5$ s | 3.3% | 28.3% |
| Organic | > $10^7$ s | < 0.01% | <0.1% |
| SSA | > $10^6$ s | <0.1% | <1.0% |

We have added this clarification to the end of new **Section 2.4** and referenced a new **Table S1** in the Supplement containing these calculations.

 *"Finally, to verify that defining the total aerosol population as the INP reservoir does not lead to unrealistic instantaneous depletion of the aerosol population, we calculated the cumulative frozen fraction over the simulation duration. As detailed in Table S1 (Supplement), for representative cloud conditions (-19.0 °C), only ~28% of the most active reservoir (mineral dust) activates after 10 hours, while organic and sea spray activated fractions remain negligible (<1%). This confirms that treating the total aerosol population as the potential reservoir is physically robust and does not result in premature exhaustion of INPs"*

*- The CNT formulas described in the manuscript (line 944) seem strange. Why is it not N_INP^CNT=N_aer * J_het * delta t? From your framework here it is not obvious at all where the aerosol-specific nucleation rates are used. In Appendix C it is explained how the heterogeneous nucleation rates are used, but I did not fully understand it and I think this should be clear in the main text.*

We concur with the issue that the implementation of CNT formulas is not directly outlined in the main text. Please see our previous response above that hopefully alleviates this issue.

The formula suggested by the reviewer ($N\_INP^{CNT}=N\_aer * J\_het * delta\ t$) represents the explicit (linear) solution to the rate equation. We assume the reviewer omitted particle surface area ($A_{aer}$) for brevity in this formulation, as $J_{het}$ is defined per unit area (cm$^{-2}$ s$^{-1}$). With this inclusion, the equation becomes equation 3:

$$\Delta N_i \approx N_{aer}(t) \cdot (J_{het}\pi d^2) \cdot \delta t$$

As discussed in our previous response, our model uses an implicit Euler form to ensure numerical stability. That is, by explicit/implicit, we refer here to the numerical solver. To demonstrate that our implicit formulation is mathematically consistent with the reviewer's expectation in the physical regime of this study, we begin with our source term equation 1:

$$S_{act} = \frac{N_{aer}(t)}{\delta t + \tau_{act}}$$

The number of newly formed ice crystals ($\Delta N_i$) over one model time step ($\delta t$) is the rate multiplied by the time step:

$$\Delta N_i = S_{act} \cdot \delta t = N_{aer}(t) \cdot \left(\frac{\delta t}{\delta t + \tau_{act}}\right) \qquad (5)$$

We substitute the definition of the characteristic activation time scale, $\tau_{act} = \frac{1}{J_{het}A_{aer}}$:

$$\Delta N_i = N_{aer}(t) \cdot \left(\frac{\delta t}{\delta t + \frac{1}{J_{het}A_{aer}}}\right)$$

Multiplying the numerator and denominator of the fraction by $J_{het}A_{aer}$ yields:

$$\Delta N_i = N_{aer}(t) \cdot \left(\frac{J_{het}A_{aer}\delta t}{J_{het}A_{aer}\delta t + 1}\right)$$

In the example case, where $J_{het}A_{aer}\delta t \ll 1$, the denominator approaches 1, and the equation simplifies to the linear form the reviewer expects:

$$\Delta N_i \approx N_{aer} \cdot J_{het} \cdot A_{aer} \cdot \delta t \qquad (6)$$

Please refer to the new text for Section 2.4, which is provided in full in the response to Major Comment 1.1.

*- In line 522, it says that for the CNT based approach, the entrainment directly adds to the total aerosol particle production = INP reservoir. How can that be true? The entrained aerosols are "fresh" in the cloud (so starting at timestep 0) and therefore have not been exposed to the conditions for such a long time that all are activated as INPs (if not, the calculation requestion above shows times below 10 s). Otherwise, the time would need to be taken into account. How is the time dependence dealt with in case of the entrained aerosols for the CNT approach - how is the time-exposure tracked?*

In the CNT-based formulation, freezing is governed by a temperature-dependent rate that depends only on the current thermodynamic conditions. The fraction of particles that freeze over a model time step $\delta t$ is obtained by integrating this rate and does not depend on how long a particle has previously remained unfrozen.

Therefore, a particle entering the cloud via entrainment at t = 5 h has the same instantaneous freezing rate as a particle present since t = 0 h, provided both experience the same thermodynamic conditions. Differences in frozen fraction arise solely from the time-integrated exposure, not from particle age.

The probability of freezing depends on the current and the observation time interval (here, the model time step, $\delta t$), rather than the particle's history. Therefore, a particle entering the cloud via entrainment at t = 5 h has exactly the same freezing probability per time step as a particle that has been suspended in the cloud since t = 0h, provided they experience the same thermodynamic conditions.

To demonstrate that tracking individual time-exposure is not necessary, we can extend the mineral dust example from our previous response (mineral dust at -19°C, $\tau_{act} \approx 30$ h), assuming all particles have the same size and the composition (i.e., $A_{aer}$ and $J_{het}$ are the same for all particles).

First, we calculate the fraction of particles that freeze over 1 hour using the integrated rate equation:

$$P_{1h} = 1 - \exp\left(-\frac{3600}{\tau_{act}}\right) \approx 0.0328$$

Then we calculate the fraction of particles that freeze in a 10-second time step, removing them, and repeating this process 360 times (3600 s in total):

$$P_{10s} = 1 - \exp\left(-\frac{10}{\tau_{act}}\right)$$

$P_{\text{total}} = 1 - (1 - P_{10s})^{360} \approx 0.0328$ .

The results are identical. This proves that the total number of frozen particles depends only on the total time the population is present, not on the individual history of specific particles.

Thus, entrainment of particles adds to the INP reservoir without the need to discriminate based on the particles' residence times. On a 10 s time step, the freezing probability is effectively the same for all particles in the reservoir.

Please refer to the new Section 2.4 (presented under Major Comment 1.1), which explicitly explains why without the need to track the time history of individual aerosol particles.

*I would suggest adding some explanations to the basic concepts used for CNT (including the requested calculations). It should also be highlighted that the case is reflecting a long-lived cloud, and the calculations are not done at the initialisation of the cloud but at a later stage. It could help to add a sketch of how the CNT scheme works.*

We hope that our responses to previous comments clarified the basic concepts and implementation of the CNT approach. We emphasize that the CNT formalism is valid for both short- and long-lived clouds, independent of the cloud time scale. Also, the activation of INPs and the size of the INP reservoir are valid from the very first time step (t = 0). There is no initialization period required at a later stage of the cloud.

In summary, accounting for above responses, we have made the following updates to the manuscript:

1. Updated Introduction: We have added clarification to the Introduction to explicitly define the terminology and physical interpretation used in this study.
2. Added Section 2.4: We have added a dedicated **Section 2.4** ("Numerical Implementation of Temperature-Threshold and Rate-Based Freezing"). This section explicitly defines the characteristic activation time scale ($\tau_{act}$) and the implicit Euler formulation used to calculate the source term $S_{act}$. We also included the representative calculation for mineral dust at average temperature of the cloud base to cloud top (-19.0 °C) to quantitatively demonstrate that the freezing process satisfies $\delta t \ll \tau_{act}$.
3. Updated Section 2.1: We have updated **Section 2.1** to highlight that the SHEBA case represents a persistent, long-lived mixed-phase boundary layer. We clarify that our CNT-based formulation is physically valid at both initialization (t = 0) and later stages. Because the characteristic freezing time (around 30 hours) is significantly longer than the time step, the model naturally handles the transition from the initial state to the evolved state.

4. Added Table S1: Finally, to address the reviewer's request for clarity on the time-evolution of the system, we have added **Table S1** in the Supplement (and referenced it in Section 2.4). This table provides the calculated cumulative frozen fractions for 1 hour and 10 hours at representative conditions (-19.0 °C). This quantitative demonstration serves the purpose of a schematic by explicitly showing that the reservoir assumption $N_{\text{INP}}^{\text{CNT (ABIFM)}}(d) = N_{\text{aer}}(d)$ remains robust throughout the 10-hour simulation. This demonstrates that only a small fraction transitions to ice crystals at each time step, while particles do not nucleate simply remain in the reservoir, available for activation in subsequent steps.

*2.) It seems that the basis of the different parameterization types (INN, INAS, CNT) is different, i.e. they don't use the same dataset to derive aerosol-specific parameters etc.. This is probably a necessity given the availability and nature of different INP measurements and judging S6 the slope of the specific types is similar, which means that the used dataset had similar freezing characteristics. However, discussion on this aspect is totally missing in the article. Especially for the mixed external aerosol population, it could have implications that the parameterizations can differ a bit from each other for specific aerosol types.*

*Besides, aerosols like mineral dust are as such poorly defined. There are many different types of dust, and they come with a variety of ice nucleation activities. That should at least be mentioned in the manuscript if not critically discussed.*

We thank the reviewer for raising this important point regarding the underlying datasets used to derive the parameterizations. We agree that a fair comparison of the impact of different freezing parameterizations on the ice crystal budget is only possible if those were derived from the same experiments, i.e., same thermodynamic conditions and observed number of freezing events.

We have clarified in the manuscript that for mineral Dust, organic, and SSA, the INAS and ABIFM parameterizations used in this study are derived from identical datasets respectively, allowing us to isolate structural uncertainty from dataset variability:

For mineral must, INN, INAS, and ABIFM parameterizations are all based on the same observational data acquired in the AIDA cloud chamber (Niemand et al., 2012; Alpert and Knopf, 2016; Knopf et al., 2021). Applying the mineral dust INAS and ABIFM parameterization for the conditions of the AIDA chamber, both yield exactly the same frozen fraction as in Arabas et al. (2025).

For organic aerosol, both the INAS and ABIFM parameterizations are derived from the same laboratory dataset of Leonardite (standard humic acid) as described in (Knopf and Alpert 2013).

For SSA, both the INAS and ABIFM schemes are derived from the same recent mesocosm and field datasets described in Alpert et al. (2022).

The only exception is INN (DeMott et al., 2010) parameterization. We use DeMott et al. (2015), which is also largely informed by the same laboratory dust data. For Organic and SSA, we use DeMott et al. (2010), which represents a global background average. We include INN (DeMott et al., 2010) because it remains a standard benchmark in cloud modeling studies.

We agree with the reviewer that ambient mineral dust very likely reflects a wide variation of mineral types with different freezing propensities. We acknowledge previous studies on this matter by citing two review articles by Murray et al. (2011) and Kanji et al. (2017). We acknowledge that this simplification does not capture the full mineralogical complexity of atmospheric dust.

We have added a paragraph in section 2.3 (lines 338 to 360) immediately following the introduction of the parameterizations, to explicitly map the schemes to their underlying datasets and discuss mineralogical diversity:

*"To ensure a robust comparison of structural uncertainty, we selected parameterization pairs derived from identical underlying datasets where possible. For mineral dust, the INN (DeMott et al., 2015), INAS (Niemand et al., 2012) and ABIFM (Alpert and Knopf, 2016) schemes were all derived from the same laboratory experiments conducted at the AIDA cloud chamber. Because these schemes share a common experimental origin, the profound differences we observe in the AC-1D model results isolate the structural uncertainty inherent in how each scheme formulates the freezing process (e.g., time-dependence) rather than differences in the dust samples.*

*Similarly, for organic aerosol, both the INAS and ABIFM parameterizations are derived from the same laboratory dataset of Leonardite (standard humic acid) as described in Knopf et al. (2013); China et al. (2017). For SSA, both schemes are derived from the same mesocosm dataset (Alpert et al., 2022). In contrast, the INN parameterization (DeMott et al., 2010) is included as a benchmark representing averaged ambient INP measurements. Therefore, comparisons involving INN parameterization are not based on the same datasets, while comparisons between INAS and CNT remain structurally constrained by their common datasets.*

*Mineral dust is a broad category containing significant mineralogical diversity. As discussed in previous studies (eg., Murray et al., 2011; Kanji et al., 2017), ambient mineral dust reflects a wide variation of mineral types with different freezing propensities. While this study treats dust as a single species consistent with the bulk dust parameterizations of Niemand et al. (2012) and DeMott et al. (2015), we note that this simplification does not capture the full mineralogical complexity of atmospheric dust."*

*3.) Since the parameterizations are all based on measurements, their range can vary, which is also mentioned in lines 147-149. However, it is not really discussed anywhere if the temperature range of the parameterizations (derived from the temperature range of the measurements if no extrapolation is used) fits the case. What is the temperature range of the observations the*

*parameterizations are based on? I would suggest adding that in S6 (and add this information and discussion to the manuscript). In case a parameterization relies on extrapolation of the dataset, is that justified?*

We have verified the temperature validity ranges for all applied parameterizations against our simulation conditions (including the sensitivity tests), where the cloud layer operates between 250.4 K (-22.8 °C) and 256.9 K (-16.3 °C).

For clarity, we group our findings by aerosol type:

1. Mineral dust (INAS and ABIFM) and SSA (INAS and ABIFM): The derivation datasets for these schemes (Niemand et al., 2012; Knopf and Alpert, 2013; Alpert and Knopf, 2016) fully cover the simulation temperature range (valid up to -12 °C and -14 °C, respectively).
2. Ambient aerosol (INN D2010): This parameterization is valid for temperature up to -9 °C, covering our entire domain.
3. Organic aerosol (INAS and ABIFM): For the Leonardite data used to derive the organic schemes (Knopf and Alpert, 2013), the experimental data extends up to approximately 257 K (-16 °C) (see their Fig. 2D). Since our warmest simulation temperature is 257 K (-16.3 °C), our application falls well within the upper bound of the observational data spread.(Alpert et al., 2011)
4. Mineral dust: (INN D2015): The temperature range for D2015 is between 238 K to 252 K. Our simulation extends approximately 4 K warmer than this range.

We updated the paragraph regarding temperature validity in the end of Section 2.3 (line 349) and table 3 as well:

*"...Regarding the temperature validity of these schemes, our simulation conditions (-22.8 °C to -16.3 °C) fall strictly within the experimentally validated bounds for the Mineral Dust INAS/ABIFM schemes (Niemand et al., 2012; Knopf and Alpert, 2013), all SSA schemes (Alpert et al., 2022), and the ambient INN scheme (DeMott et al., 2010). For the Organic INAS and ABIFM schemes, the underlying laboratory data for Leonardite extends up to ~257 K (-16 °C) (Knopf and Alpert, 2013), effectively covering our simulation range. For the Mineral Dust INN scheme (DeMott et al., 2015), our simulation extends the temperature by approximately 4 K warmer than the original dataset. We consider this minor extrapolation justified as the parameterization is constrained to decay toward zero activity as temperature increases."*

*4.) This study is highly idealised (for example, by not having any precipitation mechanisms etc.). That is no problem at all. However, I find the context of an Arctic case a bit confusing, or I don't really see a big meaning in that focus. I would suggest to rather reduce this to being an idealised study with a certain input of meteorology and aerosols (which happens to be an Arctic case).*

*If the Arctic context is kept, I am missing some information that should be added. That is:- Was the used SHEBA case a spring or autumn case (the temperature seems to be a bit low for summer)? A reference to the used case (radiosonde profile etc.) should be added for more clarity.*

*- How were the observationally constrained LES results used, and what exactly?*

*- Was the LES simulation compared to observations and produce a realistic cloud case?*

*Additionally to that, I think more limitations of the idealised setup should be discussed - is the relative importance of freezing a trustworthy result or could it be related to the idealised setup (or is there no way to answer that question). One example here that should be mentioned/more clearly discussed is not having a WBF process or any precipitation mechanisms.*

We agree with the reviewer's assessment regarding the scope of the study. While the simulation setup is informed by the SHEBA intercomparison case, the primary value of this work lies in the idealized prognostic framework (AC-1D) and the evaluation of structural differences between freezing parameterizations, rather than a detailed historical case study of specific Arctic meteorology.

In response to this suggestion, we have reframed the manuscript to emphasize the model development and sensitivity analysis aspects. We have revised the Introduction to present the SHEBA case not as the primary scientific focus, but as a representative "mixed-phase cloud testbed" used to benchmark the model behavior under realistic, observationally constrained thermodynamic conditions. Regarding the specific case details requested, we clarify that the simulation utilizes the baseline conditions from the May 7, 1998 case of the Surface Heat Budget of the Arctic (SHEBA) campaign (Uttal et al., 2002; Fridlind et al., 2012; Fridlind et al., 2018). This represents an Arctic spring regime. Our model uses the thermodynamic profiles and cloud system parameters derived from the baseline Large-Eddy Simulation (LES) results of Fridlind et al. (2012). In that study, the LES simulations were extensively compared against ground-based radar, in-situ aircraft microphysics, and microwave radiometer data to ensure the thermodynamic structure and macrophysical properties (e.g., liquid water path, boundary layer depth) were realistic and consistent with observations. Furthermore, we emphasize that the exclusion of precipitation scavenging and the Wegener-Bergeron-Findeisen (WBF) process is an intentional design choice in this idealized framework. This isolation allows us to quantify the primary ice nucleation rates and reservoir depletion without the confounding effects of hydrometeor removal or phase partitioning, thereby directly attributing budget differences to the structural formulation of the parameterizations.

We have made several revisions to the text to clarify the study's scope and context:

For abstract, we added a statement defining the model's purpose (line 14):

*"We developed a one-dimensional aerosol-cloud (AC-1D) model, which provides a novel framework to prognostically treat INP and ice crystal budgets while explicitly accounting for*

*polydisperse and multicomponent aerosol that activate INPs following different freezing parameterizations."*

For introduction, we replaced the general model description with text explicitly framing the study as a prognostic tool (line 155):

*"Building on these identified uncertainties in primary ice production (PIP), we now focus in detail on the broader characteristics of Arctic aerosol. In this study, we employ the AC-1D model as a prognostic tool designed to isolate the structural uncertainties in primary ice production (PIP). The impact of liquid-ice feedbacks (such as the Wegener-Bergeron-Findeisen process) and secondary ice production (SIP) on the ice crystal budget depends crucially on the accurate description of the primary ice formation. We note that feedbacks such as the Wegener-Bergeron-Findeisen process primarily govern ice mass growth rather than number activation, and their thermodynamic effects are already manifested in the initialization profiles. A key feature of this framework is its ability to conduct comprehensive sensitivity analyses... By generating detailed process-level data, the model serves as a robust testbed to determine how the choice of parameterization dictates the PIP and evolution of the INP reservoir."*

For section 2.1 (The SHEBA case study), we added the specific date and season to clarify the context (line 209):

*"The specific case study from the SHEBA campaign used to define our model's thermodynamic profile is based on observations of a long-lived mixed-phase cloud on 7 May 1998 (Uttal et al., 2002; Fridlind et al., 2012; Fridlind and Ackerman, 2018). This late-spring seasonal context aligns well with the timing of the ISDAC (April 2008) and ICEALOT (March-April 2008) campaigns from which the aerosol data are derived."*

For section 2.3 (Model Setup), we added the justification for excluding other microphysical processes (line 331):

*"In this setup, PIP and ice crystal sedimentation are the dominant ice microphysical processes tracked. Processes such as the Wegener-Bergeron-Findeisen (WBF) process, riming, and Secondary Ice Production (SIP) are intentionally excluded to isolate the uncertainty in INP and ice crystal budgets arising directly from the choice of the immersion freezing parameterization."*

*5.) There is a comparison and critical discussion in relation to previous studies missing - this is not the first study comparing a CNT-based parameterization scheme with other schemes (and/or different aerosol types). What is different about the findings from this study from other studies (or are there similar findings)?*

We thank the reviewer for this suggestion. We agree that it is essential to contextualize our findings within the existing body of literature. Our introduction covers the modeling literature with respect to processes in mixed-phase clouds relevant to this study. While those studies apply

different freezing schemes and/or different aerosol types, few have applied different parameterization types for the same ice nucleation pathway within the same model to isolate structural uncertainty, despite recent findings showing that these choices can lead to orders of magnitude differences in ice production (Knopf et al., 2023b; Arabas et al., 2025). Thus, an "apples-to-apples" comparison has not yet been done for a given cloud scenario.

However, in the context of global models, the reviewer is correct that there have been studies applying different parameterizations. Global model studies such as Hoose et al. (2010), Wang et al. (2014), and Liu et al. (2005) have implemented CNT-based parameterizations in General Circulation Models (GCMs) and compared them with non-CNT schemes (e.g., (Lohmann et al., 2006). These studies generally focused on improving global agreement with observations by tuning parameters or refining aerosol composition (e.g., K-feldspar fractions), rather than explicitly analyzing the process-level divergence in INP reservoir depletion timescales that occurs when switching frameworks.

In the context of Large-Eddy Simulation (LES), Savre et al. (2015a) utilized a CNT-based approach with an evolving contact angle distribution. However, they focused on how the subset of active nuclei evolves via a shifting contact angle distribution, showing that the depletion of efficient nuclei regulates ice production, rather than comparing the fundamental reservoir definitions between schemes. Our study complements this work by isolating the impact of the reservoir size definition itself. We show that the choice of parameterization scheme (CNT vs. non-CNT) causes vast differences in the INP reservoir size (total aerosol versus active subset), thereby leading to divergences by orders of magnitude in their sensitivity to cloud-top entrainment and cloud cooling rate. Our key finding is that these differences represent a significant structural uncertainty for mixed-phase cloud modeling. The discrepancy between the schemes implies that modelers must exercise caution when selecting an INP parameterization, as the choice itself can alter the sensitivity of the simulated cloud to dynamic and thermodynamic factors.

We added a paragraph acknowledging previous GCM comparisons (e.g., Hoose et al., 2010; Wang et al., 2014) while highlighting the gap in process-level structural comparisons (Line 148).

*"While previous studies have implemented different ice nucleation parameterizations in global climate models (e.g., Liu and Penner, 2005; Hoose et al., 2010; Wang et al., 2014), these efforts generally focused on improving agreement with global observations by tuning parameters or refining aerosol composition. Few studies have isolated the structural uncertainty of the parameterization itself (singular vs. CNT-based) within a controlled, prognostic framework for the same cloud scenario, despite recent findings showing that these choices can lead to orders of magnitude differences in ice production (Knopf et al., 2023b; Arabas et al., 2025). Consequently, an 'apples-to-apples' comparison of how the fundamental choice of freezing framework dictates the INP reservoir dynamics and subsequent ice crystal budget remains necessary."*

*6.) In the summary and conclusion, it should be explained and discussed more what the consequences are of the findings of this study, especially in hindsight of implementing the CNT scheme into different models. How could this implementation look for less-idealised models (global models are mentioned, but how to treat the time aspect is not captured)? I think this part is crucial since it is a paper for GMD, where the model development and specifics should be of interest.*

We thank the reviewer for this suggestion regarding the application of our findings to less-idealized models. We have expanded the Summary and Conclusions to address the consequences of our findings for a broad range of atmospheric models, ranging from Cloud Resolving Models (CRMs) to Global Climate Models (GCMs).

From the findings of this and previous studies (Fridlind et al., 2012; Savre and Ekman, 2015a; Knopf et al., 2023b; Arabas et al., 2025) which treat INPs and ice crystals prognostically, it is clear that CRMs and GCMs should treat aerosol, INPs, and ice crystals prognostically for a more realistic representation of ice formation.

From a technical perspective, prognostic treatment of INPs when described by a singular scheme necessitates multi-dimensional state variables (tracers) to handle the particles PSD and activation temperature information. This becomes very costly when applied in a GCM. The application of CNT-based parameterizations removes the activation temperature dimension, thereby significantly reducing the required memory and computational cost.

From the scientific perspective, we also argue that switching from singular to CNT-based parameterizations could shift the primary driver of the cloud system: The singular schemes are more likely to create a regime dependent on continuous aerosol replenishment (entrainment), whereas CNT schemes create a regime that is more likely driven by the INP reservoir and the intrinsic CNT-based nucleation rate. This would suggest that GCMs using CNT may become less sensitive to immediate aerosol advection and more sensitive to the time evolution of the system.

We clarify that our implicit formulation (Section 2.4) handles the time aspect robustly for large GCM time steps (e.g., 30 min) by scaling the frozen fraction proportionally to the model time interval length. We verified that applying one 30-minute step yields statistically the same ice production as summing 180 consecutive 10-second steps. However, we emphasize that this scaling strictly requires prognostic treatment of INPs. Since a large time step removes a significant cumulative fraction of the INP reservoir, the model must explicitly track the depletion of INPs. Applying a CNT scheme to diagnostic aerosol fields could lead to unphysical runaway ice production.

We have added a new paragraph to the end of Section 5 (Summary and Conclusions, Line 1078) to explicitly discuss these implications:

*"Finally, these findings have significant implications for the implementation of ice nucleation in General Circulation Models (GCMs). First, the strong sensitivity of the ice crystal budget to*

*reservoir depletion confirms that GCMs should treat INPs prognostically rather than diagnostically, to avoid unphysical ice production in long-lived clouds.*

*Second, the choice of parameterization alters the computational burden. Prognostic application of singular schemes generally requires carrying additional tracers to track the subset of particles that have already nucleated at specific temperatures (to prevent 'double counting'). In contrast, the CNT approach follows a first-order rate law, eliminating the need for INP-specific history tracers and thus offering a computationally efficient pathway for prognostic implementation.*

*Regarding the time aspect, the implicit numerical formulation presented in Section 2.4 is robust for long GCM time steps (e.g., 30 min), since it integrates the same rate law over the full time interval and yields the same cumulative freezing as many shorter sub-steps. However, this stability relies on strict prognostic subtraction of activated INPs: applying time-dependent freezing rates to diagnostic aerosol fields over long time steps would lead to systematic overestimation of ice formation. Therefore, future GCM developments should prioritize coupling time-dependent nucleation schemes to prognostic aerosol and INP budgets."*

**Minor comments:**

*- Title of the paper: A large part of the paper is on the impact of different aerosol types on ice crystal budgets (and not only the freezing parameterization), as well as cloud parameters. That should be reflected in the title.*

We agree with the reviewer that the influence of aerosol characteristics is a central theme of the study alongside the parameterization choice. We have revised the title to explicitly reflect this scope.

The new title is ***"Prognostic simulations of mixed-phase clouds with model AC-1D v1.0: The impact of aerosol types and freezing parameterizations on ice crystal budgets"***

*- Citations in the introduction: often long lists but still not comprehensive, so please add "e.g." in front of the citation list.*
*- Citations in the introduction: rarely are primary sources cited, but newer articles/reviews etc..*

We have updated the citations in the Introduction to address the reviewer's concerns regarding comprehensiveness and historical context. We added "e.g.," to non-exhaustive lists and ensured that foundational primary sources are cited alongside newer reviews.

Specifically, we added "e.g.," to the list of studies on IMF dominance at Line 48, which now reads:

*"...dominant pathway of primary ice production (PIP) in MPCs (e.g., Ansmann et al., 2008; Prenni et al., 2009; ...)."*

We also added "e.g.," to the list of parameterization developments at Line 52, reading: *"...representing specific particle types have been developed (e.g., Bauer et al., 2008; Murray et al., 2012; ...)."*

Regarding primary sources, we added the foundational citation for biogenic ice nucleation at Line 92: *"...contribute to atmospheric ice nucleation (e.g., Schnell et al., 1975; Knopf et al., 2010; Wang et al., 2012; ...)."*

Finally, we added the seminal work by Bigg (1996) to the discussion of marine INP sources at Line 96: *"...remote marine regions far from the influence of continental INP sources (e.g., Bigg, 1953; Burrows et al., 2013; Wilson et al., 2015; ...)."*

*- Introduction: I would appreciate more explanation of how immersion freezing works under sub-saturated conditions, since no particles are immersed in that case?*

We clarify that immersion freezing under "sub-saturated" conditions refers to freezing that proceeds at slightly subsaturated conditions, i.e., from diluted aqueous solutions. Hygroscopic aerosol particles deliquesce below 100% relative humidity, forming a liquid environment in which the INP is immersed.

To address this, we have updated the description in Section 2.3 (line 365) to read: *" termed ABIFM\*, which permits nucleation to occur in subsaturated conditions (Knopf and Alpert, 2013)."*

*- line 63: CNT does not use the concept of INAS, but still assumes some aerosol-specific quantity (often expressed as a contact angle) that is a similar concept.*

We agree with the reviewer. While CNT does not use the ice nucleation active sites density ($n_\mathrm{s}$) concept, it fundamentally relies on aerosol-specific quantities to determine the freezing probability. In CNT, this is often expressed as a contact angle. In our specific ABIFM (CNT) implementation, these aerosol-specific properties are captured by the fitted coefficients ($m$ and $b$) which describe the relationship between the heterogeneous nucleation rate ($J_\mathrm{het}$) and the water activity criterion ($\Delta a_\mathrm{w}$). We have revised the text to acknowledge the CNT is governed by these aerosol specific parameters.

We have updated the sentence in the **Introduction (Line 65)** to read:
*" CNT-based parameterizations, in contrast, represent immersion freezing as a rate-based process governed by aerosol-specific kinetic parameters..."*

*- line 66: Not only constant supersaturation but also a constant temperature would be needed to activate more INP with time (in case of a temperature increase, other conditions would apply).*

We agree with the reviewer that constant temperature is also a requisite condition for isolating the time-dependence of freezing events. We have revised the sentence in the **Introduction (Line 68)** to read: *"Consequently, for constant thermodynamic conditions, the cumulative number of freezing events increases with elapsed time, consistent with laboratory observations..."*

*- line 111: The term recycling could be added here for clarity.*

We have revised the sentence at Line 123 (Introduction) to read:
*"...or restoration of some of the activated INP via complete ice crystal sublimation within a turbulently mixed layer (i.e., INP recycling) if there is an ice-subsaturated layer..."*

*- line 120-122: It is true that laboratory-based parameterisations are limited by the temperature range of the measurements/instruments. However, this is true for all three parameterization types used in this study (even the CNT parameterizations use aerosol-specific information based on laboratory data) - it seems a bit strange to only mention that in the context of singular parameterization schemes.*

We agree with the reviewer that the physical limitation regarding the valid temperature range of experimental data applies to all empirical parameterizations, including the CNT-based approaches used in this study. We have addressed the specific temperature validity ranges for the parameterizations used in our simulations in detail in our response to **Major Comment 3**, where we demonstrate that our conditions largely fall within or near valid experimental bounds. To address the reviewer's point in the text, we have revised the sentence in the Introduction to broaden the scope, acknowledging that extrapolation beyond derivation conditions is a universal challenge for parameterizations derived from laboratory experiments, not just singular ones.

We have revised the text on **Line 133 (Introduction)** to remove the specific restriction to singular schemes. The sentence now reads: *"It has been pointed out that **ice nucleation parameterizations derived from laboratory experiments** may not be applicable to all atmospherically relevant conditions (temperature and humidity) (e.g., Niemand et al., 2012; Hiranuma et al., 2014; Savre et al., 2015b; Kanji et al., 2017; Ullrich et al., 2017; Burrows et al., 2022; Knopf and Alpert, 2023a), partly due to instrument limitations and the limited amount of data collected."*

*- line 122-123: The last part of the sentence (slowly continuing ide formation) is not clear to me, please split the sentence and elaborate.*

We agree with the reviewer that the original phrasing was unclear. We have split the sentence to better distinguish between data limitations and physical consequences. We now explicitly refer to **"rate-based"** which allows for sustained ice production over long timescales, as described by

Westbrook and Illingworth (2013) and Yang et al. (2013). We have revised the text at **Line 122 (Introduction)** to read:

We have revised the text at **Line 135 (Introduction)** to read:

*"...partly due to instrument limitations and the limited amount of data collected (Burrows et al., 2022). Consequently, parameterizations may fail to capture **rate-based freezing**, which could sustain continuous ice crystal production (Westbrook and Illingworth, 2013; Yang et al., 2013)."*

*- line 138: Liquid phase as a fixed quantity- what does that mean? Constant and not changing by WBF etc., because not represented in the model? What is the limitation of this assumption? (see also major comment 4)*

We clarify that treating the liquid phase as a "fixed quantity" implies that the liquid water content profile is held time-invariant throughout the simulation. The reviewer is correct that this setup intentionally decouples ice growth from liquid water depletion, thereby excluding the Wegener-Bergeron-Findeisen (WBF) process. The limitation of this assumption is that the model cannot simulate cloud glaciation or desiccation. However, as discussed in our response to **Major Comment 4**, this is a strategic design choice to isolate the structural uncertainty in the Primary Ice Production (PIP) parameterizations without the confounding effects of liquid-phase feedbacks. Furthermore, we emphasize that while the liquid water content is fixed, its specific magnitude does not influence the ice nucleation rates or prescribed model dynamics; its primary role is to maintain the saturated thermodynamic state required for immersion freezing to proceed.

We have addressed this point in the major revisions to the manuscript text. Specifically, in **Section 2.3**, we explicitly state that the WBF process is excluded to isolate the INP budget. Furthermore, in the **Discussion (Section 4.1)**, we highlight that this simplification allows us to identify the parameterization choice as a foundational source of uncertainty. (Please refer to the full revised text in the response to Major Comment 4).

*- line 247: Does that mean that the whole aerosol distribution is assumed to be activated/in cloud droplets?*

Yes, the reviewer is correct. In this idealized setup, we assume that the entire aerosol population within the cloud layer is activated into liquid droplets. By assuming full activation, we eliminate the additional complexity and uncertainty associated with CCN activation kinetics (which would vary by particle hygroscopicity and updraft velocity). This allows us to strictly isolate the sensitivity of the ice crystal budget to the **immersion freezing parameterizations** themselves, rather than confounding the results with limiting CCN regimes.

We have updated the text in **Section 2.3 (line 361)** to clearly state this assumption as a boundary condition for the study:

*"We assume that all aerosol particles are sufficiently hygroscopic to activate as cloud droplets upon entering the cloud layer. Consequently, above the liquid cloud base, the entire aerosol population is assumed to be immersed within supercooled liquid droplets."*

*- line 251: You could add one line of explanation here why/how these arrays are needed.*

We agree with the reviewer that the purpose of these arrays should be explicit. We have added a sentence clarifying that for singular (singular) schemes, these arrays are required to **discretize the cumulative INP spectrum into a differential activation spectrum**. This allows the model to pre-calculate and store the specific number of INPs that activate within the **respective discretized intervals of temperature and particle diameter**, separate from the evolving environmental conditions.

We have updated the text in **Section 2.3** (line 367) to include this specific explanation immediately after the reference to Appendix B:

*"...The calculation of INP arrays for application of the singular freezing parameterizations can be found in Appendix B. **These arrays are constructed to discretize the cumulative INP spectrum into differential temperature bins, representing the specific subset of particles that activate within each temperature interval** ......"*

*- line 463: What determines the quasi-stable plateau for the CNT param? Why is there a plateau?*

We clarify that the quasi-stable plateau observed in the CNT simulations represents a dynamic equilibrium state. Unlike the singular schemes where the activatable reservoir is rapidly exhausted, the CNT framework maintains a large reservoir of activatable INPs that freeze over time via a rate-based process. This provides a continuous source of new ice crystals that balances the continuous sink of ice crystals via sedimentation. The plateau level is therefore determined by the balance between the CNT-based nucleation rate and the ice crystal fall speed.

We have updated the text in **Section 3.1** (line 638) to explicitly explain this mechanism.

*"Consequently, $N_i$ in CNT-based simulations reach a quasi-stable plateau after an initial increase and remain orders of magnitude higher compared to the case of singular schemes (Fig. 5 and Table 7). **This plateau represents a dynamic equilibrium where the continuous rate-based ice production (sustained by the large INP reservoir) is balanced by the removal of ice crystals via sedimentation.**"*

*- line 473: The aerosol composition is also important for the CNT parameterization.*

We agree. We have updated the text to explicitly state that aerosol composition is fundamental to the CNT framework, as it determines the heterogeneous nucleation rate coefficient.

We updated **Section 3.2 (line 659)** to read: *"...the continuous ice formation rates are governed by the heterogeneous ice nucleation rate coefficient,* **which is determined by the aerosol composition (Eqs. A10-A12)."**

*- line 482: Does subsequent mean recycled (sublimated) and entrained aerosols? How is the time aspect taken into account there (see also major comment 1)?*

We clarify that "subsequent" in this context refers to the **continuous** ice formation proceeding over time due to the rate-based nature of the CNT framework. It applies to the entire available INP reservoir, which includes both the initial population and particles added via entrainment. It does not refer to recycled aerosols, as ice sublimation is negligible in this case where the sub-cloud region is largely supersaturated with respect to ice. Regarding the time aspect for entrained aerosols, as detailed in our response to **Major Comment 1**, the rate-based (memoryless) nature of CNT means that freezing probability depends only on the current time step and thermodynamic conditions, not on the particle's history. Therefore, entrained particles immediately join the reservoir and freeze at the same rate as existing particles.

We have revised the text in **Section 3.2 (Line 659)** to replace "subsequent" with "continuous" to better reflect the ongoing rate-based process and avoid confusion regarding particle sources.

*"...However, the* **continuous** *ice formation rates are governed by the heterogeneous ice nucleation rate coefficient."*

*- line 501: Is it not a colder subset of the total aerosol population?*

We clarify that in the context of the singular (singular) implementation, we refer specifically to the **spectrum of activation temperatures** defined by the parameterization. As described by Equation B2 (and the INP arrays in Appendix B), the singular hypothesis assumes a fixed population of "activatable" INPs distributed across temperature bins. Therefore, as the cloud cools, the model accesses INPs active at colder temperatures of this pre-defined INP population (i.e., those specific particles pre-destined to freeze at the new, lower temperatures). We have maintained the terminology to be consistent with this singular framework.

We have retained the phrase "total INP population" but added a reference to the specific model equation to clarify the definition (line 678).

*"...meet the fixed activation thresholds of INPs active at lower temperatures of the total INP population* **(see Eq. B2)***."*

*- line 532-533: Why do the schemes react differently on sedimentation (or if not, why is it written/discussed this way)?*

We clarify that the physical treatment of sedimentation is identical for all parameterizations. Upon reviewing the quantitative sensitivity, we find that the **relative** impact of increasing the fall speed is consistent across all schemes. Comparing the control run ($v_f = 0.3$ m s$^{-1}$) to the sensitivity run ($v_f = 1.0$ m s$^{-1}$), the ice crystal number concentration decreases by a factor of approximately 3.5 (a reduction of around 70%) for both the singular and CNT-based parameterizations at the end of the simulation.

We have revised the text in **Section 3.3.3 (line 706)** to reflect this quantitative consistency:

*"Increasing the number-weighted ice crystal fall velocity ($v_f$) from 0.3 m s$^{-1}$ (CTRL) to 1.0 m s$^{-1}$ leads to a more rapid removal of ice crystals from the cloud layer. This results in substantially lower $N_i$ across **all** parameterizations and aerosol types (Fig. 5). Specifically, increasing the fall velocity reduces $N_i$ by approximately 70% (a factor of around 3.5) relative to the control run for both singular and CNT schemes. This consistent relative reduction demonstrates that sedimentation acts as a uniform sink for all parameterizations, while the absolute resulting concentrations are scaled by the strength of the INP source term."*

*- line 543: Is it not the lower half of the cloud - it looks like the peak is there?*

The reviewer is correct. While cloud cooling affects the thermodynamic profile throughout the layer, the peak change in $N_{INP}$ is indeed located in the lower half of the cloud layer. We have corrected the text to accurately describe this vertical structure, noting that the increase occurs throughout the cloud but maximizes in the lower portion.

We have revised the text in **Section 3.4.1 (Line 722)** to read:

*"...increasing $N_{INP}$ to values up to ~4.6 and ~21.4 times the original, respectively, **throughout the cloud layer, with the maximum increase observed in the lower half of the cloud layer** (Fig. 6a, b)."*

*- line 584: "more sensitive" compared to? (all the other schemes?)*

We clarify that the INN scheme is significantly more sensitive to entrainment **compared to the ABIFM scheme** (which shows negligible response). Unlike the INAS scheme, where cloud cooling is the clearly dominant driver, the INN scheme shows a sensitivity to entrainment that is comparable in magnitude to its sensitivity to cooling. We have updated the text to be precise about this comparison.

We have revised the text in **Section 3.4.2 (Line 765)** to read:

*"Conversely, the INN scheme is more sensitive to entrainment **compared to the ABIFM scheme**, which causes a nearly five-fold increase in $N_i$ at the cloud top (Fig. 8a)."*

*- line 537: Since the parameterizations were not derived from the same dataset the aerosol type and chosen parameterization are connected and can not be easily separated?*

We clarify that for the INAS and ABIFM parameterizations used in this study, the schemes **were** derived from the same experimental datasets for each aerosol type (as detailed in our response to **Major Comment 2**). Specifically, the mineral dust schemes share the AIDA chamber data, and the organic schemes share the laboratory dataset for Leonardite (standard humic acid) as described in **Knopf and Alpert (2013)** and fitted by **China et al. (2017)**. This experimental design allows us to explicitly separate the influence of the Aerosol Type from the Parameterization Framework (structural formulation). We acknowledge that INN is the exception, as it represents a global average, and have noted this distinction in the text.

We have updated the text in **Section 4.4 (line 817)** to reinforce this point:

*"...The results show that the dominant source of ice is governed by a combination of aerosol type and the chosen freezing parameterization. **Since the INAS and ABIFM schemes for specific aerosol types are derived from common datasets (Section 2.3), these differences can be attributed to the structural formulation of the parameterization rather than discrepancies in the underlying observations.**"*

*- line 735: Vast but also time-dependent INP reservoir (connecting to major comment 1).*

We agree with the reviewer. The significance of the CNT framework lies not only in the size of the reservoir but in the fact that this reservoir is subject to **rate-based freezing**. This distinguishes it from the singular approach where the reservoir is static at a given temperature. We have revised the text (line 916) to explicitly link the vast reservoir to the time-dependent kinetics, reinforcing the connection to the concepts discussed in Major Comment 1.

*"Conversely, CNT-based approaches, which treat all aerosol particles in their composition class as potential INPs activating continuously over time (e.g., Koop et al., 2000; Knopf and Alpert, 2013, 2023a), maintain a large INP reservoir **governed by time-dependent freezing kinetics**."*

*- line 856: The first sentence is the same as 3.) before?*

We agree with the reviewer that the first sentence of point 4 repeated the finding from point 3. We have revised the text to remove the redundancy while maintaining the contrast between the linear response of aerosol loading and the non-linear response of environmental parameters.

*We have revised the text in **Section 5 (Line 1041)** to read:*

*"4. **In contrast to the linear response observed for** $N_{aer}$, cloud cooling (CCR), $w_e$, and $v_f$ have non-linear effects, with the dominant process for ice production depending on the chosen parameterization."*

*- line 880: I don't understand the comment about the computational simplicity - it is not simple if the time that aerosol particles are experiencing certain conditions has to be tracked to capture the time dependence?*

We clarify that the "computational simplicity" refers specifically to the **elimination of temperature-dependent history variables**, rather than the complexity of the rate equation itself.

As detailed in our response to Major Comment 1, the CNT framework treats nucleation as a memoryless process. Therefore, the probability of freezing depends only on the current thermodynamic conditions and the model time step, not on the particle's history. Consequently, no Lagrangian time-tracking is required.

In contrast, a prognostic implementation of a singular scheme requires significantly more computational overhead in terms of memory. Because the singular hypothesis relies on fixed activation temperatures, the model must track the **history of the aerosol population** (using the temperature-dependent INP arrays described in Appendix B) to distinguish which fraction of particles has already activated. This is necessary to prevent "double-counting" active sites if the temperature fluctuates. The CNT-based approach, however, operates as a rate-based process applied to the total available aerosol surface area at the current time step. This eliminates the need to carry additional tracer arrays for the temperature dimension, thereby reducing the memory footprint of the microphysical scheme.

*- line 970: But only settling/sedimentation is accounted for in terms of vertical distribution (no updraft etc.)?*
There appears to be a misunderstanding regarding the processes affecting the INP reservoir versus the ice crystals. In our model, **sedimentation ($S_{i_{sed}}$) is not applied to the INP reservoir (see equation C1)**, as the fall speed of interstitial aerosol particles is negligible compared to the turbulent velocity scales. Sedimentation is applied **only** to the ice crystals (see equation C2). For the INP reservoir, the vertical distribution is governed exclusively by **turbulent mixing ($S_{mix}$, Equation C2)** which parameterizes the vertical transport by updrafts and downdrafts within the well-mixed boundary layer.

We have revised the text in **Appendix B** (line 1165) to explicitly clarify the role of mixing and the exclusion of sedimentation for INPs.

*"During simulation, the INP reservoir evolves through entrainment of new INPs from external sources, permanent removal through activation when environmental temperature drops to or below a bin's activation temperature, and vertical redistribution via **turbulent mixing (represented by mixing time scale $\tau_{mix}$). Note that sedimentation is neglected for the aerosol INP reservoir."***

*- Figure 1 vs. Table 1: From the table, it looks like dust and organic aerosol are missing the largest mode, but from the values and figure 1 it is clear it is missing the largest mode - leave D1 empty instead of D3 (use the same index for the same mode for all aerosols).*

We thank the reviewer for this suggestion to improve clarity. We agree that aligning the columns by physical mode makes the table much easier to interpret. We have restructured **Table 1** so that **Mode 1** corresponds to the Aitken mode (present only in SSA), while **Mode 2** and **Mode 3** correspond to the Accumulation and Coarse modes, respectively. Consequently, we have left the first column empty for the Mineral Dust and Organic aerosol rows, shifting their parameters to columns 2 and 3.

We have replaced **Table 1** where the columns are aligned by mode size.

*- Figure 3 and many others: the dashed style type used for the entrainment rate (or settling velocity) is very hard to see on a printout (and in the pdf). I appreciate the thought of having one line type for each sensitivity type, but would still recommend having the cooling rate dash for all sensitivities for better visibility.*

We agree with the reviewer that the original figure was too cluttered. To improve legibility and provide a comprehensive view of the system sensitivities, we have expanded **Figure 4** (formerly Figure 3) to **six panels** (a 3x2 grid). We have separated the results by parameterization type (columns) and simulation scenario (rows) to allow for clear visualization of the distinct behaviors. The top row (Panels a and b) displays the baseline **Control (CTRL)** evolution, establishing the reference behavior for all aerosol types. The middle row (Panels c and d) compares the CTRL run against the **Cloud Cooling Rate (CCR)** sensitivity test. The bottom row (Panels e and f) compares the CTRL run against the **Cloud-Top Entrainment ($w_e$)** sensitivity test. This new layout ensures that the dashed sensitivity lines are clearly visible against the solid control lines for all aerosol types and facilitates a direct visual comparison between the singular and CNT-based frameworks. We have updated all relevant text citations in Section 3 to correspond to these new panel assignments.

We have replaced the figure and updated the caption for **Figure 4** to reflect this 6-panel layout.

[Figure]

*Figure 4: Time series of simulated domain-averaged activatable INP number concentration ($N_{INP}$ in $L^{-1}$) separated by parameterization type: Singular schemes (INN, INAS; left column) and CNT schemes (ABIFM, ABIFM*; right column). The panels display the baseline control (CTRL) simulation (a, b), the sensitivity to cloud cooling rate (c, d), and the sensitivity to cloud-top entrainment rate and sensitivity experiment. Simulations are initialized with different aerosol PSDs (dust, organic, and SSA particles), immersion freezing parameterizations (INN, INAS, ABIFM, ABIFM*) and cloud parameters (cloud cooling rate, cloud-top entrainment rate (e, f). Brown, green, and blue lines represent the application of aerosol PSDs of mineral dust, organic, and SSA particles, respectively. Immersion freezing parameterizations are distinguished by symbols: INN (no symbols), INAS (cross), ABIFM (triangle), and ABIFM* (circle). In panels (a-b), the thin solid lines indicate results with the baseline control cloud parameters (CTRL). The dashed lines (c-d) denote results with the cloud cooling rate (CCR) of 0.3 °C $h^{-1}$ (CCR = 0.3) and the dash-dotted lines (e-f) show the results with the cloud-top entrainment rate ($w_e$) of 1 cm $s^{-1}$ ($w_e$ = 1.0).*

- *Figures 6, 7, 8: It would be helpful to have the parameterization name next to the label (a), (b)... of the plot.*

- *Figures 6 and 7: Are there lines missing? Not all plots show all of the sensitivity studies? Or is that because the lines are lying on top of each other? This has to be fixed or explained/mentioned.*

- *Figures 6 and 7: Use the same scale for all four subplots?*

- *Figure 6: The red color looks more like brown on the print-out.*

- *Figure 6: What does the red line with value 0 mean physically? Mention.*

- *Figure 8: Some lines are on the y-axis and hard to see.*

*- Figure 8: Since the order of colors is different, does that mean that the schemes are inconsistent between the parameterization schemes (see also major comment 2)?*

We thank the reviewer for the detailed feedback on the legibility and interpretation of the vertical profile figures. We have completely redesigned these figures (now **Figures 7, 8, and 9** in the revised manuscript) to address these concerns.

Regarding the "missing lines" and overlapping curves in **Figures 7 and 8** (formerly 6 and 7): The reviewer is observing a physical result of the model. In the singular schemes (Panels a and b), entrained INPs activate instantaneously upon entering the saturated cloud layer and are immediately converted to ice crystals; consequently, they do not accumulate in the INP reservoir, resulting in a zero change ($\Delta N_{INP} = 0$) that plots directly on the vertical axis. For CNT schemes (Panels c and d), while entrainment adds particles, the relative change to the massive total aerosol reservoir is quite small ($< 0.01\%$), also resulting in lines that overlap the zero axis. We have updated the figures to better visualize small changes.

Regarding colors and scales: We have standardized the color scheme across all figures. The "Red" line now strictly indicates the zero-change reference line (physically representing no deviation from the Control simulation). We have also unified the x-axis scales where appropriate to facilitate easier comparison between subplots.

Regarding the "inconsistency" in the order of colors in **Figure 9** (formerly Figure 8): The reviewer correctly notes that the hierarchy of aerosol efficacy shifts between parameterizations. This reflects the different mathematical structures of the schemes. For **INN (Panel a)**, the sensitivity depends on the absolute aerosol number concentration because the parameterization (DeMott et al., 2010) includes a concentration-dependent exponent. For **INAS and ABIFM (Panels b–d)**, the response is linear with aerosol loading (which cancels out in a percentage calculation), so the ranking is determined solely by the **slope** of the activation curve. Since the Organic parameterization (China et al., 2017) has a steeper slope than Dust or SSA in this temperature range, it exhibits the highest relative sensitivity in these panels. This variation is a key result demonstrating the structural uncertainty between the frameworks.

We have replaced Figures 7, 8, and 9 with the improved versions and updated their captions to explicitly address the "missing" lines, the zero reference, and the scientific reason for the shifting color.

[Figure]

*Figure 7: Vertical profiles of the change in number concentration of activatable INP ($\Delta N_{INP}(z)$ in %) averaged over entire 10 h of simulation time. $\Delta N_{INP}$ differs compared to the respective CTRL runs due to the change of cloud parameters (cloud cooling rate, cloud-top entrainment rate) applying dust, organic and SSA particles, given as black, green, and blue lines, respectively. Different immersion freezing parameterizations are applied including (a) ice nucleation number based (INN), (b) ice-nucleation active sites (INAS), (c) water-activity based immersion freezing model (ABIFM), and (d) ABIFM enabling subsaturated freezing (d) ABIFM\*. Simulation results for changing cloud cooling rate (solid lines) and cloud-top entrainment rate (dashed lines) are shown. The blue shaded area denotes the cloud layer and the red line in the middle indicates the zero-change reference line (physically representing no deviation from the control simulation).*

[Figure]

**Figure 8: As in Figure 7 but for the change in number concentration of ice crystals ($\Delta N_i(z)$ in %).**

[Figure]

**Figure 9: As in Figure 7 but for the change in ice crystal formation rate ($\Delta dN_i/dt(z)$).**

*- Figure 9 (figure and caption): External = Externally mixed?*

Corrected. Thank you.

*Table 2 (and related text): What does the mixing time scale mean/do? Move this information to the Appendix, where the mixing time is mentioned.*

We clarify that the mixing time scale ($\tau_{\text{mix}}$) represents the **large-scale vertical turbulent mixing** of the entire boundary layer, rather than small-scale entrainment mixing at the interface. Mechanistically, this parameter determines the rate at which vertical gradients in number concentrations (aerosol, INPs, and ice crystals) are homogenized. The mixing term functions by relaxing the concentration at every vertical level toward the vertical mean concentration of the

entire active layer. Thus, while entrainment acts as a source term adding new particles to the cloud top, $\tau_{\text{mix}}$ governs the subsequent vertical transport, determining how quickly those fresh particles are redistributed downwards to the cloud base.

We have updated the text in **Appendix C** (after Equation C6) to explicitly define the physical role of the mixing time scale:

*"...where $\tau_{mix}$ is the PBL mixing time scale. **Physically, $\tau_{mix}$ represents the large scale vertical turbulent mixing time scale. This parameter governs the rate at which vertical gradients in scalar concentrations are homogenized toward the boundary layer mean, redistributing entrained particles throughout the cloud layer."***

*- Table 2 (and related text): The text around the entraiment is not very clear. What is it that is entrained? From the discussion etc. I can read that it is referring to particles (and not water vapour, which would be an alternative interpretation, or both), but how many particles are entrained etc.?*

We clarify that in this 1D model framework, thermodynamic variables (temperature, humidity, liquid water) are prescribed based on the LES forcing and held constant; therefore, the thermodynamic effects of entrainment (e.g., drying) are implicit in the fixed background state. The model explicitly calculates entrainment **only** for the prognostic scalar variables: **Aerosols and INPs**.

The number of particles entrained is determined by the cloud-top entrainment rate ($w_e$) acting on the concentration gradient between the free troposphere and the cloud-top layer.

Furthermore, regarding the mathematical description, we realized that we mistakenly included an incorrect formulation for the entrainment source term in the original manuscript (Equation C4). The original equation showed a summation rather than a gradient and contained a typo in the denominator.

We have corrected this in the revised manuscript to accurately reflect the implicit numerical formulation used in the model. The correct equation calculates the flux based on the difference between the free-tropospheric and cloud-top concentrations, scaled by the entrainment velocity and grid height:

$$S_{\text{ent}}\left(z_m, t, k_*^{\text{Imm}}\right) = \frac{N_{\text{INP,FT}}^{\text{Imm}}(k_*^{\text{Imm}}) - N_{\text{INP}}^{\text{Imm}}(z_m, t, k_*^{\text{Imm}})}{\delta t + \frac{\delta z}{w_e}}$$

We also added some clarification to Appendix C (after Equation C4):

*"To ensure numerical stability and avoid violating the Courant-Friedrichs-Lewy (CFL) condition, the entrainment source term $S_{ent}$ is computed implicitly:*

$$S_{ent}(z_m, t, k_*^{Imm}) = \frac{N_{INP,FT}^{Imm}(k_*^{Imm}) - N_{INP}^{Imm}(z_m, t, k_*^{Imm})}{\delta t + \frac{\delta z}{w_e}}$$

*This term represents the source of new aerosol particles and activatable INPs entering the boundary layer from the free troposphere. The number of particles entrained per time step is determined by the concentration gradient between the free troposphere and the cloud top, scaled by the entrainment velocity."*

*- Table 2 (and related text): The value of the sedimentation rate does refer to approx. which ice crystal size/habit?*

We clarify that the fixed sedimentation rate applied in the model is a number-weighted average fall speed derived directly from the detailed bin-microphysics Large-Eddy Simulations (LES) of the SHEBA case by Fridlind et al. (2012). In the context of that study, this fall velocity corresponds approximately to **unrimed plate-like or dendritic crystals** with maximum dimensions in the range of **400–600 $\mu$m.** We have added a footnote to Table 2 to provide this physical context.

*- Line 397: It makes sense that the depletion of the INP reservoir is seen in the ice production rates, but I don't understand why the cooling rate is not seen at all in the control (it is getting colder/more particles are activated).*

We wish to clarify that in the **Control (CTRL)** simulation, the thermodynamic profile (temperature and humidity) is held **constant** over time (cloud cooling rate = 0 K h$^{-1}$). We make this clearer in Table 4".

We have also added a clarifying note in **Section 3.1**:

*"Figure 4 illustrates the 10-hour domain-averaged time series of $N_{INP}$ for control (CTRL) simulations **(where thermodynamic profiles are held time-invariant)** under different IMF schemes and aerosol types, with Figure S3, S4, S5 providing a detailed view of the initial 0.1 hours.*

*- Table 7: Better plot N_i/N_i^(10 s)? The table is difficult to comprehend.*

*- Table 8: Why is the ice crystal production const. 0 in INN_CTRL - is the cooling rate too small to lead to an activation of more aerosols?*

We have grouped these comments as they both relate to the clarity and formatting of the quantitative results. Regarding to the suggestion to replace Table 7 with a plot: we note that the temporal evolution of ice crystal number concentration is already visualized in **Figure 6** (formerly Figure 5). Since the normalization factor is a constant, a normalized plot would exhibit the exact same shape and trends as Figure 6, providing limited new information. We have chosen to retain the tabular format because it provides precise quantitative depletion factors (e.g., distinguishing between 0.02 and 0.05) which are discussed in the text but are difficult to read accurately from the logarithmic scale of the figure.

However, we agree that the original table layout was difficult to comprehend. To address this and to resolve the reviewer's observation regarding "zero" production in Table 8, we have **completely reformatted Tables 6, 7, and 8**. First, we restructured the layout to list time steps as sub-rows rather than separate columns, which significantly improves readability. Second, we converted all values to **scientific notation**. This reveals that the ice production rates in the INN_CTRL case (Table 8) are not zero, but simply very small values (driven by slow entrainment) that were previously rounded to "0.00." This formatting change ensures that both the rapid depletion regimes and the small background rates are clearly visible.

*Technical corrections:*

*- Units: There are sometimes line breaks in between units or between units and numbers. Use protected space to avoid this.*
Done.
*- line 85: The sentence is incomplete.*
Fixed. Thank you.
*- line 69-89: It would be better readable if written as a list or with a line break for each aerosol type.*
Done.
*- line 142: Incomplete sentence (or what does the "and statistically" refer to?) (?).*
Fixed. Thank you.
*- Figure 1 (c): it should be Large Accumulation in the legend instead of SSA.*
Fixed. Thank you.
*- Figure 1: Add the variable D in the x-axis label.*
Done.
*- line 239: Was that noted before?*
This text has been superseded by the major revision to Section 2.2, which now explicitly discusses the pack ice vs. open water context.
*- Table 4: Make categories (Control Run...) bold or emphasize a different way.*
Done.
*- line 320: Is the comma correct here? The sentence can be read a bit wrong because of that.*
Fixed. Thank you.
*- Figure 3: The x-axis is cut from the left panel.*

Fixed. Thank you.

*- line 867: Switch . with :.*

Done.

*- line 875: Typo, it should be "freezing".*

Fixed. Thank you.

*- line 910: Does it not have to be <?*

Corrected the inequality in the Heaviside function description to reflect that activation occurs when $T(z_i, t) < T_{INP}$.

*- S7 figure caption: Typo, should be "N_aer x 0.1".*

we clarify that the notation "$N_{aer}$ x 01" is the simulation identifier (as defined in Table 4) rather than a typo; however, to avoid confusion, we have updated the caption to explicitly state that this corresponds to a factor of 0.1.

"Figure S7. Results of the sensitivity tests involved changing the aerosol number concentration ($N_{aer}$ x 10 **(increasing aerosol concentration by a factor of 10)** and $N_{aer}$ x 01 **(decreasing aerosol concentration by a factor of 0.1)**. Solid lines represent…"